# Quantification of Blue Carbon in Salt Marshes of the Pacific Coast of Canada

Stephen G. Chastain[1], Karen E. Kohfeld[1,2], Marlow G. Pellatt[1,3], Carolina Olid[4,5], Maija Gailis[6]

[1]School of Resource & Environmental Management, Simon Fraser University, Burnaby, Canada V5A 1S6. Coast Salish Territories, xʷməθkʷəy̓əm (Musqueam), Skwxwú7mesh (Squamish) & səl̓ilwətaʔɬ (Tsleil-Waututh)
[2]School of Environmental Science, Simon Fraser University, Burnaby, Canada, V5A 1S6
[3]Parks Canada, Protected Areas Establishment and Conservation Directorate, Vancouver, British Columbia, Canada V6B 6B4
[4] Department of Forest Ecology and Management, Swedish University of Agricultural Science, Umeå, Sweden
[5]UB-Geomodels Research Institute, Departament de Dinàmica de la Terra i l'Oceà, Facultat de Ciències de la Terra, Universitat de Barcelona, 08028 Barcelona, Spain
[6]Environment and Climate Change Canada, Climate Change Branch, Ottawa, Ontario, Canada

*Correspondence to*: Karen E Kohfeld (kohfeld@sfu.ca)

**Abstract.** Tidal salt marshes are known to accumulate "blue carbon" at high rates relative to their surface area, which render these systems among the Earth's most efficient carbon (C) sinks. However, the potential for tidal salt marshes to mitigate global warming remains poorly constrained because of the lack of representative sampling of tidal marshes from around the globe, inadequate areal extent estimations, and inappropriate dating methods for accurately estimating C accumulation rates. Here we provide the first estimates of organic C storage and accumulation rates in salt marshes along the Pacific Coast of Canada, within the Clayoquot Sound UNESCO Biosphere Reserve and Pacific Rim National Park Reserve, a region currently underrepresented in global compilations. Within the context of other sites from the Pacific Coast of North America, these young Clayoquot Sound marshes have relatively low C stocks but are accumulating C at rates that are higher than the global average, with pronounced differences between high and low marsh habitats. The average C stock calculated during the past 30 years is $54 \pm 5$ Mg C ha$^{-1}$ (mean $\pm$ standard error), which accounts for 81 % of the C accumulated to the base of the marsh peat layer ($67 \pm 9$ Mg C ha$^{-1}$). The total C stock is just under one-third of previous global estimates of salt marsh C stocks, likely due to the shallow depth and young age of the marsh. In contrast, the average C accumulation rate (CAR) ($184 \pm 50$ g C m$^{-2}$ yr$^{-1}$ to the base of the peat layer) is higher than both CARs from salt marshes along the Pacific coast ($112 \pm 12$ g C m$^{-2}$ yr$^{-1}$) and global estimates ($91 \pm 7$ g C m$^{-2}$ yr$^{-1}$). This difference was even more pronounced when we considered individual marsh zones: CARs were significantly greater in high marsh ($303 \pm 45$ g C m$^{-2}$ yr$^{-1}$) compared to the low marsh sediments ($63 \pm 6$ g C m$^{-2}$ yr$^{-1}$), an observation unique to Clayoquot Sound among NE Pacific Coast marsh studies. We attribute low CARs in the low marsh zones to shallow-rooting vegetation, reduced terrestrial sediment inputs, negative relative sea level rise in the region, and enhanced erosional processes. Per-hectare, CARs in Clayoquot Sound marsh soils are approximately 2-7 times greater than C uptake rates based on net ecosystem productivity in Canadian boreal forests, which highlights their potential importance as C reservoirs and the need to consider their C accumulation capacity as a climate mitigation co-benefit when conserving for other salt marsh ecosystem services.

## 1 Introduction

Coastal, vegetated ecosystems, such as seagrass meadows, mangroves, and tidal salt marshes, have recently been recognized for their ability to store large amounts of "blue carbon" within their soils and sediments (Kennedy et al., 2013; Howard et al. 2014). While blue carbon ecosystems cover approximately 0.2 % of the ocean surface, they have been estimated to be responsible for up to 50 % of total coastal ocean carbon (C) burial (including estuaries and continental platforms) (Duarte et al. 2005), and their per-area C sequestration rate is substantially greater than that of terrestrial forest soils (McLeod et al. 2011). Globally, blue C ecosystems have been estimated to sequester between 75 and 224 Tg C yr$^{-1}$ (Duarte et al. 2013). For comparison, deep ocean organic carbon burial rates were recently estimated to be $20 \pm 6$ Tg C yr$^{-1}$ (Hayes et al. 2021), with an additional burial of 222 Tg C yr$^{-1}$ estimated for shelves and platforms shallower than 1000 m (Burdige, 2007). Due to this high C storage and accumulation rate capacity per unit area, coastal vegetated ecosystems have been suggested to play an important role in climate warming mitigation (Howard et al. 2017). However, when the ecosystem is degraded, the stored C can be released, and annual C uptake by the ecosystem ceases, resulting in losses of ecosystem services (McLeod et al. 2011, Pendleton et al 2012). Thus, to better inform policies that identify priority areas for conservation, more precise measurement of C stocks and accumulation potential are needed (Howard et al. 2017).

Global estimates of salt marsh area, C stocks, and C accumulation rates (CAR) are subject to large uncertainties. Duarte et al. (2013) noted a 20-fold uncertainty in global estimates of salt marsh area (ranging from 22,000-to 400,000 km$^2$) associated with ambiguous classification schemes for wetlands. For example, some classification systems consider freshwater and saltwater marshes in the same category (Duarte et al. 2013). Similarly, the estimated, global soil C stock of all salt marshes ranges

between 0.4 and 6.5 Pg C, a 16-fold range (Duarte et al. 2013). The average global CAR for salt marshes was estimated as 91 $\pm$ 19 g C m$^{-2}$ yr$^{-1}$ by the Intergovernmental Panel on Climate Change (IPCC) (Kennedy et al., 2013) and then as 245 $\pm$ 26 g C m$^{-2}$ yr$^{-1}$ in a subsequent global compilation (mean $\pm$ standard deviation) (Ouyang and Lee 2014). Furthermore, these reviews of salt marsh CAR estimates note disproportionate representation from certain areas of the world (Ouyang and Lee 2014; Chmura et al. 2003). Some areas, such as Europe and eastern North America, have dozens of CAR data points, while others, such as western North America, East Asia, and Australia are underrepresented. Regions such as Africa, India, and South America have fewer or no data at all. The high variability in CAR from site to site combined with the 20-fold uncertainty in global marsh area estimates result in global salt marsh CAR estimates ranging from 0.9 to 31.4 Tg C yr$^{-1}$ (Ouyang and Lee 2014). This 35-fold range is seven times greater than the global range for mangroves (Ouyang and Lee 2014; Donato et al. 2011).

The extensive use of $^{137}$Cs radioisotope or a marker horizon method for sediment dating also limits comparisons of CAR between studies. The difficulties of quantifying low concentrations of $^{137}$Cs in coastal sediments, and its low retention in high organic content sediments (Davis et al. 1984), limits the applicability of this technique to estimate CAR in salt marshes and leads to overestimates of sediment accumulation rates when compared to other dating methods such as $^{210}$Pb dating (e.g. Callaway et al. 2012; Johannessen and MacDonald 2016). Unlike $^{137}$Cs, the natural radionuclide $^{210}$Pb is highly retained in organic matter, which makes it suitable for establishing the chronology of sedimentary deposits accumulated over approximately the past 100 years (Krishnaswamy et al. 1971; Arias-Ortiz et al. 2018). Also, the lower detection limits of $^{210}$Pb reduce analytical errors in the derived chronologies below those obtained by using $^{137}$Cs providing, thus, better constrained recent CAR for salt marshes (e.g., Corbett and Walsh, 2015).

The Commission for Environmental Cooperation (CEC), a tri-national governmental organization promoting scientific cooperation between Canada, the United States, and Mexico, identified the Pacific coast of Canada as a significant data gap in terms of our knowledge of the aerial extent and quantification of blue C storage and accumulation. Additionally, a review of global salt marsh CAR data identified only eight sites on the entire Pacific coast of the continent, none of which were north of 38.2 °N (Ouyang and Lee 2014). This study aims to address this data gap by providing C stock and CAR from the Pacific Coast of Canada as a part of the Government of Canada's contribution to a continent-wide assessment of blue C mitigation potential. We sampled seven salt marshes within the United Nations Educational, Scientific, and Cultural Organization (UNESCO) Clayoquot Sound Biosphere Reserve, British Columbia's Tofino Mudflats Wildlife Management Area, and Pacific Rim National Park Reserve of Canada on Vancouver Island, British Columbia (Figure 1). These mesotidal estuarine marshes, often constrained in size by surrounding topography, are typical of the marshes found on the Pacific coast of British Columbia, and therefore provide a good representation of many of the tidal wetland ecosystems found here (Ryder et al. 2007). We calculated soil C stocks (per unit area) across the high and low marsh zones of each marsh and used $^{210}$Pb dating in a subset of the collected cores to quantify CAR. We then used the extent of high and low marsh areas from aerial imagery to estimate total C storage and total annual C accumulation. Finally, we place these new data within the context of CAR estimates from salt marshes in the NE Pacific region.

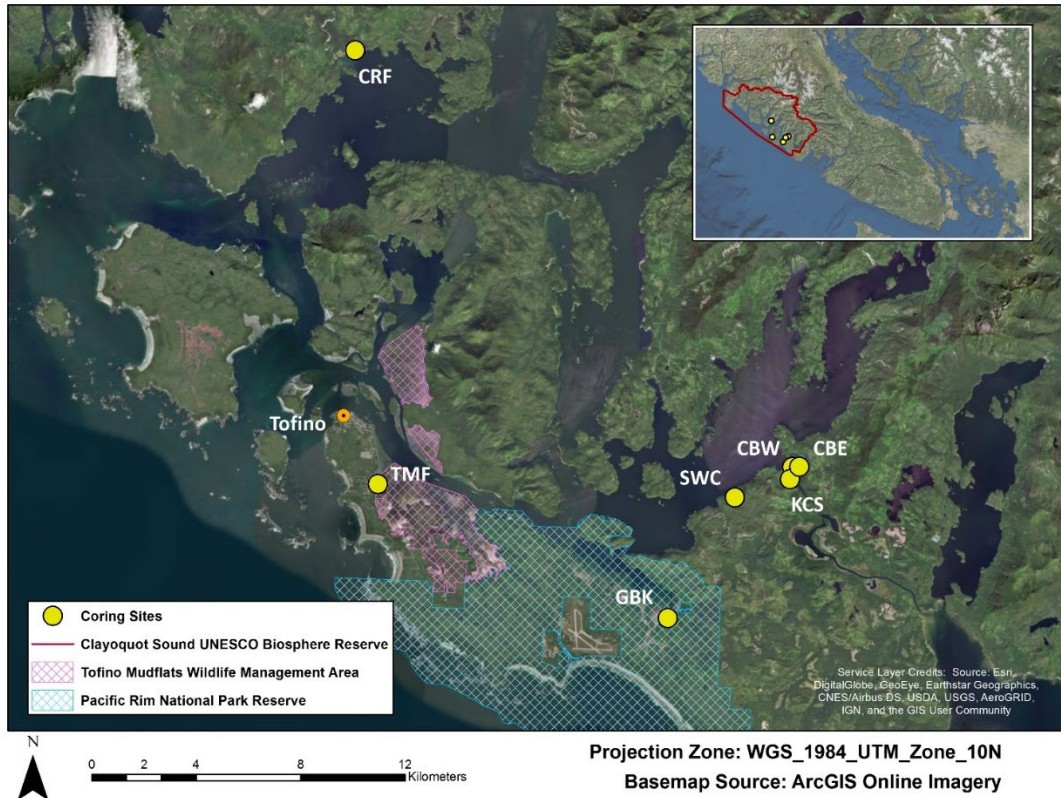

**Figure 1. Site locations. Cypress River Flats (CRF), Cannery Bay East (CBE), Cannery Bay West (CBW), Kennedy Cove South (KCS), Shipwreck Cove (SWC), Grice Bay at Kootowis Creek (GBK), and Tofino Mudflats (TMF). Pacific Rim National Park and Reserve (blue crosshatching) covers the southern portion of the map and the Tofino Mudflats Wildlife Management Area (pink crosshatching) covers portions of the southwestern area. The study region lies within the Clayoquot Sound UNESCO Biosphere Reserve, British Columbia, Canada (inset, outlined).**

## 2. Methods

### 2.1 Study Area

Clayoquot Sound is a complex system of inlets on the west coast of Vancouver Island, British Columbia, Canada. The sparsely-populated area includes several protected area designations, including the Long Beach Unit of Pacific Rim National Park and Reserve of Canada, the Province of British Columbia's Tofino Mudflats Wildlife Management Area, and the UNESCO Clayoquot Sound Biosphere Reserve, which protects 366,000 hectares of the west coast of Vancouver Island (Figure 1). The region is part of the temperate rainforest biome with high annual rainfall (3270 mm y$^{-1}$) and average annual temperature of 9.5

°C (Environment Canada, 1981-2010 averages for Tofino). While we were not able to measure the precise tidal range at each location, the mean tidal range in Tofino is approximately 2.7 m (DFO 2022).

These sites are typical of salt marshes along Canada's Pacific coast because they include small marshes along protected shorelines and bays as well as larger estuarine marshes near creeks and rivers. All sites were somewhat close to fluvial sources of varying size. Surface water salinity in the surrounding waters ranged from 5.9 at KCS to 24 in Grice Bay, and 29 at Roberts Point six km south of CRF (Postlethwaite et al. 2018).

## 2.2 Field Sampling

Within each marsh (n = 7), vegetation composition was noted in 50 x 50 cm quadrats before sediment cores (n = 34) were extracted during summer 2016 along linear transects perpendicular to the low tide shoreline, following the methodology of Howard et al. (2014). Coring spots were approximately evenly spaced along the transect (between 9 and 24 meters apart) from land to sea and spanned the low and high marsh zones. Core locations were chosen to avoid ditches and channels without organic soil accumulation; these ditches and channels made up a relatively small portion of total marsh surface area (<5 %). We note that, within this system, the partitioning between high and low marsh zones appears very closely linked with associated vegetation. Therefore, vegetation composition was recorded as an indicator of 'low' vs 'high' marsh zones (Porter 1982; Weinmann et al. 1984) around each coring spot. Coring spots were considered low marsh if the species *Triglochin maritima, Salicornia spp., Fucus ssp.* or *Ditschilis spicata* were present, and high marsh if it included *Plantago maritima, Deschampsia caespitosa, Grindelia integrifolia, Potentilla anserina, Lysimachia maritima,* or *Eleocharis ssp*. If a spot contained a mixture of these species, the majority percent cover of high or low marsh species was used to determine whether the spot was high or low marsh. *Carex lyngbyei* were often found throughout both strata and so were not considered unique to one zone. The high marsh species' ranges align approximately with the mean extreme high-water line of estuarine marshes in Clayoquot Sound, while low marsh encompasses elevations between the mean lower high water and the mean extreme high-water lines (Jefferson 1973 as cited in Deur 2000; Weinmann et al. 1984). This method was verified using detrended correspondence analysis, which showed that vegetation assemblages reflected distinct low and high marsh zones (Hill and Gauch 1980; Appendix B). This tight coupling between vegetation type and marsh zone has also been observed in other studies on the west coast of Vancouver Island (Deur et al. 2000) and studies in nearby Boundary Bay (Gailis et al. 2021). Vegetation associations related to salinity and inundation are well documented and commonly used in salt marsh delineation (MacKenzie and Moran. 2004).

Sediment cores were collected using a simple percussion coring technique in which a length of two-inch (57 mm) diameter, PVC vacuum tubing fitted with a plastic core catcher (AMS Inc.) was hammered into the ground until the depth of refusal (DoR). The DoR was reached between depths of 7 and 62 cm. In 19 of the 34 cores, the DoR was reached when the corer penetrated approximately 5-15 cm into layers of sand or gravel at the base of the marshes (see Supplemental Information, Table A2). A similar layer is reached in the eelgrass meadows that form seaward of these marshes (Postlethwaite et al., 2018). At these locations, our cores were able to sample the depth of initiation of marsh-related organic C accumulation. At eight of

the 34 locations, the cores reached depths where peat mixed substantially with sand or clay; the remaining seven cores ended in peat layers. With three exceptions, the DoR occurred at minimum %C.

At one site (GBK) a steel sledge corer (AMS Inc.) was used to extract four cores, but mechanical problems required switching to the simpler method described above. These four cores penetrated approximately 15-30 cm into the sand/gravel layer below the marsh, which contained minimal amounts of C. Marsh characteristics (e.g., average percent carbon, dry bulk density, soil C density, C stocks, and C accumulation rates) were determined by considering only the upper part of the profile containing peat and overlying sand, gravel, or clay, and these cores were excluded from our average estimates of the DoR.

While in the field, all cores were stored upright in portable coolers with ice packs until their return to the laboratory where they were photographed, logged, and stored long-term under refrigeration at 4˚C at the Parks Canada laboratory in Vancouver, British Columbia. Carbon and $^{210}$Pb lab work was completed 2-4 months after sample collection, between July and October 2016. Additional dating lab work was completed by approximately 30 months after collection.

    Use of the percussion corer resulted in sediment compaction during sample collection, which averaged about 20% across all

145 cores (range 0-55%) (Table A1). Nevertheless, we opted to use a percussion corer instead of a gouge corer because the percussion corer had a closed chamber with internal PVC sleeves. Our experience with these sediments has shown that a gouge corer would have been susceptible to disturbance and sediment mixing due to the nature of the open chamber of the corer. We also did not use a Russian corer because compaction would have been similar to what we experienced with the percussion corer, and we did not want to introduce increased contamination through the pivoting nature of the sampling chamber with the

150 Russian corer. Digging pits with a shovel was not an option as this study took place in a national park and biosphere reserve. We note below that correction for compaction was not necessary for estimation of C stocks because the C stocks were estimated directly from sediment cores and not from the overall depth of marsh soils (thus all carbon in the peat layer, regardless of compression, is included in the calculation). Furthermore, when we have estimated $^{210}$Pb-derived accumulation rates (Figure 6), we have done so in terms of cumulative mass (g cm$^{-2}$) instead of depth (e.g. Gifford and Roderick, 2003). When we do

need to account for compaction (e.g. Figure 3), we use a compaction factor as described in Gailis et al. (2021), estimated for each core by dividing the length of core penetration by the length of core recovered (Table A1).

### 2.3 Marsh Area Estimations

ArcMap 10.3 tools were used with 50 x 50 cm resolution aerial orthophotos taken in July 2014 (Government of British Columbia) to estimate the area of high and low marsh zones. The difference between high marsh and low marsh was delineated

by eye between darker-green, denser high marsh vegetation and lighter-green, salt-tolerant, and less-dense low marsh vegetation. This method was verified using the detrended correspondence analysis (e.g. Hill and Gauch 1980) of vegetation survey data and was found to accurately categorize 94 % of the cores into the correct marsh zone (see Supplemental Information, Appendix B). Similar approaches in Boundary Bay marsh in BC, Canada have resulted in a high degree of correspondence between marsh color and low vs high marsh delineation (Gailis et al., 2021).

## 2.4 Soil Carbon Content

Organic C content (%C = g C g$^{-1}$ marsh sediment) was determined using loss-on-ignition (%LOI) validated with CN Elemental and coulometric analysis (Froelich 1980). LOI analysis was performed on every 1 cm subsample by homogenizing samples with a mortar and pestle, combusting them at 550°C for four hours, weighing, and combusting again at 1000°C for two hours (Heiri et al. 2001). The percentage mass loss-on-ignition (%LOI) was estimated as:

$$\%LOI = \frac{DW_i - DW_f}{DW_i} \, x \, 100 \tag{1}$$

where $DW_i$ is initial dry weight [g] and $DW_f$ is the dry weight [g] after burning. For comparison, the %C was also estimated by measuring total C (%TC) and inorganic C (%IC) on a subset of 93 samples (see supplemental data). %TC was measured on these homogenized subsamples using dry combustion elemental analysis with an Elementar Elemental Analyzer for CN analysis at the University of British Columbia's Department of Earth, Ocean, and Atmospheric Sciences. The same subsamples were then analyzed for %IC using a UIC CM5014 $CO_2$ coulometer connected to a UIC CM5130 acidification module in the Climate, Oceans, and Paleo-Environments (COPE) laboratory at Simon Fraser University. Measurements of %IC were subtracted from the %TC measurements to estimate %C (Hodgson and Spooner 2016; Hedges and Stern 1984; Schumacher 2002; Howard et al. 2014). Inorganic C was negligible in all 93 of the subsamples analysed (max: 0.015 %) and assumed to be zero for all C calculation purposes. The strong correlation ($R^2 = 0.96$, $p < 0.05$) observed between %LOI and %C for this set of samples allowed us to use this relationship to convert %LOI to %C for all samples using the following equation, setting any negative %C value resulting from the use of a negative intercept equal to zero:

$$\%C = 0.44(\%LOI) - 1.80 \tag{2}$$

Soil C density (SCD [g C cm$^{-3}$]) was obtained as the product of the C content and the dry bulk density (DBD [g C cm$^{-3}$]):

$$SCD\left(\frac{g\,C}{cm^3}\right) = \left(\frac{\%C}{100}\right) \times DBD \tag{3}$$

where DBD was estimated after drying the sediment for no less than 72 hours at 60°C. Specifically, DBD was measured by bisecting the core while still inside its casing with a clean, serrated knife to avoid cross-contamination or inaccurate volume measurement that might be caused by extruding the core. Volume was precisely measured immediately afterward using a custom-designed, brass implement designed to remove a single cubic centimeter of soil at a time from the flat face of the bisected core.

## 2.5 Marsh profiles chronology

The naturally-occurring radionuclide [210]Pb was analysed in sediment from a subset of eight cores to determine recent C accumulation rates (CARs) (Appleby et al. 1992). One high marsh and low marsh core, each, from CBE, CRF, GBK, and TMF were chosen to represent the two marsh zones in various sites of different sizes. We used these vegetation communities to establish representativeness of the high and low marsh sites that we sampled for [210]Pb analysis. A set of subsamples (n = 6-7

for cores shorter than 20 cm, n=11-17 for longer cores) was sent to Core Scientific International (Winnipeg, Canada), Flett Research (Winnipeg, Canada), and MyCore Scientific (Dunrobin, Canada) to determine $^{210}$Pb content. Activities of $^{210}$Pb were determined by α-spectrometry through its granddaughter $^{210}$Po, assumed in secular equilibrium. The atmospheric or excess $^{210}$Pb ($^{210}$Pb$_{xs}$) fraction used to derive the age-depth model was determined as the difference between the total $^{210}$Pb activity and its parent nuclide $^{226}$Ra activity (Table A2). $^{226}$Ra activities were determined by α-spectrometry at Flett Research using calibrated geometries in a glass vessel, Spectech UCS 30 Alpha Scintillation Spectrometer purged with helium, and sealed for at least 11 days (Mathieu et al. 1988). Samples were sealed and stored for two hours before counting to ensure secular equilibrium of $^{226}$Ra daughters. $^{226}$Ra was determined through counting $^{222}$Rn activity for 60,000 seconds (Minimum Detectable Activity (MDA) = 0.0167 Bq kg$^{-1}$).

Because most of the cores did not show complete excess $^{210}$Pb profile (see Results), sediment chronologies were estimated using the Constant Flux: Constant Sedimentation (CF:CS) model (Krishnaswamy et al. 1971; Equation 4), which assumes a constant atmospheric deposition of $^{210}$Pb to the marsh surface and a constant mass accumulation rate (MAR, [g cm$^{-2}$ yr$^{-1}$]). Under these assumptions, the MAR can be obtained from the slope [cm$^2$ g$^{-1}$] of the linear best-fit line of the relationship between the natural log of the excess $^{210}$Pb activity against the cumulative mass [g cm$^{-2}$] using:

$$MAR\ (g\ cm^{-2}yr^{-1}) = \frac{0.0311\ (yr^{-1})}{slope\ (cm^2 g^{-1})} \tag{4}$$

where 0.0311 yr$^{-1}$ is the radioactive decay constant of $^{210}$Pb.

## 2.6 Carbon stocks

Organic C stocks [g cm$^{-2}$] are used as a proxy for total organic C storage in marshes and are calculated by integrating all peat layers ($n$) of the profile [cm]:

$$C\ stock\ (g\ C\ cm^{-2}) = \sum_{i=0}^{n} SCD_i \times 1\ cm \tag{5}$$

As most cores penetrated to a sand or gravel layer found at the base of the marsh (see Sect. 2.2), we are confident that the estimated C stocks ("based of peat C stocks", hereafter) capture all marsh-associated C emplaced since the formation of the marsh. Because C stocks are estimated directly from sediment cores and not from the overall depth of marsh soils, correction of core compaction was not necessary.

To facilitate comparison with other studies, we also report the mean C stock (Mg C ha$^{-1}$) to a depth of 20 cm for the region (20-cm C stocks, hereafter). Here we estimated the accumulated C to the corrected (uncompacted) depth of 20 cm in each core, with the caveat that 3 of the 34 cores did not reach a depth of 20 cm and therefore were not included in this calculation (Table A1).

Measurements of C stocks based on the accumulation of organic C up to certain depth, however, might give an incorrect impression of C sequestration capacity when comparing sites with different accumulation rates. The dating of the cores, however, allows one to circumvent this problem by normalizing C stocks to a certain age horizon. We therefore used a horizon

of 30 years before time of sampling to calculate these stocks (hereafter 30-year C stock). We chose this age-depth horizon because it represented the oldest [210]Pb date shared between the eight dated cores.

Finally, we estimated the total C storage (Mg C), in each marsh and for all seven marshes combined, by multiplying the average C stocks by the aerial extent of each marsh (Table 1). The total C storage was also estimated for low marsh and high marsh areas separately (Mg C). These estimates were made both to the depth of the base of the deepest peat (base of peat) layer and for the 20-cm horizon carbon stock estimates. Note that total C storage was not estimated for the 30-years horizon because we only had ages in four of the seven marshes, and thus there were three marshes where C storage could not be estimated to 30 years.

## 2.7 Carbon Accumulation Rates

We calculated C accumulation rates (CARs) in two ways: (1) to the "base of peat" depth of the deepest peat-containing layer overlying sand, gravel, or clay, and (2) to the greatest, common age horizon shared by all cores to allow comparisons between cores over equivalent timespans (30 years), which we call the "30-year CAR.". We include the whole core CAR calculation to assess any potential bias in the results due to higher overall C content in more recent accumulation and for easier comparison with other studies. This organic C fraction is calculated as:

$$\%C = \frac{\sum_{t=0}^{t=i\,yrs} C\,(g\,cm^2)}{\sum_{t=0}^{t=i\,yrs} m\,(g\,cm^2)} \tag{6}$$

where $i$ is either 30 years or the oldest peat-containing layer with detectable excess [210]Pb activity for the core. The CAR is the product of this %C and the MAR from Eq. 4:

$$CAR\,(g\,C\,cm^2\,yr^{-1}) = \%C_{core} \cdot MAR(g\,cm^{-2}yr^{-1}) \tag{7}$$

Average CARs were calculated for the high marsh, the low marsh, and the entire region, using the eight cores with [210]Pb dating, both for the whole cores as well as to the 30-year horizon (Table 2). Marsh-wide C accumulation rates *C Accumulation* were calculated by multiplying CAR by the marsh area, for the high marsh, low marsh, and then for the full marsh (Table 2):

$$C\,Accumulation\,(g\,C\,yr^{-1}) = \left[CAR_{high\,marsh}\,(g\,C\,m^{-2}yr^{-1}) \times area_{high\,marsh}(m^2) + CAR_{low\,marsh}\,(g\,C\,m^{-2}yr^{-1}) \times area_{low\,marsh}(m^2)\right] \tag{8}$$

Marsh-wide annual carbon accumulation was also estimated for the whole study area, by multiplying the average CARs (high marsh, low marsh, and total) by the area of the seven marshes (Table 2).

## 2.8 Statistical Analysis

We use a simple t-test to test for significant differences in marsh characteristics (depth of peat base, carbon content, DBD, and SCDs), C stocks, and CARs between high and low marsh zones. We also compare base of peat C stocks, 20-cm C stocks, 30-yr C stocks, as well as for CARs estimated to 30 years and to the base of peat layer.

**3 Results**

### 3.1 Soil Properties

With a few exceptions, marsh soils in Clayoquot Sound consisted of three layers separated by defined horizons: peat, and sand/clay layers (See Supplemental Information, Tables A2 and A3, Fig. A1). In all cores, organic C concentrations were highest in the surface layers (10-45 %) and decreased to lowest values (~ 0 %) in the deepest parts of the cores (Figure 2).

Depths of refusal in the 34 cores ranged from 7 cm in a low marsh core of CBW to a maximum of 62 cm at CBE. The average (± standard error (SE)) corrected depth of the deepest peat layer overlying a sand, gravel, or clay was $29 \pm 3$ cm for the high marsh sites and $17 \pm 3$ cm for the low marsh sites, with an overall average base of peat depth of $23 \pm 2$ cm (Fig. 3a). Differences in the deepest peat layer depths were statistically significant between the high and low marsh zones (p-value < 0.05). We note, however, that the 10 cores which did not penetrate into the sand, clay, or gravel layer were not used in these calculations.

While the average base of peat depth of high marsh cores is significantly higher than that of low marsh cores, no significant differences were found for average DBD, average % C, or average SCD (p > 0.05) (Fig. 3b-d). Carbon contents averaged $21 \pm 2\%$ in the high marsh and $18.5 \pm 2\%$ in the low marsh cores and $20 \pm 1\%$ overall (Fig. 3b) and did not vary significantly between the high and low marsh zones. DBDs and SCDs also showed similar average values and distributions in the high and low marsh cores (Fig. 3c and 3d). When corrected for compaction, soil carbon densities averaged $0.027 \pm 0.002$ g C cm$^{-3}$ for

all sites and ranged from 0.008 to 0.067 g C cm$^{-3}$.

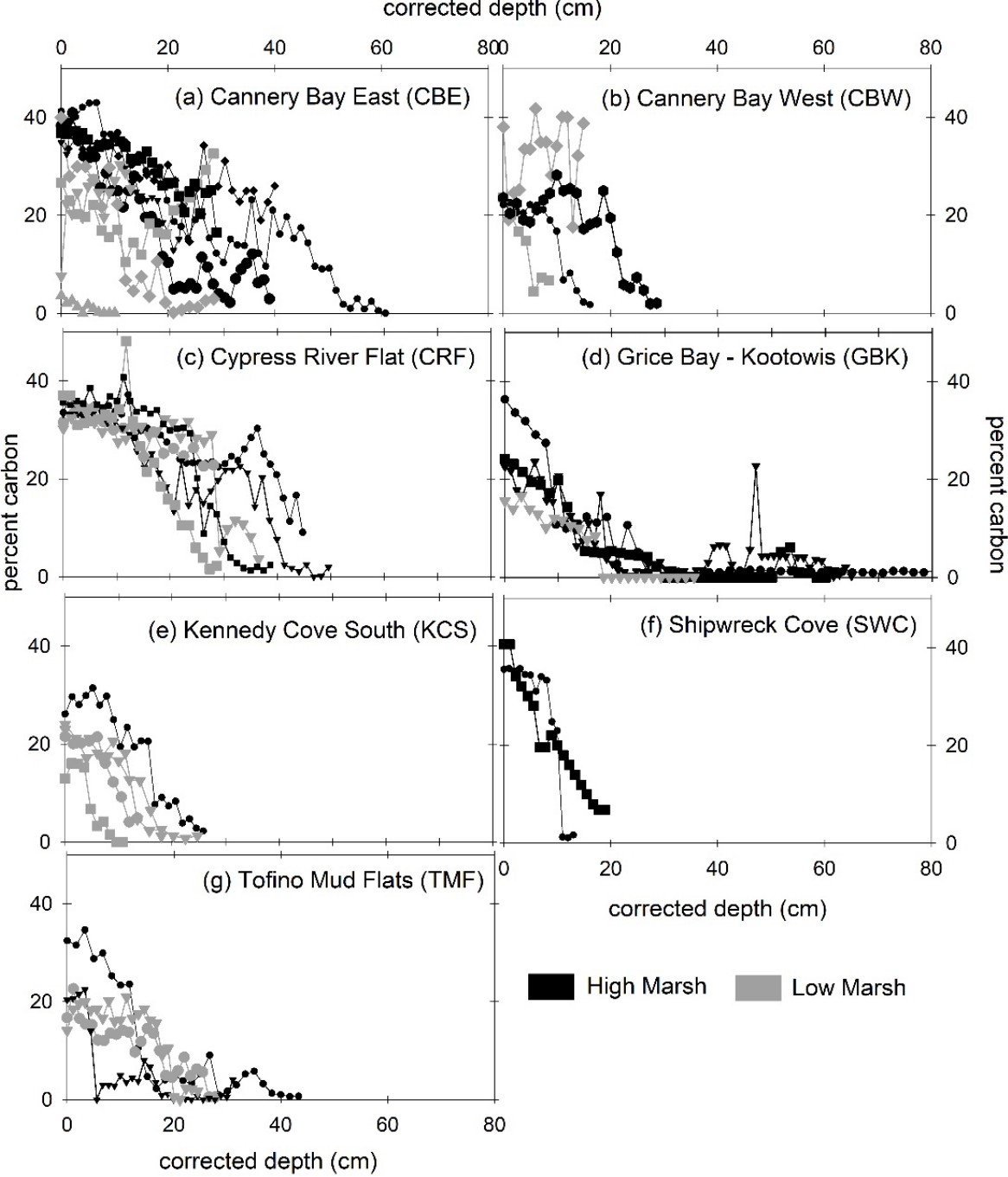

**Figure 2. Percent Carbon (vertical axis) versus depth (horizontal axis) for high marsh (black) and low marsh (grey) cores at all sites. Marsh name abbreviations are defined in Fig. 1. Depths have been corrected for core compaction.**

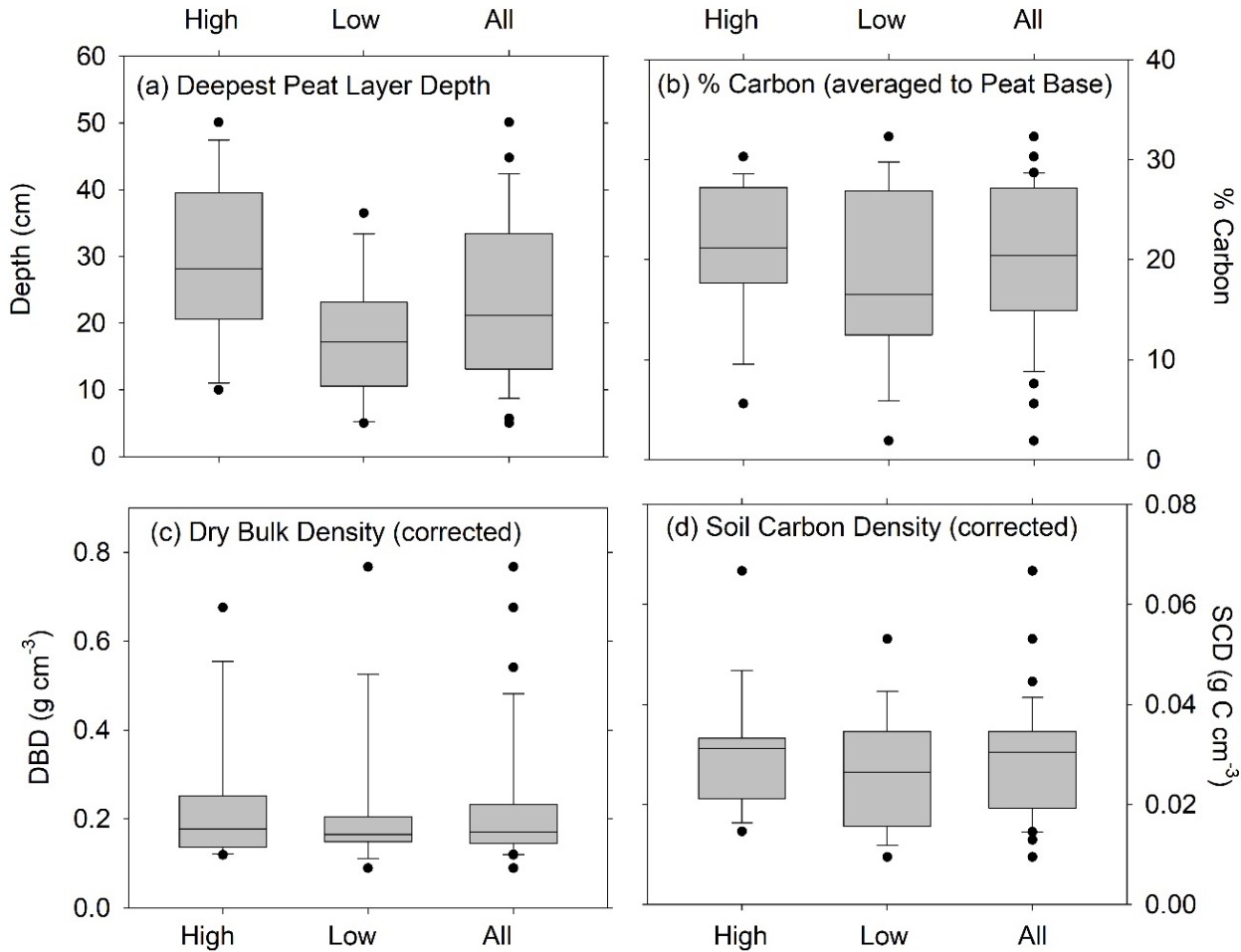

**Figure 3.** Distributions of (a) deepest depth of peat overlying sand, gravel, or clay (cm), (b) average percent carbon, (c) average dry bulk density (corrected for compaction), and (d) average soil carbon density (corrected for compaction). (b), (c), and (d) are estimated to the deepest peat depth for cores in the high marsh zones, low marsh zones, and all cores combined. Horizontal bar is the median; ends of the boxes represent the upper and lower quartiles; whiskers indicate minimum and maximum values while dots indicate outliers. Differences between depths of deepest peat layer for high and low marshes are statistically significant (t = 2.86, p-value = 0.0086). Unpaired t-tests show that the high and low marsh values are not statistically different for %C (t = 1.03, p-value = 0.31), DBD (t=0.25, p-value = 0.80) , and SCD (t=0.91, p-value = 0.37).

### 3.2 Carbon Stocks and Total Carbon Storage

The seven marshes ranged in size from 0.5 to 27 ha, with a total area of 47 ha (Table 1). The high marsh comprised 19-63 % of each individual marsh area and 58 % of the area of all seven marshes combined (Table 1).

The average C stock to the base peat layer for all cores was $67 \pm 9$ Mg C ha$^{-1}$ (mean $\pm$ SE). The base of peat C stock estimates for the high marshes ($80 \pm 14$ Mg C ha$^{-1}$) were 1.5-fold higher than those found in low marsh cores ($52 \pm 8$ Mg C ha$^{-1}$, unpaired t test, p < 0.05) (Fig. 4 and 5, Table 1).

Similar results were obtained when C stocks were calculated only for the upper 20 cm (Fig. 4, Table 1). The average 20-cm C stock was $54 \pm 5$ Mg C ha$^{-1}$. High marshes averaged $64 \pm 6$ Mg C ha$^{-1}$, which was statistically higher than the low marsh average of $43 \pm 7$ Mg C ha$^{-1}$ ($p < 0.05$). An average of 81 % of the total C stocks in a core accumulates in the top 20 cm of soil. Summing C stocks to the 30-year horizon of the eight dated cores results in an average of $56 \pm 14$ Mg C ha$^{-1}$ (Table 1; Fig. 4). Using the 30-year horizon accentuates the statistically significant difference in C stocks accumulating in the high vs low marsh zones ($p < 0.05$), with high marsh C stocks ($87 \pm 16$ Mg C ha$^{-1}$) up to ~ 4 times higher than those found in low marshes ($24 \pm 2$ Mg C ha$^{-1}$).

When combined with our estimates of marsh area, we calculate that the total C storage of these seven marshes was $4189 \pm 303$ Mg C, 69 % of which was stored in the high marsh (Table 2; Figure 5c). For only the top 20 cm, this total is $2764 \pm 149$ Mg C, 63 % of which accumulated in the high marsh zones. Finally, over the last 30 years, $2852 \pm 438$ Mg C have accumulated in these marshes, 84% of which accumulated in the high marsh zones (Table 1).

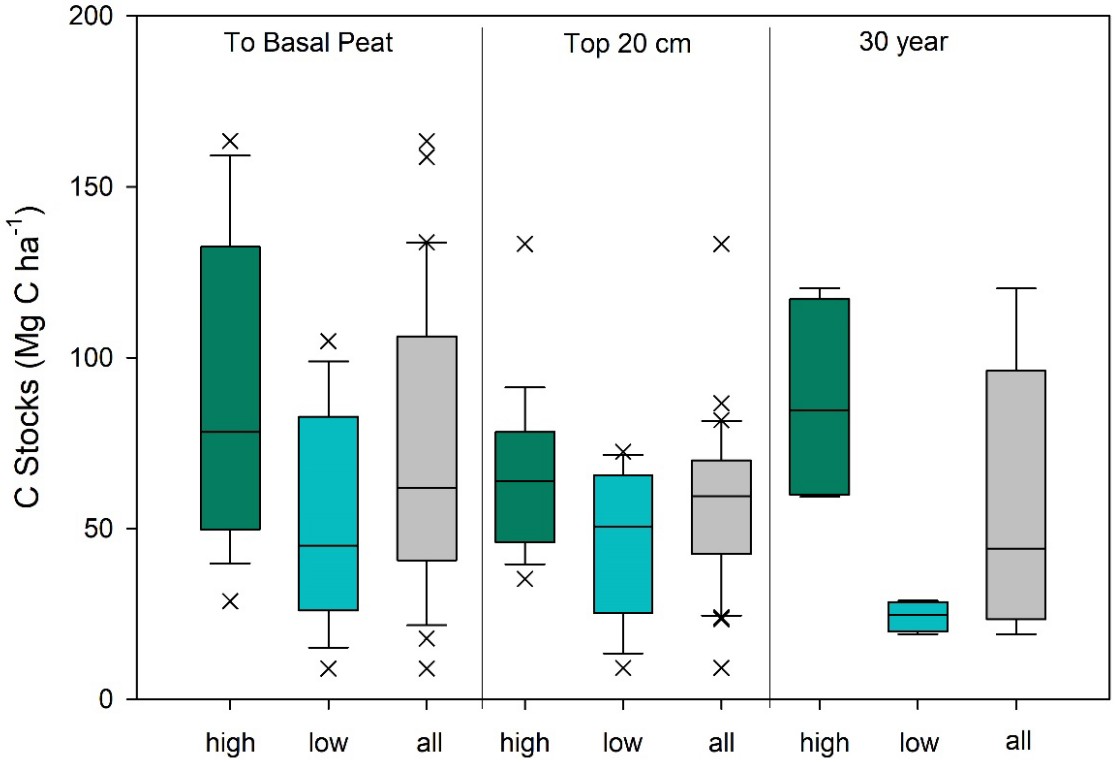

Figure 4. Carbon stock estimates for Clayoquot Sound for the high marsh zone (green), low marsh zone (blue), and combined (grey) for all cores. Estimates are made to the base of peat layer, for the surface 20 cm, and for the most recent 30 years of accretion. Note that 30-yr values are based on 4 cores each for high and low marshes (see Methods). Unpaired t-tests show statistically significant differences between the high and low marsh C stocks calculated to the base of peat depth (t = 2.91, p-value = 0.0065), 20-cm C stocks (t=2.20, p-value = 0.036) and 30-year C stocks (t = 3.95, p-value = 0.0074).

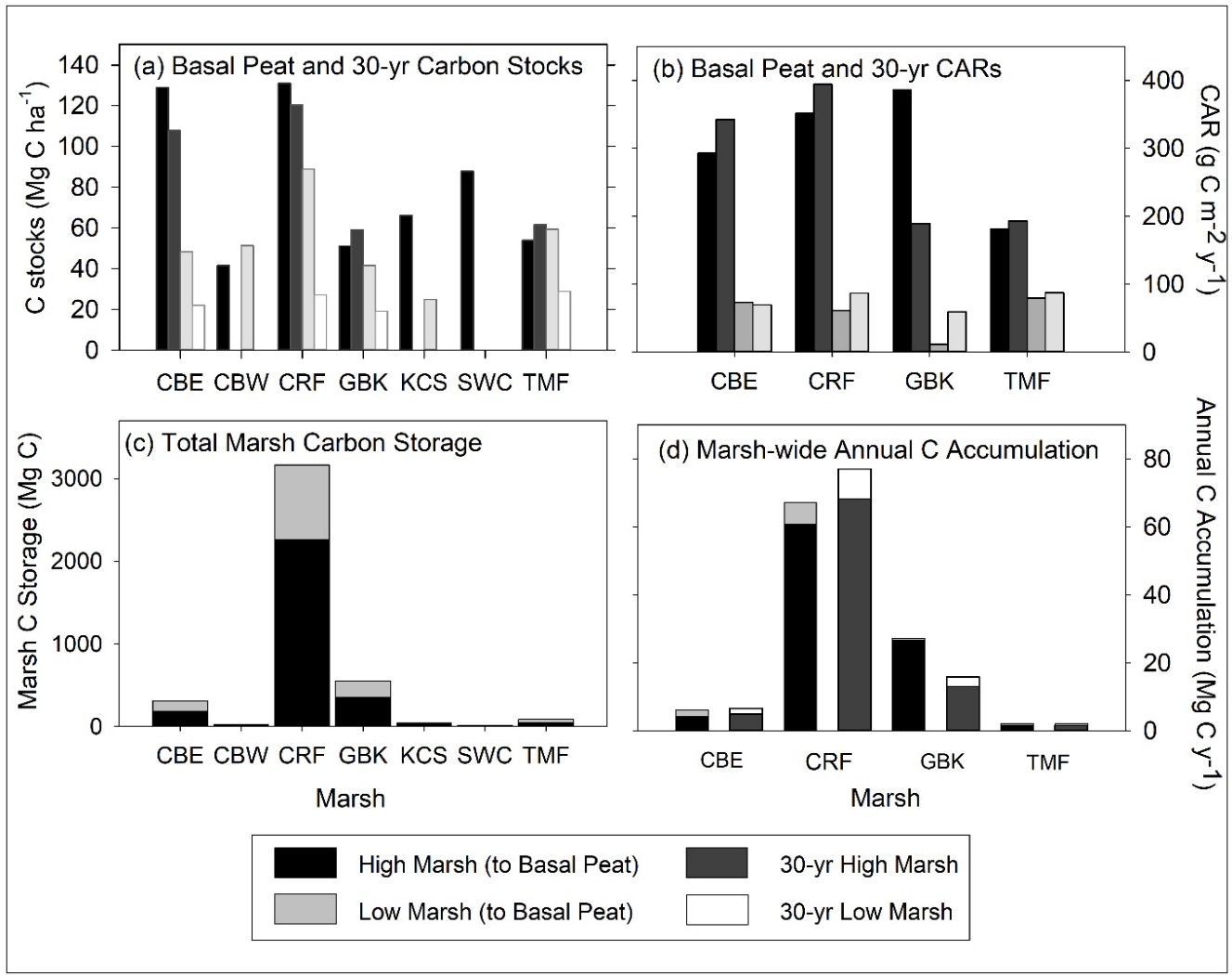


**Figure 5. (a)** High and low marsh carbon stocks (Mg C ha[-1]) measured to the base of peat layer and for the last 30 years; **(b)** High and low marsh carbon accumulation rates (CARs, g C m[-2] y[-1]) measured to the base of peat layer and for the last 30 years; **(c)** total marsh carbon (Mg C) estimate to the base of peat layer for high and low marsh zones of each marsh, and **(d)** marsh wide annual C accumulation (Mg C y-1) estimated to the base of peat layer and for the last 30 years. Paired differences between high and low marsh
CARs are statistically significant for CARs calculated to the base of peat layer (t=4.283, p-value = 0.0234) and for 30-yr CARs (t=4.043, p-value = 0.0272).(Statistical comparisons of C stocks are in Fig. 4).

**Table 1. Carbon Stocks**

| Site | Marsh Area (ha) | Carbon Stock (Mg C ha⁻¹), Ave ± SE | | | Total Carbon (Mg C), Ave ± SE | |
|---|---|---|---|---|---|---|
| | | To Peat Base | Top 20 cm | 30-yr | To Peat Base | Top 20 cm |
| *High Marsh* | | | | | | |
| CBE | 1.42 | $129 \pm 11$ | $74 \pm 4$ | 108 | $183 \pm 16$ | $106 \pm 6$ |
| CBW | 0.27 | $41 \pm 13$ | $47 \pm 0.4$ | | $11 \pm 3$ | $13 \pm 0.1$ |
| CRF | 17.31 | $131 \pm 16$ | $70 \pm 7$ | 120 | $2266 \pm 283$ | $1214 \pm 121$ |
| GBK | 6.9 | $51 \pm 6$ | $47 \pm 9$ | 59 | $353 \pm 41$ | $322 \pm 63$ |
| KCS | 0.47 | 66 | 66 | | 31 | 31 |
| SWC | 0.2 | $88 \pm 46$ | $89 \pm 45$ | | $18 \pm 9$ | $18 \pm 9$ |
| TMF | 0.82 | $54 \pm 9$ | $52 \pm 9$ | 61.5 | $44 \pm 7$ | $42.5 \pm 7.5$ |
| **Ave ± SE** | --- | $80 \pm 14$ | $64 \pm 6$ | $87 \pm 16$ | --- | --- |
| **Sum ± ε** | 27.4 | --- | --- | --- | $2906 \pm 287$ | $1746 \pm 137$ |
| *Low Marsh* | | | | | | |
| CBE | 2.57 | $46 \pm 16$ | $38 \pm 16$ | 22 | $118 \pm 41$ | $99 \pm 40$ |
| CBW | 0.23 | $51 \pm 34$ | 23 | | $12 \pm 8$ | 5 |
| CRF | 10.11 | $89 \pm 8$ | $66 \pm 4$ | 27 | $899 \pm 85$ | $663 \pm 43$ |
| GBK | 4.79 | 42 | 42 | 19 | 199 | 199 |
| KCS | 0.32 | $25 \pm 2$ | $24 \pm 4$ | | $8 \pm 0.8$ | $7.5 \pm 1$ |
| SWC | 0.83 | --- | --- | | --- | --- |
| TMF | 0.69 | $59 \pm 6$ | $63 \pm 6$ | 29 | $41 \pm 4$ | $44 \pm 4$ |
| **Ave ± SE** | --- | $52 \pm 8$ | $43 \pm 7$ | $24 \pm 2$ | --- | --- |
| **Sum ± ε** | 19.5 | --- | --- | | $1283 \pm 97$ | $1018 \pm 59$ |
| *Total Marsh* | | | | | | |
| **Ave ± SE** | --- | $67 \pm 9$ | $54 \pm 5$ | $56 \pm 14$ | --- | --- |
| **Sum ± ε** | 46.9 | --- | --- | | $4189 \pm 303$ | $2764 \pm 149$ |

Base of peat C stock was estimated by summing the mass of carbon from each 1-cm sample interval to the base of each core. Base of peat C stock values are averages of multiple cores, except where only one core was collected (e.g., high marsh KCS and low marsh GBK). The 30-yr high and low marsh average C stocks are based only on CBE, CRF, GBK, and TMF. 30-y marsh carbon (Mg C) is estimated by multiplying high marsh areas by average high marsh 30-yr C stocks ($2384 \pm 438$ Mg C), low marsh area by average low marsh 30-yr C stock ($468 \pm 19$ Mg C), which is summed to get a total of $2852 \pm 438$ Mg C.

## 3.3 Age-depth relationships and mass accumulation rates over time

Activities of $^{210}$Pb ranged from 2 to 372 Bq kg⁻¹, with maximum activities found at the surface. Activities of $^{226}$Ra ranged from 1.6-4.3 Bq kg⁻¹ and were subtracted from the $^{210}$Pb$_{total}$ to obtain the excess $^{210}$Pb ($^{210}$Pb$_{xs}$). For most of the cores, the $^{210}$Pb$_{sup}$ level was not reached. Only GBK 1-4 and TMF 1-2 showed a complete $^{210}$Pb$_{xs}$ profile.

Activities of $^{210}Pb_{xs}$ showed an almost monotonic decline with depth, suggesting that processes such as bioturbation, mixing, or disconformities have not disturbed the records, and the logarithmic $^{210}Pb_{xs}$ profile plotted against the cumulative mass resulted in a linear profile (Fig. 6).

**3.4 Mass Accumulation Rates and Carbon Accumulation Rates**

The "basal age", defined as the age at the base of the deepest peat layer, ranged from 13 to 140 years (Table 2). At
all sites, the basal ages are consistently older in low marsh cores (106 ± 17 yrs) when compared to high marsh cores (33 ± 9 yrs). As a result, the MARs were an order of magnitude higher in high marsh zones compared to low ones, with average MAR of 3978 ± 1231 and 399 ± 67 g m$^{-2}$ yr$^{-1}$ in the low marsh sediment, respectively, averaging 2188 ± 885 g m$^{-2}$ yr$^{-1}$ (range 262-6977 g m$^{-2}$ yr$^{-1}$) for the whole marsh (Table 2).

The average base of peat CAR for the dated cores was 184 ± 50 g C m$^{-2}$ yr$^{-1}$ (range 47 to 386 g C m$^{-2}$ yr$^{-1}$, Table 2). Similar
value was obtained by using the 30-year average CARs, with an average of 178 ± 46 g C m$^{-2}$ yr$^{-1}$ (range 59 to 394 g C m$^{-2}$ yr$^{-1}$) (Table 2; Fig. 5b). In the base of peat estimates, CARs in the four high marsh cores (average 303 ± 45 g C m$^{-2}$ yr$^{-1}$) were significantly higher than the low marsh core average (63 ± 7 g C m$^{-2}$ yr$^{-1}$; p < 0.05). This difference was also apparent when comparing the 30-year average CARs (280 ± 52 g C m$^{-2}$ yr$^{-1}$ and 76 ± 7 g C m$^{-2}$ yr$^{-1}$ for high and low marsh, respectively, p < 0.05).

When we use the high and low marsh areas with the average of the CARs calculated to the base of the peat in each core, we find that the high marsh accumulates 83 ± 12 Mg C yr$^{-1}$, and the low marsh accumulates 13 ± 1.4 Mg C yr$^{-1}$ (Fig. 5c, Table 2). Combining these estimates produces an annual marsh-wide C accumulation of 96 ± 12 Mg C yr$^{-1}$, 86% of which is accumulating annually in the high marsh zones. Similar results are obtained when using the 30-yrs CARs, with high marshes accumulating 85 % of marsh-wide annual C accumulation (77 ± 14 Mg C yr$^{-1}$) (Table 2).


**Table 2. Core and Marsh Wide Carbon Accumulation Rates**

| Site | Area (ha) | Carbon Accumulation Rate (CAR, g C m$^{-2}$ y$^{-1}$) | | Marsh wide Carbon Accumulation[b] (Mg C y$^{-1}$), Ave ± SE | | Sedimentation Characteristics, Ave ± SE[c] | | |
|---|---|---|---|---|---|---|---|---|
| | | 30-yr | Peat Base | 30-yr | Peat Base | Basal Age (years before 2016)[d] | LSR (cm yr$^{-1}$) | MAR (g cm-$^2$ yr$^{-1}$) |
| *High Marsh* | | | | | | | | |
| CBE 1-1 | 1.42 | 342 ± 25 | 293 ± 21 | 4.9 ± 0.4 | 4.2 ± 0.3 | 54.1 ± 3.9 | 0.97 ± 0.56 | 0.156 ± 0.011 |
| CRF 2-1 | 17.31 | 394 ± 41 | 352 ± 37 | 68.2 ± 7.1 | 60.9 ± 6.4 | 26.7 ± 2.8 | 1.08 ± 0.55 | 0.242 ± 0.025 |
| GBK 1-2 | 6.9 | 189 ± 9 | 386 ± 18 | 13.0 ± 0.6 | 26.6 ± 1.2 | 13.3 ± 0.6 | 1.80 ± 1.40 | 0.697 ± 0.033 |
| TMF 2-1 | 0.82 | 193 ± 8 | 182 ± 8 | 1.6 ± 0.1 | 1.5 ± 0.1 | 38.5 ±1.7 | 1.00 ± 0.80 | 0.496 ± 0.022 |
| **Ave ± SE** | 27.4[a] | 279.5 ± 52 | 303 ±45 | 76.6 ± 14.3 | 83 ±12 | 33.2 ± 8.7 | 1.21 ±0.20 | 0.40 ± 0.12 |
| *Low Marsh* | | | | | | | | |
| CBE 1-5 | 2.57 | 69 ± 21 | 65 ± 20 | 1.8 ± 0.5 | 1.9 ± 0.6 | 131.4 ± 39.4 | 0.175± 0.060 | 0.0346 ± 0.0104 |
| CRF 3-2 | 10.11 | 87 ± 20 | 62 ±14 | 8.8 ± 2.0 | 6.3 ± 1.4 | 139.5 ± 32.5 | 0.167 ± 0.075 | 0.0262 ± 0.0061 |
| GBK 1-4 | 4.79 | 59 ± 12 | 47 ± 10 | 2.8 ± 0.6 | 2.3 ± 0.5 | 87.8 ± 18.5 | 0.133 ± 0.19 | 0.0408 ± 0.086 |
| TMF 1-2 | 0.69 | 88 ± 10 | 80 ± 9 | 0.6 ± 0.1 | 0.6 ± 0.1 | 66.7 ± 7.6 | 0.36 ± 0.27 | 0.0579 ± 0.0066 |
| **Ave ± SE** | 19.5[a] | 76 ± 8.2 | 63.5 ± 7 | 14.8 ± 1.4 | 13 ± 1.4 | 106.3 ± 17.4 | 0.21 ± 0.051 | 0.040 ± 0.0067 |
| *Total Marsh* | | | | | | | | |
| **Ave ± SE** | 46.9[a] | 178 ± 46 | 184 ± 50 | 83 ± 21[e] | 86 ± 23[e] | 69.7 ± 16.5 | 0.711 ± 0.212 | 0.219 ± 0.089 |

[a]Marsh areas represent the sums for the high, low, and total marsh.
[b]Marsh-wide Carbon Accumulation rates are estimated as the average (high, low, total) CAR multiplied by the marsh area (first column) for all seven marshes.
[c] Associated DBDs, %C, and SCD values are in Table A1.
[d]Basal Age reflects age determined in deepest sediment layer containing peat overlying sand, clay, or gravel.
[e]The total marsh-wide carbon accumulation values stated here are based on using the average total marsh CAR multiplied by the total marsh area. The values stated in the text are slightly higher because they are based on adding the low marsh and high marsh annual carbon accumulation values together.



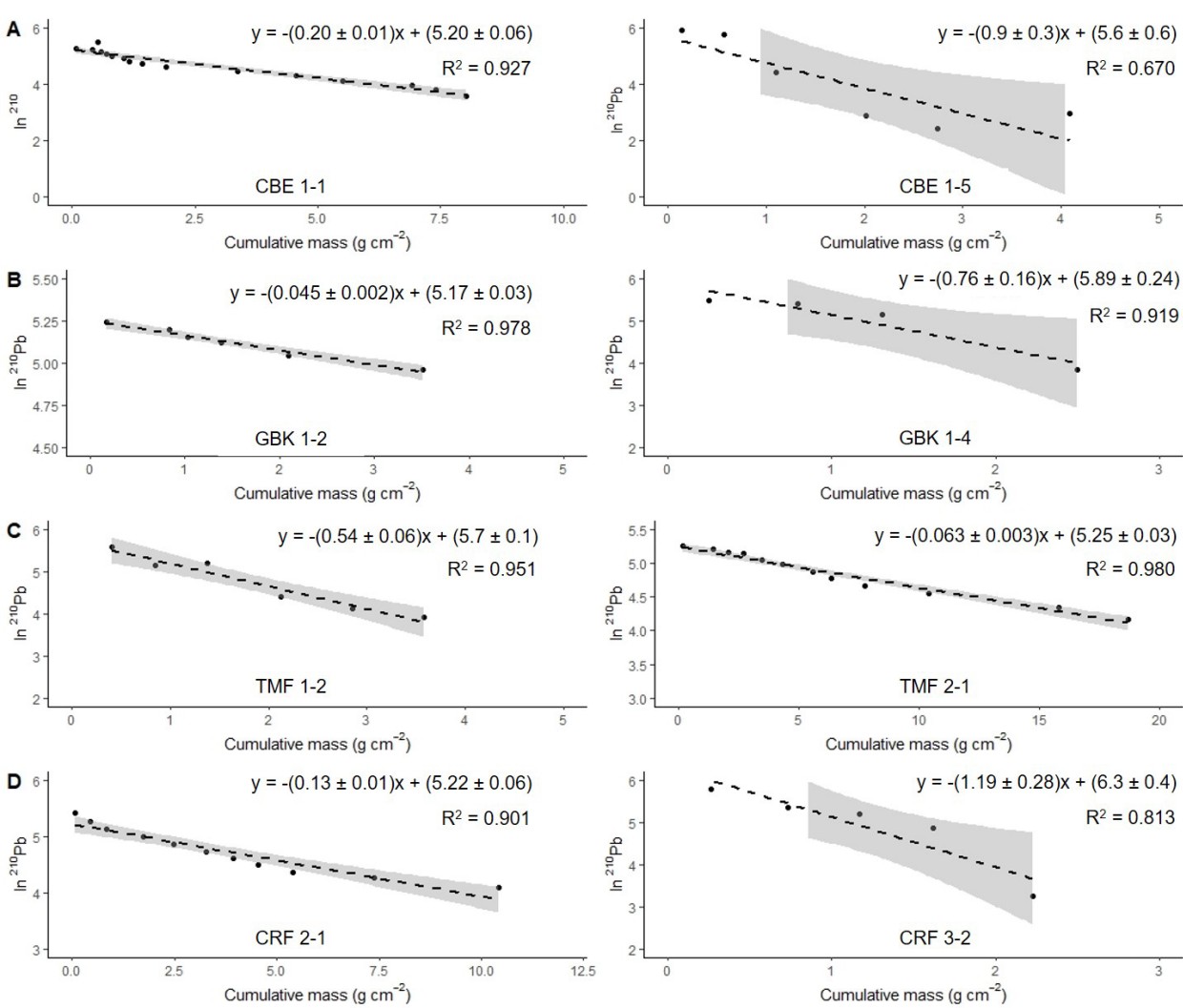

**Figure 6. Natural log of Excess $^{210}$Pb activity versus cumulative mass from the surface. The equation in each graph shows the results from the regression analyses used for the calculation of the mean accumulation rate (g cm$^{-2}$ y$^{-1}$) using the CF:CS model. Grey shaded area represents 95% confidence intervals.**

## 4 Discussion

### 4.1 Carbon Stocks

We calculate that the total C storage from salt marshes in Clayoquot Sound is 4189 ± 303 Mg C, with only 31% accumulated in the low marsh (see Sect. 3.2). Similar, small contributions from the low marsh zone have been found in other estuarine systems (Connor et al. 2001; Adams et al. 2012). In nearby Boundary Bay, BC, Gailis et al. (2021) also found that the low marsh contributed a disproportionately small fraction (15%) to the total marsh-wide C stocks relative to their area. As with Boundary Bay, we found that the small low marsh contribution was driven in part by the smaller proportion of low marsh area

(42% of total) in Clayoquot Sound. Gailis et al. (2021) also attributed lower C stocks in low marsh sediments to vegetation type and associated rooting depths (which are shallower in low marsh *Salicornia* spp). Similar differences in vegetation are observed in the Clayoquot Sound marshes. Furthermore, when normalized to a constant depth horizon (20 cm), the C stocks in Clayoquot Sound low marsh zones are 28% lower than those in the high marsh zones (Fig. 4). However, the main drivers of the smaller C stocks are significantly shallower depths of the base of the peat layer in low marsh relative to the high marsh cores in Clayoquot Sound (Fig. 3a). High marsh peat layers are on average 71% thicker than those found in the low marsh zones, suggesting that low marsh sediments are simply too shallow to accumulate as much C as the high marsh zones.

The C stocks in Clayoquot Sound marshes are less than one-third of the globally averaged estimate of 250 Mg C ha[-1] for salt marsh C stocks in the upper 50 cm (Chmura et al. 2003; Pendleton et al. 2012). This is true whether we consider the base of peat, 20-cm, or 30-yr estimates (Table 1). One possible contributor to these discrepancies could be the use of different formulas relating %C to LOI. Chmura et al. (2003) utilized the formula of Craft et al. (1991) to convert LOI measurements to %C. In contrast, we used an empirical relationship based on measurements collected from Clayoquot Sound marsh samples. Comparison of these two regression equations suggest that estimates of %C are very similar for values of %C equal to or less than 30%, but the %C estimates diverge for percentages above 30% (Fig. A3), with calculated differences in %C exceeding 20% at %LOIs above 80%. However, we note that the interquartile (Q1:Q3) range of our %C values fall between 4.4 and 28.8%, suggesting that most of our samples have %C values of less than 30%, where the two equations produce similar results. To test the potential impact of the different equations, we compared C stocks (estimated to peat base) that were calculated using the two different %C-LOI relationships for the eight cores that were [210]Pb dated (Fig. A4). Using the Craft et al. (1991) regression inflates our C stock values by about 30%, but this increase is not sufficiently large to account for the full difference between our C stocks and the global values in Chmura et al. (2003), which are three times greater than our estimated C stocks. The C stocks in these seven Clayoquot Sound marshes are consistent with regional values found in Boundary Bay, BC (83 ± 30 and 39± 24 Mg C ha[-1] for high and low marsh, respectively, Gailis et al. 2021), in Stillaguamish, Washington (60 ± 4 Mg C ha[-1]; Poppe and Rybczyk, 2019), and Snohomish estuary, Washington (72 Mg C ha[-1] and 54 Mg C ha[-1] for the top 30 cm of two marshes, Crooks et al., 2014). However, these C stocks are an order of magnitude lower than estimates of total soil C mass in high and low marshes along the Pacific coast of the USA (543 ± 47 and 411 ± 70 Mg C ha[-1], respectively (Kauffman et al. 2020)).

The lower overall C stocks in these Pacific marshes may in part be attributable to a lower overall C content: the average soil C density of the seven Clayoquot Sound marshes is 0.027 ± 0.002 g C cm[-3], which is lower than regional estimates for the Oregon Coast (0.034 ± 0.011 g C cm[-3]; Peck et al. 2020) and the global mean of 0.039 ± 0.003 g C cm[-3] (Chmura et al. 2003). However, low overall C content is not likely to be the main cause of the lower C stocks overall because these SCDs are comparable to the means for the US (0.028 ± 0.013 g C cm[-3]; Holmquist et al. 2018) and Pacific Northwest marshes (0.023 ± 0.002 g C cm[-3]) in which C stocks are an order of magnitude higher than Clayoquot Sound (Kauffman et al. 2020).

A more likely cause is simply the shallow thickness of the Clayoquot Sound marshes, with the depth to the base of the deepest peat layer averaging 23 ± 2 cm (17 cm for low marsh and 29 for high marsh). Similarly shallow depths of refusal (penetrating

into sand or clay layers) have been found in seagrass beds of Clayoquot Sound (Postlethwaite et al. 2018) as well as in Boundary Bay marsh (Gailis et al., 2021). In contrast, a sampling of seagrass and salt marsh systems along the US Pacific Coasts suggest that these ecosystems have substantially thicker soil C depths, with > 85% having mean soil profile depths greater than 2 m (Kauffman et al. 2020). Because data represented in many syntheses come from regions with marshes that have depths of one meter or greater (e.g. Macreadie et al. 2017), extrapolating the results to the total marsh area likely overestimates total C storage. Thus, we recommend that ongoing work to assess blue C contents globally begin to consider shallower depth horizons and/or age horizons as a means for creating a global comparison.

**4.2 Carbon Accumulation Rates, Comparisons of Marsh Zones and Pacific Coast Marshes**

While marsh C stocks in Clayoquot Sound are substantially lower than the global and Pacific coast averages, the average CARs are actually higher than those found in other parts of the Pacific Coast, especially when we separate measurements by marsh zone (Fig. 7). When all $^{210}$Pb-dated sites are combined, we obtain an average CAR for the NE Pacific region of $112 \pm 12$ g C m$^{-2}$ yr$^{-1}$ (mean $\pm$ SE, n=18). While this value is comparable to the global average CAR of $91 \pm 2.3$ g C m$^{-2}$ yr$^{-1}$ (Kennedy et al., 2013), both are substantially lower than the regionally averaged CAR of $184 \pm 50$ g C m$^{-2}$ yr$^{-1}$ for Clayoquot Sound (Fig. 7c). When we consider only high marsh cores, the average CAR for Clayoquot Sound further increases to $303 \pm 45$ g C m$^{-2}$ yr$^{-1}$, substantially larger than the NE Pacific high-marsh average of $145 \pm 26$ g C m$^{-2}$ yr$^{-1}$ (Fig. 7a). In fact, the NE Pacific high marsh average is driven substantially by the inclusion of Clayoquot Sound marshes and drops to $96 \pm 12$ g C m$^{-2}$ yr$^{-1}$ when Clayoquot Sound marshes are excluded. This comparison suggests that while the overall C storage in Clayoquot Sound marshes is low, the current rates of accumulation are much higher than the regional average, particularly in the high marsh zones. Thus, assuming these rates of C accumulation continue, high marsh zones of Clayoquot Sound have the potential to play a more important role as C sinks in the future.

One effort has been made to assemble a global compilation and synthesis of CARs within salt marsh ecosystems (Ouyang and Lee, 2014). Here we compare our results with this compilation with the caveats that (a) Ouyang and Lee (2014) relied heavily on $^{137}$Cs dating or marker horizon methods for their estimates, and (b) appears to have some instances of double-counting of sites (e.g., averages of 3 sites were included as a 4th site) and minor issues with attribution of some geographic locations. That said, we note that our new average estimate for NE Pacific CARs of $112 \pm 12$ g C m$^{-2}$ yr$^{-1}$ is substantially lower than estimates compiled by Ouyang and Lee (2014), whether considering the global average (245 g C m$^{-2}$ yr$^{-1}$), the NE Pacific average (167 g C m$^{-2}$ yr$^{-1}$, n = 6), the NW Atlantic region (172 g C m$^{-2}$ yr$^{-1}$; n=64; 35.0-47.4 °N), or the subset of cores from Atlantic Canada (188 g C m$^{-2}$ yr$^{-1}$; n=40; 43.6-47.4 °N). For the NE Pacific region, this compilation predominantly included sites where $^{137}$Cs-dating was used to estimate CARs (Ouyang and Lee, 2014). The remaining NE Pacific CAR studies (n=2) used marker horizon methods, which, when included, increase the average CAR to $174 \pm 128$ g C m$^{-2}$ yr$^{-1}$. Marker horizon and $^{137}$Cs dating methods are shown to produce CARs higher than those calculated from $^{210}$Pb (Chmura et al. 2003; Callaway et al. 2012). This is due to gradual decomposition in the sediment column, averaging CAR over a shorter time period, and to a shallower depth, oversampling high C soils vis-à-vis lower C soils at depth. As most of the CAR data points in the NW Atlantic region in

Ouyang and Lee's 2014 summary were also calculated using [137]Cs dating or marker horizon methods (n=62), these also represent roughly equivalent values to Clayoquot Sound if the observed bias is considered.

Differences between Clayoquot Sound and regionally averaged CARs are not apparent for low marsh zone, for which the Clayoquot Sound average of $63 \pm 7$ g C m$^{-2}$ yr$^{-1}$ is statistically similar to the regional estimate of $74 \pm 6$ g C m$^{-2}$ yr$^{-1}$ (Fig. 7b). Within the context of Clayoquot Sound, the low marsh zone exhibits substantially lower C stocks and accumulation rates than the high marsh zones. Basal ages (age of the deepest peat layer overlying sand) are substantially older in low marsh sediments when compared to the high marsh, resulting in significantly lower sedimentation rates and CARs.

Our observation of significantly lower CARs in the low marsh compared to the high marsh could result from several factors. As mentioned previously, the vegetation in the low marsh zone is dominated by *Salicornia* spp, whose lower productivity and shallow root systems could be producing and trapping less C (Kelleway et al. 2017; Gailis et al. 2021). A second, potential contributor to slow accumulation rates of C could be the relatively low sediment discharge from terrestrial sources such as the Kennedy River (Nuwer and Kiel, 2005). Previous work in Clayoquot Sound attributed low carbon accumulation rates in eelgrass, in part, to relatively clear waters (secchi disc measurements) and C/N ratios that reflect marine rather than terrestrial sediment contributions (Postlethwaite et al. 2018).

A third important feature of the Clayoquot Sound marshes is that the region is experiencing vertical land motion that counteracts the effects of sea level rise (Mazzotti et al. 2007, 2008; Montillet et al. 2018). Previous work suggests that, in many cases, marshes are responding to rising sea levels by migrating inland (Kirwan et al. 2016). In contrast, in our study region the nearby town of Tofino experiences uplift of 2.86 mm y$^{-1}$, such that relative sea level is dropping by 1.08-1.15 mm y$^{-1}$ (Montillet et al. 2018). In fact, sea level change along the Pacific Coast of Canada where these marshes are developing averages -0.76 $\pm$1.32 mm/y (James et al. 2014), suggesting that, while some areas along the Pacific Canadian coast are exposed to sea level rise, average conditions for regional salt marshes are likely comparable to what is being experienced in Clayoquot Sound. Recent work along the Oregon coast suggests that relative sea level rise is a dominant control on vertical accretion rates in tidal marsh systems, and areas experiencing negative sea level rise (or land uplift) have the lowest rates of C accumulation (Peck et al. 2020). The negative sea level rise and relatively low C accumulation rates, particularly in the low marsh settings of Clayoquot Sound and Pacific Rim National Park, are consistent with this result.

A final discriminating characteristic between the high and low marsh cores in Clayoquot Sound are the [210]Pb inventories (Fig. A5), which suggest that the high and low marsh zones are subjected to different erosional processes. Given that these cores all come from the same area one would assume that all the cores should have similar inputs of [210]Pb (Olid et al. 2010; Arias-Ortiz et al. 2018). However, a comparison of the [210]Pb inventories shows that the high marshes accumulate much more [210]Pb than the low marshes. This difference, together with the lower concentrations of the [210]Pb in the upper layers of the low marsh cores, could result from focusing / soil formation causing greater [210]Pb accumulation in the high marshes and/or reduced accumulation in the low marshes due to erosion and winnowing processes. One possible explanation here is that the extensive, shallow flats in the low marsh regions of Clayoquot Sound are subjected to more intensive tidal erosion and therefore removal of soil organic

material, which results in lower accumulation rates relative to the high marsh zones. Further studies are needed to estimate the significance of these drivers within the region.

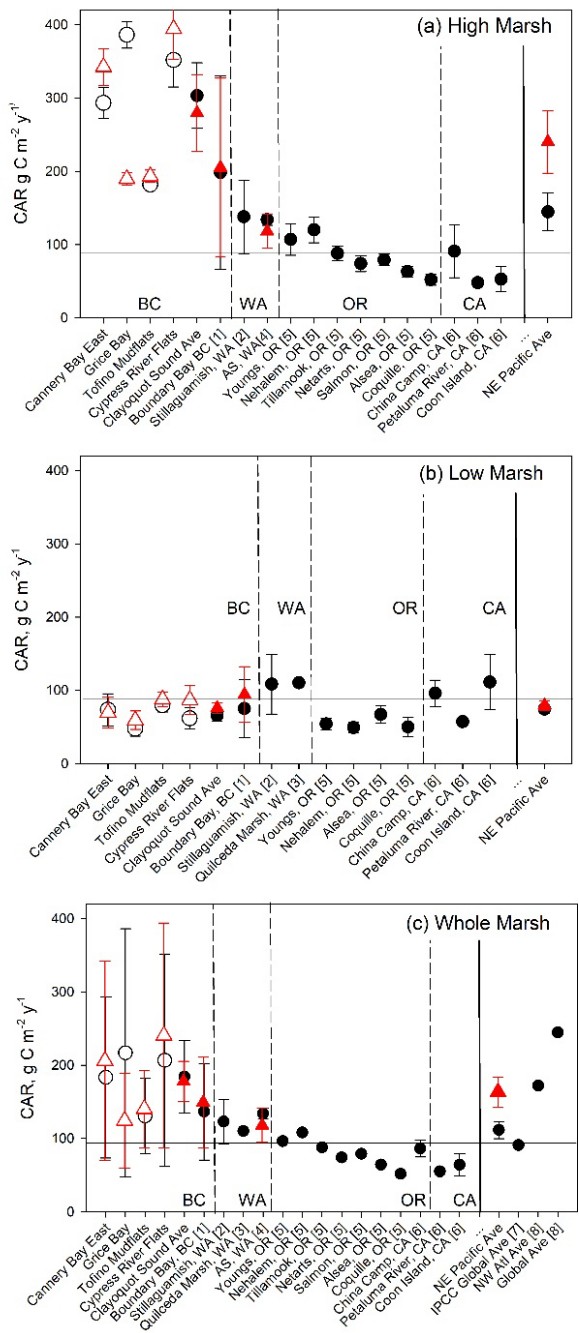

**Figure 7. Comparison of Clayoquot Sound CARs with other salt marshes with ²¹⁰Pb dating, for whole core (black circles) and 30-year (red triangle) estimates. Individual Clayoquot Sound marshes are indicated with open symbols followed by a regional mean (± SE). Additional site data taken from: [1] Boundary Bay, BC, Canada (Gailis et al., 2021); [2] Stillaguamish, WA, USA (Poppe and**

Rybczyk, 2019); [3] Snohomish Estuary, WA, USA (Crooks et al., 2014); [4] Animal Slough (AS) in southern Puget Sound, WA, USA (Note that red triangle is 50-yr average; Drexler et al., 2019); [5] coastal OR, USA (Peck et al., 2020); [6] San Fransciso Bay, CA, USA (Callaway et al., 2012). Global and other regional averages are taken from [7] Kennedy et al. (2013) and [8] Ouyang and Lee (2014); only a fraction of data (<10%) in [8] were dated using [210]Pb. Horizontal line represents Kennedy et al. (2013) global average.

### .3. Comparison of Whole-Core and 30-year CARs

Estimating CARs to a consistent age horizon of 30 years allowed us to compare C accumulation between marshes and marsh zones, but it also reveals interesting information about differences in CAR depending on the approach used to estimate these values. Previous comparisons in nearby Boundary Bay marsh showed that CARs were 20% higher during the last 30 years when compared to the whole core estimates (Gailis et al. 2021). Because %C decreases with depth in all 34 cores from Clayoquot Sound, we might anticipate that CARs for the most recent years will always result in a higher CAR than a measurement of C stock to the depth of minimum [210]Pb detectable activity. This can be partially attributed to remineralization of soil C over time, which would decrease overall %C in the deeper portions of the sediment cores (Callaway et al. 2012; Johannessen and MacDonald 2016). In Clayoquot Sound, however, the CARs estimated for the last 30 years were not substantially different from our estimates made to the base of the peat layer (Table 2), particularly in the high marsh zone where the average age for the base of the peat layer was 33 ± 9 years. Slightly larger differences were seen in the slowly-accumulating low marsh zones, where the 30-year CAR was about 15% higher.  Overall, the young ages of these marshes result in small differences between the two estimates.

The examination of 30-year CARs as well as CARs in relatively young marshes could have interesting implications for future blue C assessments. Many marsh restoration projects in which C is being measured are currently on time scales of 10 to 30 years past the initiation of restoration (e.g. Abbott et al. 2019; Drexler et al. 2019). Vegetational succession and C storage in a restored tidal marsh can take over 10 years (Havens et al., 2002; Craft et al. 2003), and even a 30-year monitoring period would capture a period of gradually improving C sequestration potential in restored tidal marsh soils (e.g. Howe et al. 2009). However, the inflation of CARs in younger sediments in the Clayoquot Sound and Boundary Bay sediments could provide a caution that CARs measured from relatively young marker horizons are at risk of overestimating total C in sediments, especially if these CARs are used to project C sequestration over longer time periods. Several recent studies have shown substantially higher CARs in restored vs natural marshes after approximately 10 years (e.g., Poppe and Rybczyk, 2019; Drexler et al. 2019). The causes of these inflated CARs can be very site-specific, depending on the types of colonizing vegetation (Kelleway et al. 2017) and the relative contributions of allochthonous mineral materials (Drexler et al., 2019). In the case of Clayoquot Sound, it is possible that these high CARs will be sustained, making this marsh extremely important as a regional C sink.  However, we also cannot rule out the possibility that longer-term processes, such as remineralization and sediment redistribution, could reduce the actual amount of C stored as the marshes age, such that the accumulation rates become more comparable to those found on other parts of the Pacific coast (Fig. 7).

## 4.4 Comparison with carbon storage in boreal forest ecosystems

The relevance of blue C storage potential to climate change mitigation depends on the scale over which it is considered. In the literature, the carbon accumulation potential of blue carbon is often compared with terrestrial ecosystems, and it is often stated that, per unit area, blue C ecosystems accumulate carbon at higher estimated rates than terrestrial forest soils (McLeod et al. 2011).

A first-order comparison for Canadian ecosystems shows the same pattern of higher carbon uptake rates per unit area in tidal wetlands when compared with measurements of net ecosystem productivity of terrestrial forests. Forested areas in Canada have been estimated to take up carbon at rates ranging from 35 g C m$^{-2}$ yr$^{-1}$ (Canada-wide estimate, Stinson et al., 2011) to 63 g C m$^{-2}$ yr$^{-1}$ (British Columbia, Peng et al., 2014). The 30-year CAR for salt marshes in Clayoquot Sound are approximately 2-7 times higher, averaging $178 \pm 46$ g C m$^{-2}$ yr$^{-1}$. Note that we considered forests' net ecosystem productivity, which is higher than C uptake rates of $4.6 \pm 2.1$ g C m$^{-2}$ yr$^{-1}$ for boreal forest soils cited by McLeod et al. (2011).

Consideration of the areal extent of each of these ecosystems is extremely important for contextualizing blue C uptake among Canadian ecosystems. Estimates of total salt marsh area in Canada range from 44,000 ha (Bridgham et al., 2006) to 111,274 ha (Mcowen et al., 2017). In contrast, the boreal forest ecosystem is one of Canada's largest terrestrial biomes and encompasses approximately 270 million ha (Kurz et al. 2013). Our Clayoquot Sound data represent only a small area of a single region of the west coast, but if we assume our CAR estimate of $184 \pm 50$ g C m$^{-2}$ yr$^{-1}$ from Clayoquot Sound approximates the average for all tidal salt marshes in Canada, Canada's marshes would accumulate somewhere between 81,000 and 205,000 Mg C yr$^{-1}$. This annual carbon accumulation from Canadian salt marshes is between 0.4 and 2.9 % of the carbon estimated to accumulate annually by Canadian boreal forests (6,750,000-18,090,000 Mg C yr$^{-1}$).

Our back-of-the-envelope comparison is not intended to provide a formal estimate of carbon sequestration in Canadian salt marshes but instead highlights several important assumptions and uncertainties that are important to large-scale assessments of blue carbon sequestration. The first is that regional-to-global estimates of blue C accumulation rates depend on regionally specific measurements of salt marsh C stocks and accumulation rates, and extrapolation of blue C accumulation rates based on globally derived averages (or averages from regions other than the one in question) can lead to large uncertainties in blue C accumulation rates. Our calculation above assumes that the CAR estimates from Clayoquot Sound are representative of the pan-Canadian average for CARs in salt marshes. Using the published value for Atlantic Canada (188 g C m$^{-2}$ yr$^{-1}$, Ouyang and Lee, 2014) would not substantially change our calculation. However, if the average pan-Canadian CAR is closer to the global estimate of $91 \pm 2.3$ g C m$^{-2}$ yr$^{-1}$ (Kennedy et al., 2013), our estimate of annual carbon accumulation would halve. Thus, a pan-Canadian estimate of CAR (and its associated uncertainty) would be useful for improving the estimated carbon sequestration rates in Canadian salt marshes.

Second, our comparison with C sequestration in Canadian forests is based on net ecosystem productivity estimates from 2011 and 2013. With climate change, the increased prevalence of natural disturbances such as forest fires and insect outbreaks is projected to result in episodic losses of carbon from the above-ground and soil carbon pools (e.g. Stinson et al., 2011, Walker

et al., 2019). Between 1990-2008, 24 ± 19 Tg C yr-1 was removed from the forest carbon pool by fire (Kurz et al., 2013), with British Columbia forests becoming a sustained source of $CO_2$ since 2003 (e.g. Giles-Hansen and Wei, 2022). The degree to which climate change will impact forest carbon sequestration in future will depend on several factors, including increases in disturbance frequency and intensity (Walker et al. 2019), subsequent recovery and shifts in forest species dominance (e.g. Mack et al. 2021), as well as management practices (Price et al. 2013). At the same time, as sea level rise is currently a minimal

threat to the Canadian Pacific salt marshes, they are likely to continue to function as efficient C sinks despite global warming. Finally, in addition to the regionally-specific nature of processes controlling blue C accumulation, we note that estimation of the areal extent of these marshes varies by an order of magnitude within Canada alone, and determinations of the relative proportions and high-to-low marsh settings have not been quantified. We note, for example, that while large-scale databases suggest that the areal extent of salt marsh in Boundary Bay, BC, is 1,207 ha (CEC, 2016a), our field research in Boundary Bay

has determined that the actual marsh area is only one-quarter the size at 275 ha (Gailis *et al.*, 2021). This suggests that estimation of blue C potential in policy-specific contexts requires a more detailed determination of the areal extent of these marshes, especially when considering provincial-to-national scales. We further suggest that the small area over which blue C accumulates likely influences the scale over which it is best considered for natural land carbon sequestration and accounting. Blue C accumulation occurs in only 0.016-0.1 % the land area of boreal forests nationwide. On a global, national, and even

provincial scale, total blue C sequestration in salt marshes is relatively small compared to the large areal expanses of forest. However, the very high, per-unit area sequestration of carbon in salt marsh ecosystems can increase its importance at local-to-regional scales and should be considered as an important co-benefit to restoration activities and conservation for other purposes.

### 4.5 Future Work

While this study has provided a useful first step in quantifying blue C stocks, storage, and accumulation rates, we note the following considerations in estimating the potential of blue C within this region. First, our study has estimated blue C in 47 ha of high and low marsh areas of Clayoquot Sound. We expect that these mesotidal estuarine marshes, often constrained in size by surrounding topography, are typical of the marshes found on the Pacific coast of British Columbia, the total estimated area of which is 60 square kilometres, or 6000 ha (Ryder et al. 2007). However, given variability in riverine discharge and tectonic

settings along the coast, we anticipate that additional measurements (additional marshes, with differing riverine and sedimentary inputs, and variable uplift setting) will be necessary to understand the variability in carbon accumulation along the Pacific Coast of Canada. Second, our study has focused on estimation of carbon stocks and accumulation rates within these marshes, and full estimation of blue C potential will ultimately require estimation of greenhouse gas fluxes of carbon dioxide, methane, and nitrous oxide within these marshes (e.g. Chmura et al. 2011). The brackish nature of some of these marshes could

mean that some emit substantial contributions of methane, which may counter their effectiveness as C sinks (e.g. Poffenbarger et al. 2011; Abdul-Aziz et al. 2018; Huang et al. 2019; Huertas et al. 2019; Li et al. 2021; Wei et al. 2020). Finally, this study has revealed the importance of classifying high vs low marsh zones to best quantify carbon stocks and accumulation rates, as

the different vegetation growth and sedimentary processes operating in these zones can critically influence CARs, and therefore estimates of C sequestration. Future work thus needs to involve better quantification of the areal extent and classification of

salt marshes, both within Clayoquot Sound, along the Pacific coast, and in Canada as a whole.

**5 Conclusion**

Our work helps address the data gap for North American blue C by producing CAR estimates using $^{210}$Pb-derived mass accumulation rates of salt marsh sediments, showing CAR comparable to other salt marsh locations in the Northeast Pacific coast. Importantly, this work demonstrates that the marshes of Clayoquot Sound along the Pacific Coast of Canada exhibit

substantially smaller C stocks when compared with other regional salt marshes to the south, along the Pacific Coast of the United States. We attribute these lower stocks to substantially shallower (< 50 cm) depths of marsh initiation in all high and low marsh settings. At the same time, while carbon stocks are lower than regional and global averages, the rates of carbon accumulation in these marshes are comparable and even higher than regional estimates. We show that both 30-year CARs and CARs estimated to the peat base are substantially greater than regional averages generated using only $^{210}$Pb-derived mass

accumulation rates. However, significant variability exists between marshes, and different radioisotope dating methods in CAR measurement create uncertainty. Our results support existing findings that blue C soils accumulate carbon at high rates, limited only by the spatial extent of such ecosystems. These results provide a natural analogue for monitoring carbon accretion over timescales relevant for restoration activities.

# Appendix A Supplemental Tables and Figures

 **Table A1. Core Locations, Soil Carbon Characteristics, and Carbon Stocks**

| Core ID | Lat. (49.- ° N) | Long. (125.- ° W) | Marsh Zone | Compacted (Complete) Core Length (cm) | *Average Uncorrected DBD ± SE to Peat Base (g cm⁻³) | Average %C ± SE To Peat Base | *Average Corrected SCD ± SE to Peat Base (g C cm⁻³) | C Stock to Peat Base (Mg C ha⁻¹) | **C Stock to 20 cm (Mg C ha⁻¹) | 30-yr C Stock (Mg C ha⁻¹) |
|---|---|---|---|---|---|---|---|---|---|---|
| *Cannery Bay East* | | | | | | | | | | |
| CBE 1-1 | 0.14139 | 0.6662 | high | 47 (62) | 0.320 ± 0.037 | 23.4 ± 1.8 | 0.031 ± 0.0013 | 158.7 | 62.9 | 107.7 |
| CBE 1-2 | 0.14142 | 0.66629 | high | 38 (40) | 0.309 ± 0.030 | 17.0 ± 1.9 | 0.032 ± 0.0019 | 129.9 | 78.0 | |
| CBE 1-3 | 0.14147 | 0.66636 | high | 24 (24) | 0.148 ± 0.012 | 28.3 ± 1.6 | 0.038 ± 0.0008 | 90.3 | 79.2 | |
| CBE 1-4 | 0.14152 | 0.66639 | low | 14 (14) | 0.154 ± 0.010 | 23.5 ± 1.5 | 0.035 ± 0.0018 | 48.4 | - | |
| CBE 1-5 | 0.14155 | 0.66644 | low | 20 (30) | 0.226 ± 0.021 | 18.7 ± 1.4 | 0.029 ± 0.004 | 85.7 | 43.6 | 21.9 |
| CBE 2-2 | 0.1414 | 0.66618 | high | 30 (30) | 0.16 ± 0.014 | 30.3 ± 1.0 | 0.045 ± 0.0020 | 133.8 | 86.7 | |
| CBE 2-3 | 0.14142 | 0.66614 | high | 40 (40) | 0.119 ± 0.006 | 28.4 ± 1.0 | 0.032 ± 0.0006 | 132.1 | 65.7 | |
| CBE 2-4 | 0.14144 | 0.66609 | low | 20 (30) | 0.179 ± 0.024 | 28.6 ± 2.0 | 0.033 ± 0.0036 | 39.8 | 62.5 | |
| CBE 2-5 | 0.14151 | 0.66602 | low | 30 (30) | 0.767 ± 0.056 | 1.9 ± 0.5 | 0.013 ± 0.0029 | 9.0 | 9.2 | |
| *Cannery Bay West* | | | | | | | | | | |
| CBW 1 | 0.14115 | 0.66983 | high | 16 (20) | 0.158 ± 0.017 | 20.7 ± 0.6 | 0.026 ± 0.0017 | 28.7 | 46.7 | |
| CBW 2 | 0.14115 | 0.66982 | low | 16 (23) | 0.266 ± 0.061 | 15.9 ± 3.3 | 0.025 ± 0.0029 | 17.8 | 23.5 | |
| CBW 3 | 0.14113 | 0.66962 | low | 7 (7) | 0.165 ± 0.004 | 32.3 ± 1.8 | 0.053 ± 0.0031 | 84.9 | - | |
| CBW 4 | 0.14112 | 0.66955 | high | 24 (30) | 0.157 ± 0.023 | 17.8 ± 1.7 | 0.019 ± 0.0020 | 53.6 | 47.4 | |
| *Cypress River Flats* | | | | | | | | | | |
| CRF 1-1 | 0.27905 | 0.90754 | high | 38 (46) | 0.168 ± 0.008 | 26.9 ± 1.1 | 0.036 ± 0.0013 | 163.2 | 81.7 | |
| CRF 1-2 | 0.27896 | 0.90758 | low | 16 (30) | 0.168 ± 0.009 | 28.7 ± 1.0 | 0.025 ± 0.0013 | 76.1 | 57.8 | |
| CRF 2-1 | 0.27935 | 0.90932 | high | 37 (51) | 0.207 ± 0.021 | 22.5 ± 1.5 | 0.028 ± 0.0014 | 118.7 | 57.5 | 120.3 |
| CRF 2-2 | 0.27916 | 0.90926 | low | 26 (38) | 0.205 ± 0.025 | 25.3 ± 1.9 | 0.028 ± 0.0009 | 104.8 | 66.7 | |
| CRF 3-1 | 0.2789 | 0.911 | high | 32 (40) | 0.225 ± 0.045 | 27.1 ± 2.3 | 0.032 ± 0.0016 | 110.6 | 105.9 | |

| Core ID | Lat. (49.-° N) | Long. (125.-° W) | Marsh Zone | Compacted (Complete) Core Length (cm) | *Average Uncorrected DBD ± SE to Peat Base (g cm$^{-3}$) | Average %C ± SE To Peat Base | *Average Corrected SCD ± SE to Peat Base (g C cm$^{-3}$) | C Stock to Peat Base (Mg C ha$^{-1}$) | **C Stock to 20 cm (Mg C ha$^{-1}$) | 30-yr C Stock (Mg C ha$^{-1}$) |
|---|---|---|---|---|---|---|---|---|---|---|
| CRF 3-2 | 0.27882 | 0.91087 | low | 23 (30) | 0.192 ± 0.023 | 27.4 ± 2.3 | 0.035 ± 0.0021 | 85.8 | 94.8 | 27.2 |
| *Grice Bay – Kootowis Creek* | | | | | | | | | | |
| GBK 1-1 | 0.08754 | 0.73238 | high | 16 (31)*** | 0.232 ± 0.032 | 18.3 ± 2.3 | 0.016 ± 0.0012 | 41.0 | 35.2 | |
| GBK 1-2 | 0.08756 | 0.73261 | high | 19 (32)*** | 0.440 ± 0.060 | 10.0 ± 1.8 | 0.015 ± 0.0016 | 51.2 | 40.0 | 59.2 |
| GBK 1-3 | 0.08763 | 0.73271 | high | 23 (26)*** | 0.257 ± 0.027 | 15.7 ± 1.4 | 0.032 ± 0.0028 | 60.9 | 64.5 | |
| GBK 1-4 | 0.08771 | 0.73283 | low | 13 (20)*** | 0.298 ± 0.017 | 12.0 ± 0.8 | 0.023 ± 0.0009 | 41.6 | 41.6 | 19.1 |
| *Kennedy Cove South* | | | | | | | | | | |
| KCS 1-1 | 0.13696 | 0.67082 | high | 24 (31) | 0.227 ± 0.041 | 21.4 ± 2.1 | 0.030 ± 0.0045 | 66.2 | 66.2 | |
| KCS 1-2 | 0.13707 | 0.67085 | low | 14 (21) | 0.248 ± 0.074 | 15.0 ± 6.9 | 0.018 ± 0.0012 | 27.4 | - | |
| KCS 1-3 | 0.13714 | 0.67093 | low | 16 (36) | 0.427 ± 0.097 | 11.1 ± 9.5 | 0.009 ± 0.0016 | 25.6 | 24.0 | |
| KCS 1-4 | 0.13719 | 0.67096 | low | 10 (20) | 0.296 ± 0.085 | 14.7 ± 2.2 | 0.014 ± 0.0012 | 28.9 | 28.9 | |
| KCS 1-5 | 0.1372 | 0.67107 | low | 10 (12) | 0.509 ± 0.125 | 7.6 ± 2.1 | 0.015 ± 0.0029 | 17.8 | 17.8 | |
| *Shipwreck Cove* | | | | | | | | | | |
| SWC 1-1 | 0.12995 | 0.69943 | high | 29 (29) | 0.248 ± 0.093 | 27.6 ± 3.4 | 0.033 ± 0.0030 | 42.3 | 43.9 | |
| SWC 2-1 | 0.13014 | 0.69908 | high | 18 (20) | 0.541 ± 0.124 | 21.0 ± 2.6 | 0.067 ± 0.0093 | 133.4 | 133.4 | |
| *Tofino Mud Flats* | | | | | | | | | | |
| TMF 1-1 | 0.13014 | 0.88689 | high | 27 (45) | 0.326 ± 0.068 | 18.6 ± 3.4 | 0.019 ± 0.0012 | 45.4 | 42.8 | |
| TMF 1-2 | 0.1302 | 0.88688 | low | 26 (30) | 0.241 ± 0.008 | 14.0 ± 0.8 | 0.038 ± 0.0012 | 53.0 | 57.5 | 28.9 |
| TMF 2-1 | 0.12989 | 0.88661 | high | 28 (31) | 0.751 ± 0.066 | 5.6 ± 1.5 | 0.022 ± 0.0041 | 62.4 | 61.0 | 61.5 |
| TMF 2-2 | 0.13017 | 0.88665 | low | 27 (30) | 0.231 ± 0.010 | 17.2 ± 0.7 | 0.035 ± 0.0010 | 65.9 | 69.5 | |

*Ave DBD values were not corrected for compaction (see Table A3 for corrected DBDs), but SCD values have been corrected for compaction.

** Cores that do not extend to 20 cm and do not penetrate a sand/gravel/clay layer at the base were excluded from the 20 cm calculation.

***GBK cores were extracted using a steel sledge corer that penetrated 15-30 cm into the sand and clay layers. The total core lengths were 46 (88), 60 (100), 59 (66) and 24 (37) cm for GBK 1-1, 1-2, 1-3, and 1-4, respectively. Depths provided here are just below the transition from peat to sand, two cm into the minimal %C layer.

**Table A2. Downcore distribution of $^{210}$Pb and $^{226}$Ra in eight cores used for $^{210}$Pb dating.**

| Upper Depth (cm) | Sediment Type[d] | Supported $^{210}$Pb ($^{226}$Ra, Bq/kg) | $^{210}$Pb total (Bq/kg) | +/- 1 SD (Bq/kg) | $^{210}$Pb$_{exs}$ (Bq/kq) | +/- 1 SD (Bq/kg) | Slope (cm$^2$/g) | Age (yr before 2016) | Age SD (yr) |
|---|---|---|---|---|---|---|---|---|---|
| **CBE 1-1 (HM)[a]** | | | | | | | | | |
| 0 | soil[d] | 2.03 | 196.2 | 7.2 | 194.2 | 7.2 | 0.20 ±0.014 | 0.51 | 0.04 |
| 4 | soil | 2.03 | 187.6 | 7.1 | 185.6 | 7.1 | | 2.70 | 0.19 |
| 5 | soil | 2.03 | 245.7 | 9.4 | 243.7 | 9.4 | | 3.27 | 0.24 |
| 6 | soil | 2.03 | 172.5 | 6.8 | 170.5 | 6.8 | | 3.79 | 0.27 |
| 7 | peat | 2.03 | 158.5 | 6.5 | 156.5 | 6.5 | | 4.43 | 0.32 |
| 8 | peat | 2.03 | 149.1 | 6.3 | 147.0 | 6.3 | | 5.14 | 0.37 |
| 10 | peat | 2.03 | 137.4 | 6.0 | 135.3 | 6.0 | | 6.68 | 0.48 |
| 11 | peat | 2.03 | 124.2 | 5.5 | 122.2 | 5.5 | | 7.38 | 0.53 |
| 13 | peat | 2.03 | 112.7 | 5.2 | 110.7 | 5.3 | | 9.12 | 0.66 |
| 16 | peat | 2.03 | 102.5 | 4.8 | 100.4 | 4.8 | | 12.33 | 0.89 |
| 21 | peat | 2.03 | 86.2 | 4.5 | 84.2 | 4.5 | | 21.57 | 1.55 |
| 24 | peat | 2.03 | 76.3 | 4.1 | 74.3 | 4.1 | | 29.21 | 2.10 |
| **25*** | **peat** | **2.03** | | | | | | **31.5** | **2.27** |
| 27 | peat | 2.03 | 62.0 | 3.8 | 59.9 | 3.8 | | 35.31 | 2.54 |
| 32 | peat | 2.03 | 53.5 | 3.6 | 51.4 | 3.6 | | 44.37 | 3.19 |
| 34 | peat | 2.03 | 46.6 | 3.2 | 44.6 | 3.2 | | 47.51 | 3.42 |
| 37 | peat | 2.03 | 36.7 | 2.8 | 34.7 | 2.8 | | 51.56 | 3.71 |
| **38**** | **peat** | **2.03** | | | | | | **54.1** | **3.90** |
| 42 | sand | 2.03 | 28.1 | 2.6 | 26.0 | 2.7 | | | |
| 46 | sand | 2.03 | 24.6 | 0.5 | 22.6 | 0.5 | | | |
| 14 ($^{226}$Ra) | | 1.90 | | 0.55 | | | | | |
| 44 ($^{226}$Ra) | | 2.16 | | 0.46 | | | | | |
| **CBE 1-5 (LM)[b]** | | | | | | | | | |
| 0 | soil+peat | 1.74 | 374.0 | 9.5 | 372.2 | 9.5 | 0.90 ±0.30 | 4.04 | 1.21 |
| 3 | soil+peat | 1.74 | 322.3 | 9.8 | 320.5 | 9.9 | | 16.46 | 4.93 |
| **6*** | **soil+peat** | **1.74** | **83.9** | **5.1** | **82.2** | **5.1** | | **31.77** | **9.52** |
| 10 | soil+peat | 1.74 | 19.2 | 2.0 | 17.4 | 2.1 | | 58.04 | 17.39 |
| 14 | soil+peat | 1.74 | 12.7 | 1.9 | 11.0 | 2.0 | | 79.12 | 23.70 |
| 18 | soil+peat | 1.74 | 20.7 | 2.2 | 19.0 | 2.2 | | 118.11 | 35.38 |

| Upper Depth (cm) | Sediment Type[d] | Supported $^{210}$Pb ($^{226}$Ra, Bq/kg) | $^{210}$Pb total (Bq/kg) | +/- 1 SD (Bq/kg) | $^{210}$Pb$_{exs}$ (Bq/kq) | +/- 1 SD (Bq/kg) | Slope (cm$^2$/g) | Age (yr before 2016) | Age SD (yr) |
|---|---|---|---|---|---|---|---|---|---|
| **19\*\*** | **soil+peat** | 1.74 | | | | | | **131.39** | **39.36** |
| 6 ($^{226}$Ra) | | 1.38 | | 0.94 | | | | | |
| 18 ($^{226}$Ra) | | 2.10 | | 0.46 | | | | | |
| **CRF 2-1 (HM)[a]** | | | | | | | | | |
| 0 | soil | 1.58 | 225.2 | 17.5 | 223.6 | 17.5 | 0.13±0.013 | 0.33 | 0.03 |
| 4 | soil | 1.58 | 194.2 | 16.4 | 192.6 | 16.4 | | 1.86 | 0.19 |
| 8 | soil | 1.58 | 169.3 | 15.4 | 167.7 | 15.4 | | 3.44 | 0.36 |
| 13 | peat | 1.58 | 149.3 | 14.5 | 147.8 | 14.5 | | 7.20 | 0.75 |
| 16 | peat | 1.58 | 129.9 | 13.7 | 128.4 | 13.8 | | 10.23 | 1.07 |
| 19 | peat | 1.58 | 114.1 | 12.9 | 112.5 | 13.0 | | 13.58 | 1.42 |
| 22 | peat | 1.58 | 101.1 | 12.0 | 99.5 | 12.0 | | 16.27 | 1.70 |
| 25 | peat | 1.58 | 91.8 | 11.4 | 90.2 | 11.5 | | 18.84 | 1.97 |
| 28 | peat | 1.58 | 80.0 | 11.1 | 78.4 | 11.1 | | 22.36 | 2.33 |
| **30\*\*** | **sand+peat** | | | | | | | **26.7** | **2.79** |
| **31\*** | **sand** | **1.58** | **72.1** | **10.2** | **70.5** | **10.3** | | **30.52** | **3.18** |
| 34 | sand | 1.58 | 61.6 | 9.7 | 60.1 | 9.8 | | | |
| **CRF 3-2 (LM)[b]** | | | | | | | | | |
| 1 | soil+peat | 1.58 | 326.6 | 14.3 | 325.0 | 14.4 | 1.19±0.28 | 9.93 | 2.31 |
| 4 | peat | 1.58 | 212.6 | 7.1 | 211.0 | 7.2 | | 27.89 | 6.50 |
| **5\*** | **peat** | | | | | | | **31.3** | **7.30** |
| 8 | peat | 1.58 | 182.8 | 7.5 | 181.2 | 7.6 | | 44.70 | 10.42 |
| 11 | peat | 1.58 | 130.5 | 5.6 | 129.0 | 5.7 | | 61.90 | 14.42 |
| 14 | peat | 1.58 | 27.0 | 2.5 | 25.5 | 2.8 | | 85.20 | 19.85 |
| **18\*\*** | **peat** | | | | | | | **139.5** | **32.49** |
| 20 | peat+sand | 1.58 | 2.2 | 1.2 | | | | | |
| 8 ($^{226}$Ra) | | 2.50 | | 1.17 | | | | | |
| 14 ($^{226}$Ra) | | 0.67 | | 0.67 | | | | | |
| **GBK 1-2 (HM)[a]** | | | | | | | | | |
| 0 | soil | 2.63 | 191.6 | 7.1 | 189.0 | 7.1 | 0.045±0.0021 | 0.24 | 0.01 |
| 4 | soil+peat | 2.63 | 183.0 | 7.0 | 180.3 | 7.0 | | 1.20 | 0.06 |

| Upper Depth (cm) | Sediment Type[d] | Supported $^{210}$Pb ($^{226}$Ra, Bq/kg) | $^{210}$Pb total (Bq/kg) | +/- 1 SD (Bq/kg) | $^{210}$Pb$_{exs}$ (Bq/kq) | +/- 1 SD (Bq/kg) | Slope (cm$^2$/g) | Age (yr before 2016) | Age SD (yr) |
|---|---|---|---|---|---|---|---|---|---|
| 5 | soil+peat | 2.63 | 175.1 | 6.8 | 172.5 | 6.8 | | 1.48 | 0.07 |
| 7 | soil+peat | 2.63 | 169.7 | 6.7 | 167.1 | 6.7 | | 1.99 | 0.09 |
| 9 | peat | 2.63 | 157.6 | 6.5 | 155.0 | 6.5 | | 3.00 | 0.14 |
| 12 | peat | 2.63 | 145.0 | 6.2 | 142.4 | 6.2 | | 5.03 | 0.24 |
| 16 | peat+sand | 2.63 | 131.8 | 5.9 | 129.2 | 5.9 | | 8.11 | 0.39 |
| 19 | peat+sand | 2.63 | 117.8 | 5.2 | 115.2 | 5.2 | | 11.90 | 0.57 |
| **20\*\*** | **peat+sand** | | | | | | | **13.26** | **0.63** |
| 22 | sand | 2.63 | 101.4 | 4.9 | 98.7 | 4.9 | | 16.08 | 0.76 |
| 27 | sand | 2.63 | 89.3 | 4.5 | 86.6 | 4.5 | | 23.50 | 1.12 |
| **33\*** | **sand+clay** | **2.63** | **75.9** | **3.9** | **73.3** | **3.9** | | **31.26** | **1.49** |
| 37 | clay | 2.63 | 57.4 | 0.5 | 54.8 | 0.5 | | 36.58 | 1.74 |

**GBK 1-4 (LM) [c]**

| Upper Depth (cm) | Sediment Type[d] | Supported $^{210}$Pb ($^{226}$Ra, Bq/kg) | $^{210}$Pb total (Bq/kg) | +/- 1 SD (Bq/kg) | $^{210}$Pb$_{exs}$ (Bq/kq) | +/- 1 SD (Bq/kg) | Slope (cm$^2$/g) | Age (yr before 2016) | Age SD (yr) |
|---|---|---|---|---|---|---|---|---|---|
| 0 | peat | 2.63 | 244.9 | 5.9 | 242.3 | 5.9 | $0.76 \pm 0.16$ | 6.13 | 1.29 |
| 2 | peat | 2.63 | 221.8 | 6.0 | 219.1 | 6.0 | | 19.37 | 4.08 |
| **4\*** | **peat** | **2.63** | **175.5** | **6.1** | **172.9** | **6.1** | | **32.11** | **6.76** |
| 8 | peat | 2.63 | 49.3 | 2.4 | 46.6 | 2.4 | | 61.29 | 12.90 |
| 10 | peat-sand | 2.63 | 7.0 | 0.8 | 4.3 | 0.8 | | 77.71 | 16.36 |
| **11\*\*** | **peat+sand** | | | | | | | **87.76** | **18.47** |
| 12 | Sand | 2.63 | 8.4 | 0.8 | 5.8 | 0.9 | | | |
| 14 | Sand | 2.63 | 6.8 | 0.7 | 4.2 | 0.7 | | | |
| 16 | Sand | 2.63 | 6.5 | 0.7 | 3.8 | 0.7 | | | |
| 18 | Sand | 2.63 | 5.4 | 0.7 | 2.8 | 0.7 | | | |
| 20 | Sand | 2.63 | 7.4 | 0.8 | 4.8 | 0.8 | | | |
| 22 | Sand | 2.63 | 116.0 | 4.3 | 113.4 | 4.3 | | | |
| 23 | Sand | 2.63 | 7.9 | 1.0 | 5.3 | 1.0 | | | |
| 3 ($^{226}$Ra) | | 2.58 | | 0.38 | | | | | |
| 20 ($^{226}$Ra) | | 2.68 | | 0.38 | | | | | |

**TMF 1-2 (LM) [b]**

| Upper Depth (cm) | Sediment Type[d] | Supported $^{210}$Pb ($^{226}$Ra, Bq/kg) | $^{210}$Pb total (Bq/kg) | +/- 1 SD (Bq/kg) | $^{210}$Pb$_{exs}$ (Bq/kq) | +/- 1 SD (Bq/kg) | Slope (cm$^2$/g) | Age (yr before 2016) | Age SD (yr) |
|---|---|---|---|---|---|---|---|---|---|
| 1 | soil+peat | 4.33 | 269.3 | 6.9 | 264.9 | 6.9 | $0.54 \pm 0.061$ | 6.91 | 0.79 |
| 3 | peat+mud | 4.33 | 173.9 | 6.4 | 169.5 | 6.4 | | 14.51 | 1.65 |
| 5 | peat+mud | 4.33 | 184.0 | 5.9 | 179.7 | 5.9 | | 23.66 | 2.69 |

| Upper Depth (cm) | Sediment Type [d] | Supported $^{210}$Pb ($^{226}$Ra, Bq/kg) | $^{210}$Pb total (Bq/kg) | +/- 1 SD (Bq/kg) | $^{210}$Pb$_{exs}$ (Bq/kq) | +/- 1 SD (Bq/kg) | Slope (cm$^2$/g) | Age (yr before 2016) | Age SD (yr) |
|---|---|---|---|---|---|---|---|---|---|
| **7\*** | **peat+mud** | | | | | | | **33.0** | **3.76** |
| 8 | peat+mud | 4.33 | 86.6 | 4.3 | 82.2 | 4.3 | | 36.78 | 4.19 |
| 11 | peat+mud | 4.33 | 65.9 | 3.9 | 61.6 | 3.9 | | 49.22 | 5.61 |
| 14 | peat+mud | 4.33 | 54.0 | 3.2 | 49.7 | 3.3 | | 61.82 | 7.04 |
| **15\*\*** | **peat+mud** | | | | | | | **66.7** | **7.59** |
| 18 | sand | 4.33 | 7.1 | 1.5 | 2.8 | 1.5 | | | |
| 18 ($^{226}$Ra) | | 4.33 | | 0.33 | | | | | |

**TMF 2-1 (HM)** [b]

| Upper Depth (cm) | Sediment Type [d] | Supported $^{210}$Pb ($^{226}$Ra, Bq/kg) | $^{210}$Pb total (Bq/kg) | +/- 1 SD (Bq/kg) | $^{210}$Pb$_{exs}$ (Bq/kq) | +/- 1 SD (Bq/kg) | Slope (cm$^2$/g) | Age (yr before 2016) | Age SD (yr) |
|---|---|---|---|---|---|---|---|---|---|
| 0 | soil | 3.50 | 194.4 | 7.2 | 190.9 | 7.2 | 0.063± 0.0027 | 0.34 | 0.01 |
| 4 | peat+mud | 3.50 | 184.9 | 7.0 | 181.4 | 7.0 | | 2.92 | 0.13 |
| 5 | peat+mud | 3.50 | 177.8 | 6.9 | 174.3 | 6.9 | | 4.19 | 0.18 |
| 6 | peat+mud | 3.50 | 173.1 | 6.8 | 169.6 | 6.8 | | 5.40 | 0.24 |
| 7 | peat+mud | 3.50 | 159.9 | 6.5 | 156.4 | 6.5 | | 7.04 | 0.31 |
| 8 | peat+mud | 3.50 | 148.2 | 6.3 | 144.7 | 6.3 | | 8.71 | 0.38 |
| 10 | peat+mud | 3.50 | 133.5 | 6.0 | 130.0 | 6.0 | | 11.29 | 0.49 |
| 11 | sand+peat | 3.50 | 120.9 | 5.4 | 117.4 | 5.4 | | 12.76 | 0.56 |
| 13 | sand+peat | 3.50 | 109.3 | 5.1 | 105.8 | 5.1 | | 15.65 | 0.68 |
| 16 | sand+peat | 3.50 | 98.1 | 4.6 | 94.6 | 4.7 | | 20.97 | 0.92 |
| **21\*** | **sand+peat** | **3.50** | **80.3** | **4.3** | **76.8** | **4.3** | | **31.84** | **1.39** |
| 24 | sand+peat | 3.50 | 67.9 | 3.8 | 64.4 | 3.8 | | 37.69 | 1.65 |
| **25\*\*** | **sand+peat** | | | | | | | **38.5** | **1.68** |
| 27 | sand+gravel | 3.50 | 52.7 | 0.5 | 49.2 | 0.6 | | | |
| 6 ($^{226}$Ra) | | 3.50 | | 0.33 | | | | | |

[a] measurements from Core Scientific International

[b] measurements from Flett Research Ltd

[c] measurements from My Core Scientific Laboratory

d "soil" represents a black, fluffy surface organic layer overlying peat in several cores. Brown peat layer below contained extensive, decaying plant roots. The transition between these two layers was visually distinct and also reflected in changes in dry bulk density

\* Depth of 30-yr horizon

\*\* Depth basal age (age determined in deepest sediment layer containing peat overlying sand, clay, or gravel)

Note that the 30-yr horizon can occur below the depth of the basal age.

**Figure A1. Core stratigraphies.** Dated high marsh cores (asterisks) were CBE1-1, CRF2-1, GBK1-2, and TMF2-1; Dated low marsh cores were CBE1-5, CRF3-2, GBK1-4, and TMF1-2. Corrected depth (cm) reflects correction for core compaction. Surface organic layer ("soil" in table ) overlying peat in several cores was black, fluffy, highly organic material overlying the brown peat layer below, which contained extensive plant roots that had not entirely decayed yet. The transition between these two layers was visually very distinct and also reflected by changes in dry bulk density.


**Figure A2. %LOI-to-%C Relationship. Relationship between measured LOI values and calculated %C, using elemental analyser EA data on set of 93 subsamples. Measurements from core CBE 1-5 were not used for calculating this relationship due to suspected measurement error.**

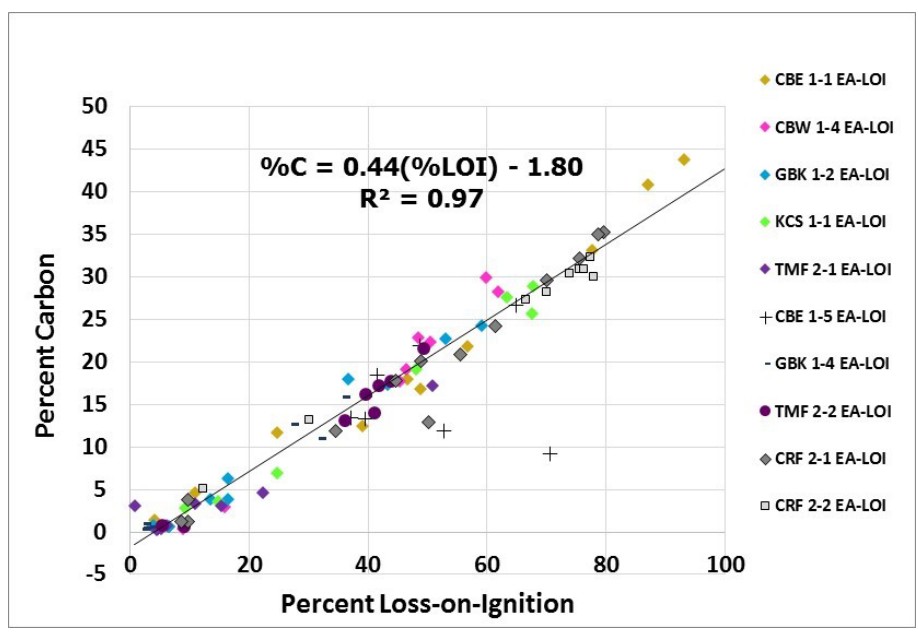

**Figure A3. Comparison of %OC calculation using the empirical regression determined for our study area (blue) and the model of Craft et al. (1991) (orange).**

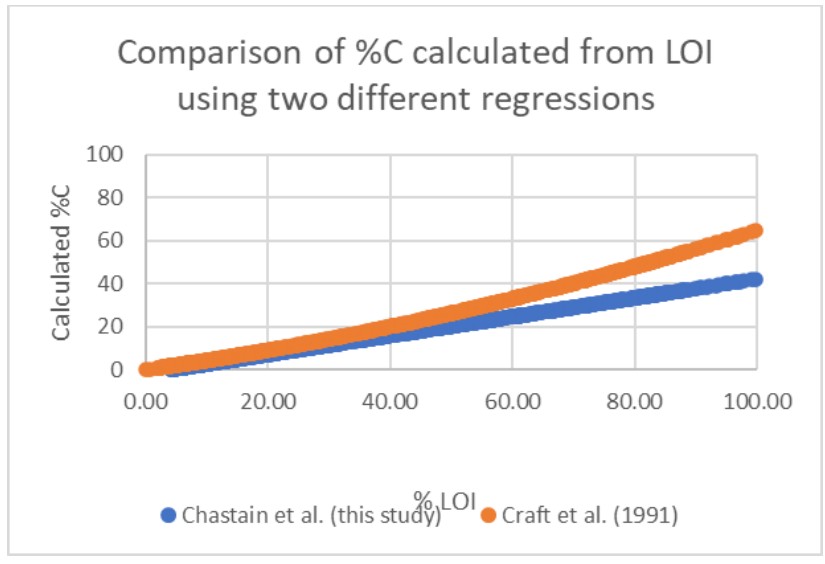

**Figure A4. Comparison of C stocks (Mg C/ha) estimated using the Craft et al. (1991) equation versus C stocks estimated using this study's empirical relationship for southern BC, for 8 cores from our study region. Comparison suggests that using the Craft et al. (1991) relationship would produce C stocks that are ~32% greater than our estimates. This difference, while substantial, would not account for global C stocks that are 3 times larger than those found in Clayoquot Sound salt marshes.**


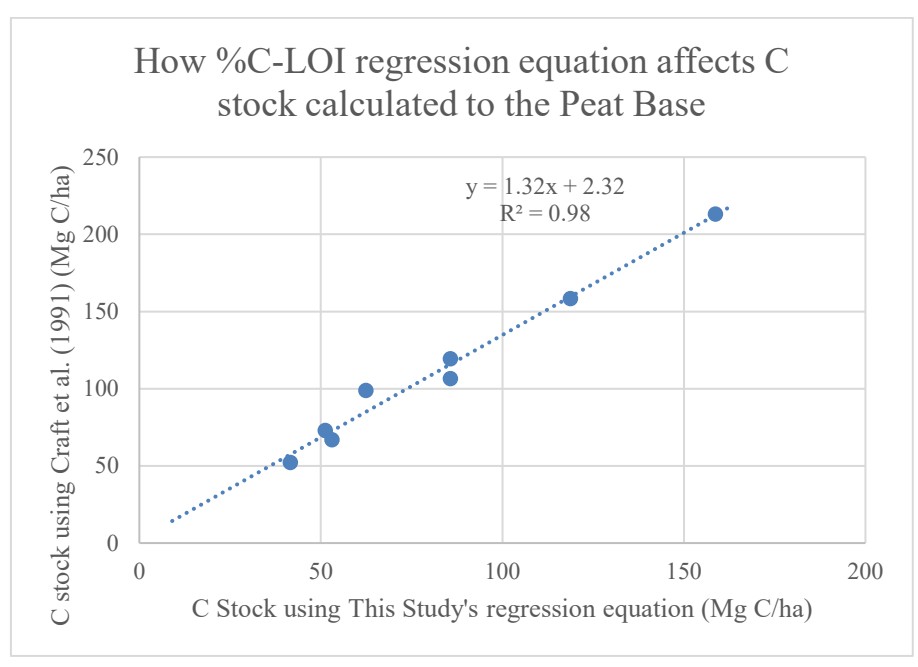


**Figure A5.** $^{210}$Pb inventory (Bq m$^{-2}$) in high and low marsh cores.

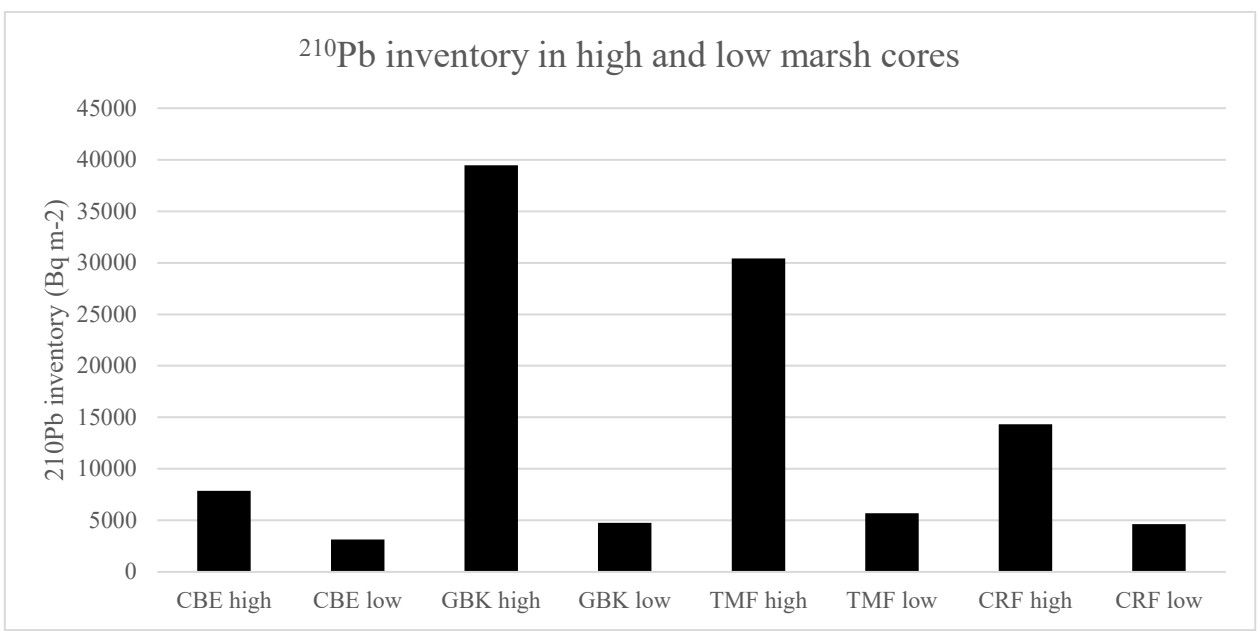

## Appendix B Groundtruthing Marsh Zone Designations: Detrended Correspondence Analysis


We used colour orthophotos to visually determine the areas of marsh zones using ArcMap 10.5.1 area measurement tools. A vegetation survey also determined the marsh zone of each core sample site, and vegetation data were examined using a detrended correspondence analysis to verify that vegetation assemblages reflected distinct low and high marsh zones identified from the orthophotos (Hill and Gauch 1980).


Our comparison showed that differentiating high and low marsh using orthophotos matched with field vegetation data for 32 of 34 (94 %) of cores. Both CRF 1-2 and CRF 2-2 were classified as low marsh by vegetation survey but fell within the high marsh using the visual orthophotography method. These cores lie 16 m (CRF 1-2) and 12 m (CRF 2-2) away from the low marsh boundary as measured using orthophotos, which is less than their distances from the nearest high marsh cores (17 m and 23 m, respectively). All other cores fell within the correct marsh zone.


We also conducted a detrended correspondence analysis (Hill and Gauch 1980) of vegetation data to determine that vegetation encountered in the vicinity of the core samples reflected distinct marsh zones using Canoco v4.5 software. We classified of marsh zones by presence/absence and percent cover of low marsh or high marsh vegetation (e.g. Deur 2000). The analysis showed accurate fit of low marsh cores with low marsh vegetation and high marsh cores with high marsh vegetation, plus the addition of a somewhat indistinct, third cluster of vegetation possibly representing the upland vegetation. This square root-


transformed model accounted for 33.2% of all variance in the vegetation dataset (sum of eigenvalues = 3.29). Cores with low marsh vegetation clustered together while high marsh cores clustered separately. An additional, slightly distinct third cluster of upland vegetation indicates that some high marsh cores may have been extracted from near the boundary with a freshwater-dominated upland vegetation or salt-tolerant meadow. The distinction between a salt marsh and a bordering freshwater area has complicated efforts to classify marshes by salinity (Duarte et al. 2013), but this result shows that clustering of vegetation


type corresponds reasonably well with each site's designation as high or low marsh.

**Figure B1 Detrended correspondence analysis results for Marsh vegetation data. Low marsh cores (top, purple squares) corresponded reasonably well with vegetation identified as low marsh, and high marsh cores corresponded with a distinct cluster of high marsh vegetation. The far bottom-right may indicate a population of less salt-tolerant, upland vegetation but it is indistinct from the high marsh.**


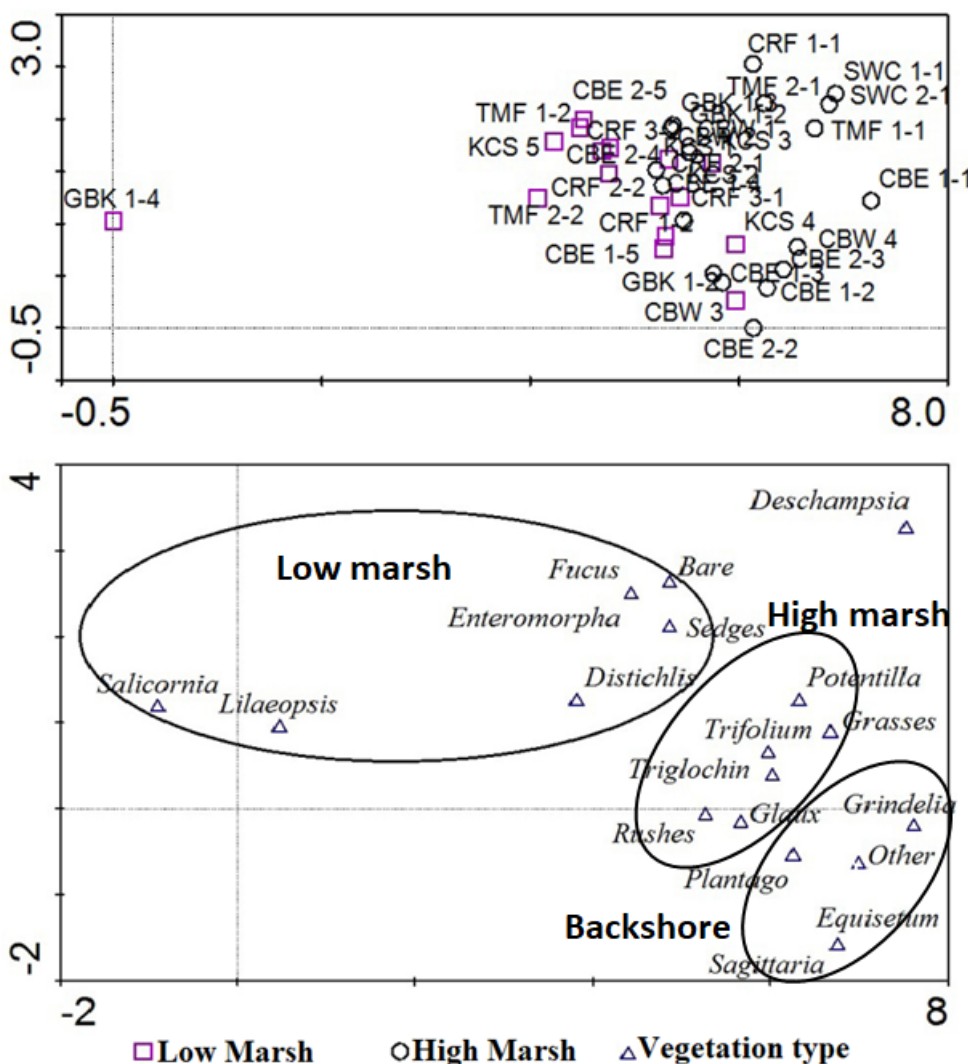

**Figure B2 Reference used for determining marsh stratum based on vegetation, adapted from Deur 2000. Dashed vertical lines from left to right represent mean time (1 m), mean lower high water (1.5 m), mean higher high water (1.85 m), and maximum water level (2.85 m).**

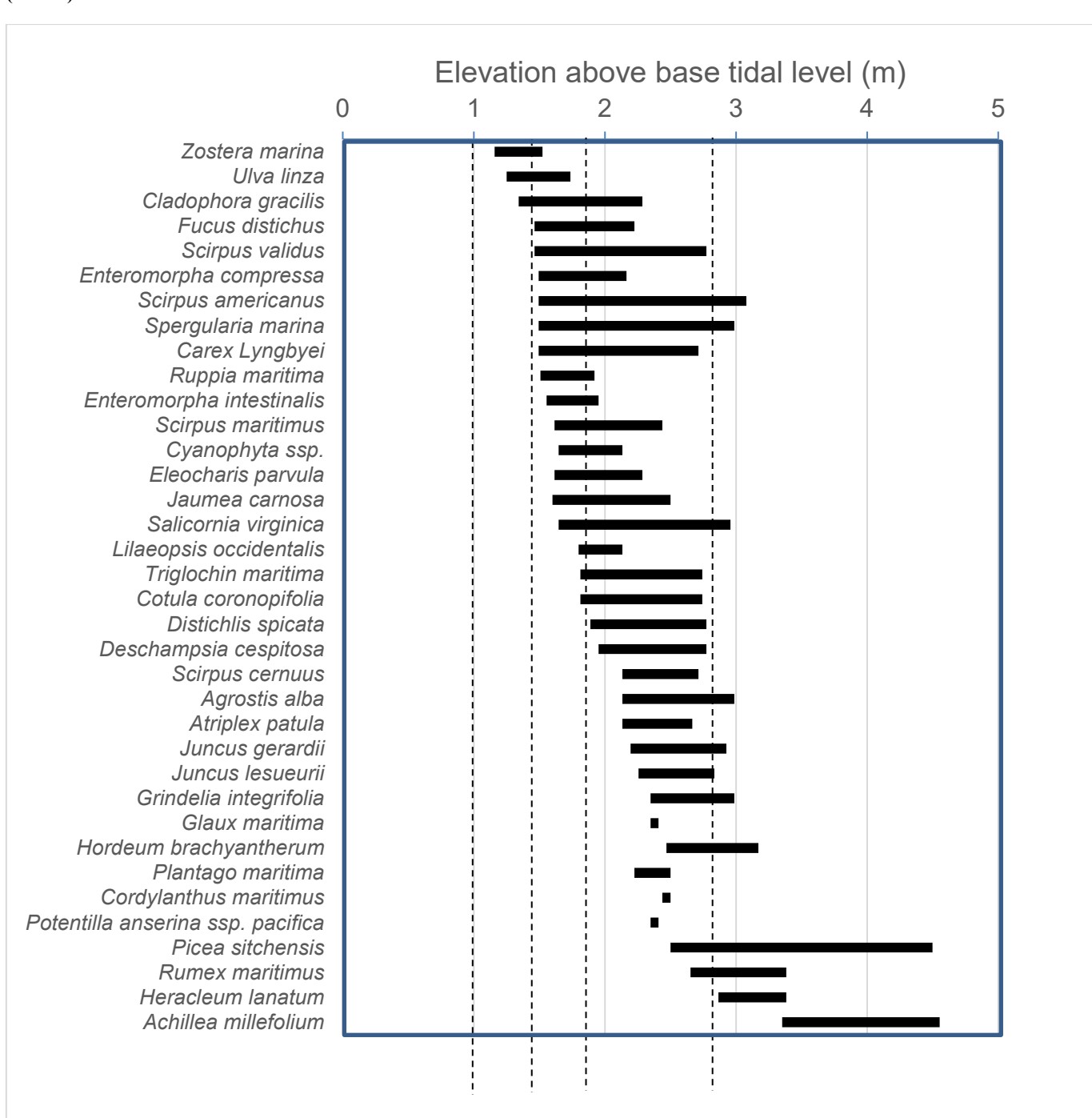

## Data Availability

Data produced as part of this project are currently being submitted to the Pangaea database with the following doi: https://doi.pangaea.de/10.1594/PANGAEA.947825

## Author contribution

MGP and KEK acquired funding for the project. SC, MGP, and KEK designed the experiments and SC and MGP carried them out. SC conducted laboratory measurements and SC, CO, MG, MGP and KEK contributed to data analysis. KEK, SC, and CO were responsible for data curation. KEK and SC prepared the manuscript and figures with contributions from all co-authors.

## Competing interests

The authors declare that they have no conflict of interest.

## Acknowledgements

The authors extend their gratitude to the following individuals for their contributions to this research, alphabetically: Dr. Richard Atleo, Celeste Barlow, Hasini Basnayake, Dr. Douglas Deur, Dan Harrison, Hannah Jensen, Victoria Lamothe, Dr. Dana Lepofsky, Gemma MacFarland, Aimee McGowan, Bryn Montgomery, Yiga Phuntsok, Ellie Simpson, Maureen Soon, and Dr. Nancy Turner. We thank Parks Canada for in-kind support at both Pacific Rim National Park Reserve and the Vancouver office, the Raincoast Education Society for their expertise in the field and access to their grounds to reach our study sites, and the Ahousaht First Nation for allowing us to collect samples from their territory.

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
