# Peer review of "Quantification of Blue Carbon in Salt Marshes of the Pacific Coast of Canada"

_Biogeosciences, 2021_

## Referee Comment (RC1)

Review

Chastain, S. G., Kohfeld, K. E., Pellatt, M. G., Olid, C., and Gailis, M.: Quantification of Blue Carbon in Salt Marshes of the Pacific Coast of Canada, Biogeosciences Discuss. [preprint], https://doi.org/10.5194/bg-2021-157, in review, 2021.

This comprehensive study fills a major gap in our knowledge of tidal marsh accretion and blue carbon as there is a lack of data on tidal marshes of the Northern NE Pacific coast. A major contribution is not only the geographic aspect, but also observations of C accumulation rates under regressive sea levels and the evaluation of low versus high marsh. The thorough, detailed explanation of all calculations makes the methodology clear and *most* of the results (see comments on compaction) justifiable. The approach to comparing 30-yr C stocks is novel and perhaps should be adapted as a standard for future studies of blue carbon stocks where dating models are available.

This work on the British Columbia coast could even further advance blue carbon science by providing details on the geomorphic context of each marsh. There is nascent research showing that the C stock of marshes is related to their geomorphic context (see van Ardenne, Jolicouer, Bérubé, Burdick, Chmura. 2018. The Importance of Geomorphic Context for Estimating the Carbon Stock of Salt Marshes. *Geoderma* 330:264-275). It would be useful to know if it plays a role in these British Columbia marshes, e.g., behind spits, on lagoons, fluvial marshes (as per Kelley JT, Gehrels WR, Belknap, DF, 1995. Late Holocene relative sea—level rise and the geological development of Tidal Marshes at Wells, Maine, U.S.A. *J. Coast. Res*. 11, 136–153.) or at least be available for future meta-analyses.

On line 359 – Authors state that C stocks per ha are less than 1/3 that of global estimates, undoubtedly due to the shallow marsh deposits that are less than the 50 cm depth used by Chmura et al. (2003). The estimate of Chmura et al. (2003) also utilized a formula published by Craft et al. (Craft CB, Seneca ED, Broome SW. 1991. Loss on ignition and Kjeldahl digestion for estimating organic carbon and total nitrogen in estuarine marsh soils: Calibration with dry combustion. *Estuaries* 14:175– 179.) to convert LOI to %OC and the authors used their own conversion, which results in lower values than what would be produced using Craft's. Would the stock still be <1/3 if authors had used the conversion of Craft et al? It would not be a terribly difficult exercise and would help to stimulate a re-evaluation of global carbon stocks.

The comparison of C accumulation rates in tidal marshes of Canada's Pacific coast to that of boreal forests is interesting and one cannot argue with the point that the considerably greater area of boreal forest makes them (presently) a greater C sink, despite the slow rates of C storage in the latter ecosystem. However, authors should recognize that with climate change the increased prevalence of forest fires would result in episodic losses of the carbon. If fire frequencies are too high, then there may not be time for succession to proceed to the needle leaf forest, shifting the landscape to a semi-permanent deciduous forest, with reduced carbon storage potential (see Melvin et al. 2015 *Ecosystems* 18:1472-1488). As sea level rise is not a threat to the Canadian Pacific salt marshes they are likely to continue to function as efficient C sinks despite global warming, and policy makers should be alerted to this fact.

Authors compare their results to averages reported in the review by Ouyang and Lee (2014). As this review has a number of errors with respect to double-counting records (e.g., averages of 3 sites were included as a 4th site) and attribution of geographic locations, its reports should be used with caution.

Some cores had high levels of **compaction**, due to use of percussion corers. (This type of coring should be the last choice when working in wetland soils as there are other devices that can be used which produce negligible or no compaction. For instance, authors do not mention trying a narrow diameter Dutch gouge corer, which often saves the day – or simply shoveling out a block and coring through the excavated material.) Although the compaction not a problem when calculating stocks to the base of the marsh deposit, it can affect bulk densities, thus carbon densities and the calculation of accumulation rates (one of the dated cores had 41% compaction). At line 200, the text states, "Here we estimated the accumulated C to the corrected (uncompacted) depth of 20 cm". Use of lead-210 inventories and 30 yr stocks help to address the complication of compaction, but authors should note how compaction was corrected for and how bulk densities were adjusted – this is very important and should be in the methodology. I assume that there was a threshold for compaction level beyond which cores were not used for calculation of bulk or carbon density and certainly lead-210.

Shouldn't the regression for the relationship of %LOI and %C be forced through zero? With a negative intercept a sample with no organic matter, thus 0% LOI would have a negative amount of carbon – an impossibility.

Clarification of and distinction amongst the terms "topsoil", "humus" and "peat" is needed. What is "topsoil" in a marsh? This term is not commonly used for wetland soils. The manuscript states see "Supplemental Information", but there is no explanation there. Also, the term "humus" is seldom used in wetland soils. Presumably it plant litter that is gradually broken down with depth? A bit of explanation would be helpful, even if just in a footnote to the Appendix table.

Line 518- Why would tidal amplitude be a driver of methane emissions? The paper cited on this line (Poffenbarger et al. 2011) reports that salinity, as a proxy for marine sulfates, is an important correlate.

The text and Figure B1 include "backshore" vegetation, a term not commonly seen in salt marsh ecology – it would be good to cite a paper that describes what this designates, beyond "less salt tolerant" vegetation. On Line 585 is the phrasing "freshwater-dominated backshore or salt-tolerant meadow" intended to indicate that these two are synonymous? I note that *Plantago maritima* is included in the "circle" of backshore vegetation, yet the text (line 114) includes it in high marsh. The distribution of *Plantago maritima* on the east and west Atlantic coasts does not suggest it has a low salt tolerance, so it might be advisable to adjust the bounds of the circle.

**Technical Editing**

Authors appropriately compare data to IPCC estimates. It would be preferable to cite the source chapter in the IPCC document: Kennedy HA, Alongi DM, Karim A, Chen G, Chmura GL, Crooks S, Kairo JG, Liao B, Lin G. 2013. Chapter 4 Coastal Wetlands In: *Supplement to the 2006 IPCC Guidelines on National Greenhouse Gas Inventories: Wetlands*.

Line 115 - Note that there has been a botanical revision of *Glaux maritima* to *Lysimachia maritima.*

Line 185 - Khrishnaswamy should be spelled Krishnaswamy

Line 356 – This statement could be more direct and not couched as "probably". If there is little difference in C density, then it is obvious that the shallower the soil/sediment/peat, the less carbon stock in that location.

Line 440 - Ryczik should be spelled Rybczyk.

Line 583 – why not replace "from close to" with "near"?

---

## Author Comment (AC1)

**RESPONSE TO RC1 COMMENT ON: Chastain, S. G., Kohfeld, K. E., Pellatt, M. G., Olid, C., and Gailis, M.: Quantification of Blue Carbon in Salt Marshes of the Pacific Coast of Canada, Biogeosciences Discuss. [preprint], https://doi.org/10.5194/bg-2021-157, in review, 2021.**

We thank Reviewer 1 for this helpful and detailed review of this manuscript. Below we outline our responses red text.

REVIEWER 1:
This comprehensive study fills a major gap in our knowledge of tidal marsh accretion and blue carbon as there is a lack of data on tidal marshes of the Northern NE Pacific coast. A major contribution is not only the geographic aspect, but also observations of C accumulation rates under regressive sea levels and the evaluation of low versus high marsh. The thorough, detailed explanation of all calculations makes the methodology clear and *most* of the results (see comments on compaction) justifiable. The approach to comparing 30-yr C stocks is novel and perhaps should be adapted as a standard for future studies of blue carbon stocks where dating models are available.

This work on the British Columbia coast could even further advance blue carbon science by providing details on the geomorphic context of each marsh. There is nascent research showing that the C stock of marshes is related to their geomorphic context (see van Ardenne, Jolicouer, Bérubé, Burdick, Chmura. 2018. The Importance of Geomorphic Context for Estimating the Carbon Stock of Salt Marshes. *Geoderma* 330:264-275). It would be useful to know if it plays a role in these British Columbia marshes, e.g., behind spits, on lagoons, fluvial marshes (as per Kelley JT, Gehrels WR, Belknap, DF, 1995. Late Holocene relative sea—level rise and the geological development of Tidal Marshes at Wells, Maine, U.S.A. *J. Coast. Res*. 11, 136–153.) or at least be available for future meta-analyses.

This is an interesting suggestion by the Reviewer 1. A direct comparison with the geomorphic contexts in van Ardenne et al. 2018 is somewhat challenging. The terrain around our study area does not really lead to the formation of spits and lagoons. Many of our sites were enclosed bays but they were not really cut off by spits. All locations were somewhat close to fluvial sources of varying size. Thus, applying the exact categories of van Ardenne et al. (2018) could be somewhat contrived here.

We do note that very recent work by van Ardenne, Hughes, and Chmura (JGR-Biogeosciences, 2021) has examined this question – albeit in fresher salt marsh systems - on the central BC coast. They argue that relating carbon density and marsh depth to geomorphology is difficult on a geomorphologically dynamic coastline as is found in our study area. We suggest that this might be an interesting topic to revisit in future.

On line 359 – Authors state that C stocks per ha are less than 1/3 that of global estimates, undoubtedly due to the shallow marsh deposits that are less than the 50 cm depth used by Chmura et al. (2003). The estimate of Chmura et al. (2003) also utilized a formula published by Craft et al. (Craft CB, Seneca ED, Broome SW. 1991. Loss on ignition and Kjeldahl digestion for estimating organic carbon and total nitrogen in estuarine marsh soils: Calibration with dry combustion. *Estuaries* 14:175– 179.) to convert LOI to %OC and the authors used their own

conversion, which results in lower values than what would be produced using Craft's. Would the stock still be <1/3 if authors had used the conversion of Craft et al? It would not be a terribly difficult exercise and would help to stimulate a re-evaluation of global carbon stocks.

Thank you for this interesting suggestion as a possible cause for the differences between the global and Pacific coast C stock estimates. The two equations mentioned here for calculating %OC from LOI are as follows:

Craft et al. (1991) polynomial regression: %OC = 0.40 [LOI] + 0.0025 [LOI]$^2$
Chastain et al. (this study) linear regression: $\%OC = 0.44(\%LOI) - 1.80$

We examined the effect of using the Craft et al. (1991) regression by calculating the differences in %OC that would result from using Craft et al.'s equation (see Figure R1.1.). The calculated %OC values are fairly similar for low values of %LOI (<~30%), but the %OC values diverge for %LOI values above this point, with calculated differences in %OC exceeding 20% at %LOIs above 80%. However, we note that the interquartile (Q1:Q3) range of our %C values fall between 4.39 to 28.84%, suggesting that most of our samples have %C values of less than 30% where the equations produce similar results. This seems to imply that the differences may not be too large.

[Figure]

*Figure R1.1. Comparison of %OC calculation using the empirical regression determined for our study area (blue) and the model of Craft et al. (1991) (orange).*

However, to test the potential impact of the different equations, we conducted a quick comparison of C stocks (estimated to peat base) generated using the two different %C-LOI relationships, just for the 8 cores that were [210]Pb dated (see Figure R1.2). Using the Craft et al. (1991) regression inflates our C-stock values by about 30%, but this is not sufficiently large to account for the full difference between our C stocks and the global values in Chmura et al. (2003) (where global average estimate is 3 times greater than ours).

**RESPONSE TO RC1 COMMENT ON: Chastain, S. G., Kohfeld, K. E., Pellatt, M. G., Olid, C., and Gailis, M.: Quantification of Blue Carbon in Salt Marshes of the Pacific Coast of Canada, Biogeosciences Discuss. [preprint], https://doi.org/10.5194/bg-2021-157, in review, 2021.**

*We prefer to use our equation because it is site specific but plan to note the potential effect of these different equations in our text.*

[Figure]

*Figure R1.2. Comparison of C stocks (Mg C/ha) estimated using the Craft et al. (1991) equation versus C stocks estimated using this study's empirical relationship for southern BC, for 8 cores from our study region. Comparison suggests that using the Craft et al. (1991) relationship would produce C stocks that are ~32% greater than our estimates. This difference, while substantial, would not account for the 3-fold difference between global C stocks and those found in Clayoquot Sound salt marshes.*

The comparison of C accumulation rates in tidal marshes of Canada's Pacific coast to that of boreal forests is interesting and one cannot argue with the point that the considerably greater area of boreal forest makes them (presently) a greater C sink, despite the slow rates of C storage in the latter ecosystem. However, authors should recognize that with climate change the increased prevalence of forest fires would result in episodic losses of the carbon. If fire frequencies are too high, then there may not be time for succession to proceed to the needle leaf forest, shifting the landscape to a semi-permanent deciduous forest, with reduced carbon storage potential (see Melvin et al. 2015 *Ecosystems* 18:1472-1488). As sea level rise is not a threat to the Canadian Pacific salt marshes they are likely to continue to function as efficient C sinks despite global warming, and policy makers should be alerted to this fact.

*We agree that these caveats should be added to the paper and plan to update this discussion based on more recent research on the impact of wildfires on the permanence of carbon storage. We thank the reviewer for the reference provided as a starting point. We will incorporate additional studies and information as needed (e.g. Kurz et al. 2013; Natural Resources Canada 2020, Zhao et al. 2021).*

**RESPONSE TO RC1 COMMENT ON: Chastain, S. G., Kohfeld, K. E., Pellatt, M. G., Olid, C., and Gailis, M.: Quantification of Blue Carbon in Salt Marshes of the Pacific Coast of Canada, Biogeosciences Discuss. [preprint], https://doi.org/10.5194/bg-2021-157, in review, 2021.**

Authors compare their results to averages reported in the review by Ouyang and Lee (2014). As this review has a number of errors with respect to double-counting records (e.g., averages of 3 sites were included as a 4th site) and attribution of geographic locations, its reports should be used with caution.

We agree that the Ouyang and Lee (2014) has incorporated some errors. Our comparison with the Ouyang and Lee (2014) paper was basically intended to point out the absence of dating for those records, which we discuss on lines 575-589.  We recognize, however, that we first mention the Ouyang and Lee (2014) comparison very early in the paper, prior to our discussion of these complications (Line 63-79).

To address this point we intend to incorporate additional caveats about Ouyang and Lee (2014) paper (i.e., potential issues with double-counting in addition to the reliance on $^{137}$Cs dates already mentioned), in the introduction when we first address the topic.

Some cores had high levels of **compaction**, due to use of percussion corers. (This type of coring should be the last choice when working in wetland soils as there are other devices that can be used which produce negligible or no compaction. For instance, authors do not mention trying a narrow diameter Dutch gouge corer, which often saves the day – or simply shoveling out a block and coring through the excavated material.) Although the compaction not a problem when calculating stocks to the base of the marsh deposit, it can affect bulk densities, thus carbon densities and the calculation of accumulation rates (one of the dated cores had 41% compaction). At line 200, the text states, "Here we estimated the accumulated C to the corrected (uncompacted) depth of 20 cm". Use of lead-210 inventories and 30 yr stocks help to address the complication of compaction, but authors should note how compaction was corrected for and how bulk densities were adjusted – this is very important and should be in the methodology. I assume that there was a threshold for compaction level beyond which cores were not used for calculation of bulk or carbon density and certainly lead-210.

We used the percussion corer as it was a closed chamber with internal PVC sleeves.  This was chosen over a gouge corer because they are susceptible to disturbance and sediment mixing due to the nature of the open chamber of the corer. We did not use a Russian corer as compaction would be similar because the nature of the marsh sediments and we did not want to introduce increased contamination by the pivoting nature of the sampling chamber. Digging pits with a shovel was not an option as this study took place in a national park and biosphere reserve.

To clarify this point we will add a short explanation in the methodology to explain why we sampled with a percussion corer, and the potential uncertainties or errors associated with the corer.

**RESPONSE TO RC1 COMMENT ON: Chastain, S. G., Kohfeld, K. E., Pellatt, M. G., Olid, C., and Gailis, M.: Quantification of Blue Carbon in Salt Marshes of the Pacific Coast of Canada, Biogeosciences Discuss. [preprint], https://doi.org/10.5194/bg-2021-157, in review, 2021.**

We will also review our description of the methodology to clarify when and where we corrected for compaction and how.

Shouldn't the regression for the relationship of %LOI and %C be forced through zero? With a negative intercept a sample with no organic matter, thus 0% LOI would have a negative amount of carbon – an impossibility.

Thank you for pointing this out. The relationship between %C and %LOI suggests that we measure zero %C in samples where LOI is not completely zero (below approximately 10% LOI). Although negative values of %C are obviously not possible, forcing the equation through zero would overestimate %C in these low LOI samples.  Therefore, all calculations producing a negative value for %C were adjusted to zero %C.  This occurred in 41 of 835 samples measured. Our methods have been clarified to reflect this change.

Clarification of and distinction amongst the terms "topsoil", "humus" and "peat" is needed. What is "topsoil" in a marsh? This term is not commonly used for wetland soils. The manuscript states see "Supplemental Information", but there is no explanation there. Also, the term "humus" is seldom used in wetland soils. Presumably it plant litter that is gradually broken down with depth? A bit of explanation would be helpful, even if just in a footnote to the Appendix table.

We take this point and have changed the term "topsoil" (which was used to describe the fibrous organic material within and below the root zone) as "peat."  We will also modify the term "humus" and add a more detailed explanation to the Appendix table.

Line 518- Why would tidal amplitude be a driver of methane emissions? The paper cited on this line (Poffenbarger et al. 2011) reports that salinity, as a proxy for marine sulfates, is an important correlate.

We appreciate this comment and we can replace the Poffenbarger et al. (2011) publication in this context, as there are several better citations that have measured changes in methane emissions associated with tidal activity and sea level rise (e.g. Abdul-Aziz et al. 2018; Emery et al. 2021; Huang et al. 2019; Huertas et al. 2019; Li et al. 2021; Wei et al. 2020.).

We will add 1-2 sentences to this discussion to reflect the results of these works and relate them to how the measurement of CH4 emissions will be important for assessing the overall blue C potential of the Clayoquot Sound marshes studied here.

**RESPONSE TO RC1 COMMENT ON: Chastain, S. G., Kohfeld, K. E., Pellatt, M. G., Olid, C., and Gailis, M.: Quantification of Blue Carbon in Salt Marshes of the Pacific Coast of Canada, Biogeosciences Discuss. [preprint], https://doi.org/10.5194/bg-2021-157, in review, 2021.**

The text and Figure B1 include "backshore" vegetation, a term not commonly seen in salt marsh ecology – it would be good to cite a paper that describes what this designates, beyond "less salt tolerant" vegetation.

We have taken this term from the following resource: Green Shores | Resources
https://stewardshipcentrebc.ca/green-shores-home/gs-resources/glossary/

Here "Backshore" is defined as "The upper zone of a beach (or land above the OHWM) beyond the reach of normal waves and tides, landward of the beach face. The backshore is subject to periodic flooding by storms and extreme tides, and is often the site of dunes and back-barrier wetlands"

We can change this term to "upland vegetation" if the reviewer is more comfortable with that.

On Line 585 is the phrasing "freshwater-dominated backshore or salt-tolerant meadow" intended to indicate that these two are synonymous? I note that *Plantago maritima* is included in the "circle" of backshore vegetation, yet the text (line 114) includes it in high marsh. The distribution of *Plantago maritima* on the east and west Atlantic coasts does not suggest it has a low salt tolerance, so it might be advisable to adjust the bounds of the circle.

Point taken and we have adjusted the bounds of the circle.

**Technical Editing**
Authors appropriately compare data to IPCC estimates. It would be preferable to cite the source chapter in the IPCC document: Kennedy HA, Alongi DM, Karim A, Chen G, Chmura GL, Crooks S, Kairo JG, Liao B, Lin G. 2013. Chapter 4 Coastal Wetlands In: *Supplement to the 2006 IPCC Guidelines on National Greenhouse Gas Inventories: Wetlands*.

Will do.

Line 115 - Note that there has been a botanical revision of *Glaux maritima* to *Lysimachia maritima.*

*Will change*

Line 185 - Khrishnaswamy should be spelled Krishnaswamy

Will change

Line 356 – This statement could be more direct and not couched as "probably". If there is little difference in C density, then it is obvious that the shallower the soil/sediment/peat, the less carbon stock in that location.

**RESPONSE TO RC1 COMMENT ON: Chastain, S. G., Kohfeld, K. E., Pellatt, M. G., Olid, C., and Gailis, M.: Quantification of Blue Carbon in Salt Marshes of the Pacific Coast of Canada, Biogeosciences Discuss. [preprint], https://doi.org/10.5194/bg-2021-157, in review, 2021.**

Will change

Line 440 - Ryczik should be spelled Rybczyk.

Will change

Line 583 – why not replace "from close to" with "near"?

Will change

REFERENCES CITED:

Abdul-Aziz, O. I., Ishtiaq, K. S., Tang, J., Moseman-Valtierra, S., Kroeger, K. D., Gonneea, M. E., et al. (2018). Environmental controls, emergent scaling, and predictions of greenhouse gas (GHG) fluxes in coastal salt marshes. *Journal of Geophysical Research: Biogeosciences*, 123, 2234– 2256. https://doi-org.proxy.lib.sfu.ca/10.1029/2018JG004556

Craft, C. & Seneca, E. & Broome, Stephen. (1991). Loss on Ignition and Kjeldahl Digestion for Estimating Organic Carbon and Total Nitrogen in Estuarine Marsh Soils: Calibration with Dry Combustion. Estuaries. 14. 175-179. 10.2307/1351691.

Emery, H. E., Angell, J. H., Tawade, A., and Fulweiler, R. W.: Tidal rewetting in salt marshes triggers pulses of nitrous oxide emissions but slows carbon dioxide emission, Soil Biology and Biochemistry, 156, 108197, 2021.

Huang J, Luo M, Liu Y, Zhang Y, Tan J. Effects of Tidal Scenarios on the Methane Emission Dynamics in the Subtropical Tidal Marshes of the Min River Estuary in Southeast China. *Int J Environ Res Public Health*. 2019;16(15):2790. Published 2019 Aug 5. doi:10.3390/ijerph16152790

Huertas, I. E., de la Paz, M., Perez, F. F., Navarro, G., and Flecha, S.: Methane Emissions From the Salt Marshes of Doñana Wetlands: Spatio-Temporal Variability and Controlling Factors, Frontiers in Ecology and Evolution, 7, 2019.

KurzW.A., ShawC.H., BoisvenueC., StinsonG., MetsarantaJ., LeckieD., DykA., SmythC., and NeilsonE.T.. Carbon in Canada's boreal forest — A synthesis. *Environmental Reviews*. **21**(4): 260-292. https://doi.org/10.1139/er-2013-0041

Li Y, Wang D, Chen Z, Chen J, Hu H, Wang R. Methane Emissions during the Tide Cycle of a Yangtze Estuary Salt Marsh. *Atmosphere*. 2021; 12(2):245. https://doi.org/10.3390/atmos12020245

**RESPONSE TO RC1 COMMENT ON: Chastain, S. G., Kohfeld, K. E., Pellatt, M. G., Olid, C., and Gailis, M.: Quantification of Blue Carbon in Salt Marshes of the Pacific Coast of Canada, Biogeosciences Discuss. [preprint], https://doi.org/10.5194/bg-2021-157, in review, 2021.**

Natural Resources Canada,The State of Canada's Forests. Annual Report 2020. 2020, Canadian Forest Service, Ottawa. 88 p. https://www.nrcan.gc.ca/our-natural-resources/forests-forestry/state-canadas-forests-report/16496

van Ardenne, L. B., Hughes, J. F., & Chmura, G. L. (2021). Tidal marsh sediment and carbon accretion on a geomorphologically dynamic coastline. *Journal of Geophysical Research: Biogeosciences*, 126, e2021JG006507. https://doi.org/10.1029/2021JG006507

Wei, S., Han, G., Chu, X., Song, W., He, W., Xia, J., and Wu, H.: Effect of tidal flooding on ecosystem $CO_2$ and $CH_4$ fluxes in a salt marsh in the Yellow River Delta, Estuarine, Coastal and Shelf Science, 232, 106512, 2020.

Zhao, B., Zhuang, Q., Shurpali, N. *et al.* North American boreal forests are a large carbon source due to wildfires from 1986 to 2016. *Sci Rep* **11,** 7723 (2021). https://doi.org/10.1038/s41598-021-87343-3

---

## Author Comment (AC2)

*Response to Reviewer 2 on Chastain et al., "Quantification of blue carbon in salt marshes of the Pacific Coast of Canada" submitted to Biogeosciences        14 February 2022*

*We thank Reviewer 2 for the helpful comments provided on this manuscript. Below we have put Reviewer comments in bold text and our responses in italicized text.*

**The paper reports data on 'blue' carbon stocks and 'blue' carbon accumulation rates from seven salt marshes at the west coast of Vancouver Island, BC, Canada. These seven salt marshes cover a total area of 47 ha. The authors differentiated between high and low marsh through identification of indicator plant species. They took in total 34 cores of the organic (peat) layer down to the underlying sand or gravel bed. In 10 cases the cores did not penetrate into the sand, clay or gravel layer. Eight cores (four from high and four from low marsh) were dated using 210Pb. The authors found an average total C stock to the base of the peat layer of 67 ± 9 Mg C ha-1 (mean ± SE) for all cores, which was less than one third of the globally averaged estimate of 250 Mg C ha-1 for salt marsh C stocks. In contrast, the average base of peat carbon accumulation rate (CAR) was 184 ± 50 g C m-2 yr-1, and in the high marsh even on average 303 ± 45 g C m-2 yr-1, which was about five times higher than in the low marsh areas. It has to be noted, though, that these CARs were based on four dated columns only per low and high marsh. In the discussion part, the authors put their findings into perspective of data from other salt marsh ecosystems along the Pacific and Atlantic coast of North America, and claim that they have addressed the knowledge gap regarding the carbon accumulation potential of these ecosystems. Finally, the compare the carbon accumulation of their salt marsh system with that of Canadian boreal forest and conclude that the carbon accrual rates are much higher in the salt marsh, but acknowledge that the salt marshes cover approximately only 0.016-0.1% of the area covered by boreal forests in Canada and that their absolute magnitude of carbon accumulation is only minor.**

**While the work appears to have been conducted in a scientifically sound way, and also the data have been well evaluated and compared with the literature, the representativeness of the studied system remains vague. The authors have taken 34 peat deposit cores in a range of about 25 km in an area that is subject to negative relative sea level rise due to uplift of this part of the coast. They attribute the below average carbon stocks of their salt marshes, amongst other reasons, to this relative sea level drop. But they do not detail to which extent this is representative or not for the vast Pacific coastline of North America, not to speak of the Atlantic coast. Also the effect of tidal range (the be more precise, its local differences) on the partitioning of marshes in 'high' and 'low' marshes, which obviously has a large effect on carbon accumulation rates and total C stocks, has not been addressed. That means, the authors did not put their 'mesotidal' system into perspective of other parts of the NE Pacific coast.**

**They only wrote "We expect that these mesotidal estuarine marshes, often constrained in size by surrounding topography, are typical of the marshes found on the Pacific coast of British Columbia" (L510-511). How is the situation in systems with higher or lower tidal range**

**than the one encountered here? In a nutshell, the authors should address the open question regarding the representativeness of their system in more detail.**

*Regarding the first point, the reviewer is correct that, within the financial constraints of the project, we have sampled 34 peat deposits from a marsh area of 47 ha within the Clayoquot Sound UNESCO Biosphere Reserve and Pacific Rim National Park Reserve and have been able to provide $^{210}$Pb dating for 8 (4 high marsh, 4 low marsh) cores. While this sampling may sound limited, it provides a substantial expansion over what has been sampled previously and adds the first $^{210}$Pb dated materials for the area. While some studies have provided a larger number of sampled and dated cores (e.g. Callaway et al. 2012; Suir et al. 2019; Brown, 2021), the assessment of carbon accumulation rates in many other marsh blue carbon studies are often completely lacking in 210Pb-dated quantification, particularly of high and low marsh environments separately (see e.g. supplemental information from Ouyang and Lee 2014 compilation). Therefore, this work represents a step forward in quantifying salt marsh processes, particularly on a previously under-sampled coastline.*

*That said, we appreciate the Reviewer's concern that we clarify our assumptions regarding our study's representativeness for the Pacific Coast of Canada and will do so in the revision. In particular, we can provide further discussion of sea level change along the Pacific Coast of Canada, which averages -0.76 ±1.32 mm/y (negative sea level change at our site is approximately -1.58 mm/y) (James et al. 2014). We will discuss how these differences in sea level rise could impact deposition.*

*Regarding the second point, we will include in the methods of our revised text that we were not able to measure the precise tidal range at each location and note that the average tidal range in nearby Tofino is ~2.7 m.  Within this system, the partitioning between high and low marsh zones (defined previously by tidal range) appears very closely linked with associated vegetation (which is seen in our canonical correspondence analysis). This tight coupling between vegetation type and marsh zone has also been observed in other studies on the west coast of Vancouver Island (Deur et al. 2000) and studies in nearby Boundary Bay (Gailis et al. 2021). Vegetation associations related to salinity and inundation are well documented and commonly used in salt marsh delineation (MacKenzie and Moran. 2004).*

*Importantly, we do not make any claim that these sites are at all representative of the Atlantic coast, and in fact, intended to make the point that the Pacific Coast environment is substantially different from the geomorphological and depositional environments along the passive margin of the Atlantic coast. We will clarify the text to make sure that this inference cannot be drawn from the text.  Finally, we note as well that recent global syntheses (Wang et al. 2020) have suggested that tidal range variables were not a significant driver of CAR in tidal wetlands.  We will include this in our discussion as well.*

**L136-137: How long were the cores stored under refrigeration?**

*Cores were collected in June and September, 2016. While in the field, cores were stored in portable coolers with ice packs. Long-term storage of cores was at Parks Canada laboratory in Vancouver, BC at 4C.*

**L207: How representative were these eight dated cores for your whole system (four for the high marsh, and four for the low marsh)?**

*As described above, we used the species composition of marsh vegetation communities to determine low and high marsh designations, by examining vegetation found within 50x50cm quadrats. We used these vegetation communities to establish representativeness of the high and low marsh sites that we sampled for $^{210}$Pb analysis. We can add a sentence to our methods clarifying this point.*

**L 485-488: Here you write "Our Clayoquot Sound data represent only a small area of a single region of the west coast, but if we assume our CAR estimate of 184 ± 50 g C m-2 yr-1 from Clayoquot Sound approximates the average for all tidal salt marshes in Canada,…". But the question is whether this generalization of your findings is justified. And if so, on what basis / with what assumptions?**

*We agree completely with the reviewer that our calculation is a vast oversimplification (although not unlike oversimplifications that have recently been published in other blue carbon literature). Our purpose in this text was not to provide a definitive estimate of blue carbon accumulation rates in Canadian salt marshes, but to provide a "back-of-the-envelope" comparison of the scale of accumulation relative to other ecosystems. We can make this point clearer in our text, to avoid having these numbers taken and repeated out of context.*

*We note that the CAR used in our calculation does fall within the range of expected values for worldwide and NW Atlantic CAR estimates of Ouyang and Lee (2014). However, in our revision we expand this analysis to encompass a wider range of possible CARs*

***References***
*Brown, D. R. (2021), Coastal wetland soil carbon sequestration revealed from sediment core profiles, Ph.D. thesis, Southern Cross University, DOI:https://doi.org/10.25918/thesis.167*

*Callaway, John & Borgnis, Evyan & Turner, Robert & Milan, Charles. (2012). Carbon Sequestration and Sediment Accretion in San Francisco Bay Tidal Wetlands. Estuaries and Coasts. 35. 10.1007/s12237-012-9508-9.*

*Deur, D.: A Domesticated Landscape: Native American plant cultivation on the Northwest coast of North America, PhD dissertation, Louisiana State University, Baton Rouge, LA, USA, pp. 69-251, 2000.*

*James, TS, JA Henton, LJ Leonard, A Darlington, DL Forbes, M Craymer (2014), Relative sea-level projections in Canada and the adjacent mainland United States, Geological Survey of Canada Open File 7737, 72 p. doi:* **10.4095/295574**

*MacKenzie, W.H. and J.R. Moran. 2004. Wetlands of British Columbia: a guide to identification. Res. Br., B.C. Minisitry of Forests, Victoria, B.C. Land Management Handbook No. 52*

*Ouyang, X. and Lee, S. Y.: Updated estimates of carbon accumulation rates in coastal marsh sediments, Biogeosciences, 11, 5057–5071, https://doi.org/10.5194/bg-11-5057-2014, 2014.*

*Suir et al. (2019), Comparing carbon accumulation in restored and natural wetland soils of coastal Louisiana, International Journal of Sediment Research, 34 (2019): 600-607.*

*Wang, F., Sanders, C.J., Santos, I.R., Tang, J., Schuerch, M., Kirwan, M.L., Kopp, R.E., Zhu, K., Li, X., Yuan, J. and Liu, W., 2021. Global blue carbon accumulation in tidal wetlands increases with climate change. National science review, 8(9), 296.*

---

## Author Response (AR1)

**RESPONSE TO REVIEWER COMMENTS: Chastain, S. G., Kohfeld, K. E., Pellatt, M. G., Olid, C., and Gailis, M.: Quantification of Blue Carbon in Salt Marshes of the Pacific Coast of Canada, Biogeosciences Discuss. [preprint], https://doi.org/10.5194/bg-2021-157, in review, 2021.**

*We thank both reviewers for their helpful and detailed reviews of this manuscript. Below we detail how we have addressed each review within our revised manuscript. Comments from reviewers are in bold; our responses are italicized. We provide the location of changes using the line numbers from the 'track changes' version of our resubmitted manuscript, with line numbers* *highlighted in yellow**.*
* * *
**REVIEWER 1:**
**This comprehensive study fills a major gap in our knowledge of tidal marsh accretion and blue carbon as there is a lack of data on tidal marshes of the Northern NE Pacific coast. A major contribution is not only the geographic aspect, but also observations of C accumulation rates under regressive sea levels and the evaluation of low versus high marsh. The thorough, detailed explanation of all calculations makes the methodology clear and *most* of the results (see comments on compaction) justifiable. The approach to comparing 30-yr C stocks is novel and perhaps should be adapted as a standard for future studies of blue carbon stocks where dating models are available.**

**This work on the British Columbia coast could even further advance blue carbon science by providing details on the geomorphic context of each marsh. There is nascent research showing that the C stock of marshes is related to their geomorphic context (see van Ardenne, Jolicouer, Bérubé, Burdick, Chmura. 2018. The Importance of Geomorphic Context for Estimating the Carbon Stock of Salt Marshes. *Geoderma* 330:264-275). It would be useful to know if it plays a role in these British Columbia marshes, e.g., behind spits, on lagoons, fluvial marshes (as per Kelley JT, Gehrels WR, Belknap, DF, 1995. Late Holocene relative sea—level rise and the geological development of Tidal Marshes at Wells, Maine, U.S.A. *J. Coast. Res.* 11, 136–153.) or at least be available for future meta-analyses.**

*A direct comparison with the geomorphic contexts in van Ardenne et al. 2018 is somewhat challenging because the terrain around our study area does not involve formation of spits and lagoons. Many of our sites were enclosed bays but they were not really cut off by spits. All locations were somewhat close to fluvial sources of varying size. Thus, applying the exact categories of van Ardenne et al. (2018) could be somewhat contrived here.*

*We do note that recent work by van Ardenne et al. (2021) has examined this question – albeit in fresher marsh systems - on the central BC coast. They argue that relating carbon density and marsh depth to geomorphology is difficult on a geomorphologically dynamic coastline as is found in our study area. We suggest that this might be an interesting topic to revisit in future.*

*We have added a comment about geomorphology here (**In 106-108**):*

*"These sites are typical of salt marshes along Canada's Pacific coast because they include small, pocket marshes encompassing an enclosed, semi-circular area of coastline as well as larger,*

**RESPONSE TO REVIEWER COMMENTS: Chastain, S. G., Kohfeld, K. E., Pellatt, M. G., Olid, C., and Gailis, M.: Quantification of Blue Carbon in Salt Marshes of the Pacific Coast of Canada, Biogeosciences Discuss. [preprint], https://doi.org/10.5194/bg-2021-157, in review, 2021.**

*estuarine marshes. Unlike geomorphological settings in Atlantic Canada (e.g. van Ardenne et al. 2018), we do not see extensive formation of spits and lagoons; many of our sites were in enclosed bays but were not cut off by spits. All sites were somewhat close to fluvial sources of varying size. Surface water salinity in the surrounding waters ranged from 5.9 at KCS to 24 in Grice Bay, and 29 at Roberts Point six km south of CRF (Postlethwaite et al. 2018)."*

**On line 359 – Authors state that C stocks per ha are less than 1/3 that of global estimates, undoubtedly due to the shallow marsh deposits that are less than the 50 cm depth used by Chmura et al. (2003). The estimate of Chmura et al. (2003) also utilized a formula published by Craft et al. (Craft CB, Seneca ED, Broome SW. 1991. Loss on ignition and Kjeldahl digestion for estimating organic carbon and total nitrogen in estuarine marsh soils: Calibration with dry combustion. *Estuaries* 14:175– 179.) to convert LOI to %OC and the authors used their own conversion, which results in lower values than what would be produced using Craft's. Would the stock still be <1/3 if authors had used the conversion of Craft et al? It would not be a terribly difficult exercise and would help to stimulate a re-evaluation of global carbon stocks.**

*Thank you for this interesting suggestion as a possible cause for the differences between the global and Pacific coast C stock estimates. The two equations mentioned here for calculating %OC from LOI are as follows:*

*Craft et al. (1991) polynomial regression: %OC = 0.40 [LOI] + 0.0025 [LOI]$^2$*
*Chastain et al. (this study) linear regression: %OC=0.44(%LOI)−1.80*

*We examined the effect of using the Craft et al. (1991) regression by calculating the differences in %OC that would result from using Craft et al.'s equation (see Figure R1.1.). The calculated %OC values are fairly similar for low values of %LOI (<~30%), but the %OC values diverge for %LOI values above this point, with calculated differences in %OC exceeding 20% at %LOIs above 80%. However, we note that the interquartile (Q1:Q3) range of our %C values fall between 4.39 to 28.84%, suggesting that most of our samples have %C values of less than 30% where the equations produce similar results. This seems to imply that the differences may not be too large.*

*However, to test the potential impact of the different equations, we conducted a quick comparison of C stocks (estimated to peat base) generated using the two different %C-LOI relationships, just for the 8 cores that were $^{210}$Pb dated (see Figure R1.2). Using the Craft et al. (1991) regression inflates our C-stock values by about 30%, but this is not sufficiently large to account for the full difference between our C stocks and the global values in Chmura et al. (2003) (where global average estimate is 3 times greater than ours).*

*We prefer to use our equation because it is site specific but plan to note the potential effect of these different equations in our text.*

[Figure]

***Figure R1.1 (new Fig A3 in appendix). Comparison of %OC calculation using the empirical regression determined for our study area (blue) and the model of Craft et al. (1991) (orange).***

[Figure]

***Figure R1.2 (new Fig A4 in appendix). Comparison of C stocks (Mg C/ha) estimated using the Craft et al. (1991) equation versus C stocks estimated using this study's empirical relationship for southern BC, for 8 cores from our study region. Comparison suggests that using the Craft et al. (1991) relationship would produce C stocks that are ~32% greater than our estimates. This difference, while substantial, would not account for global C stocks that are 3 times larger than those found in Clayoquot Sound salt marshes.***

*To address this concern, we have added the following text to the Discussion (ln 391-402): "The C stocks in Clayoquot Sound marshes are less than one-third of the globally averaged estimate of 250 Mg C ha$^{-1}$ for salt marsh C stocks in the upper 50 cm (Chmura et al. 2003; Pendleton et al. 2012). This is true whether we consider the base of peat, 20-cm, or 30-yr*

*estimates (Table 1). One possible contributor to these discrepancies could be the use of different formulas relating %C to LOI. Chmura et al. (2003) utilized the formula of Craft et al. (1991) to convert LOI measurements to %C. In contrast, we used an empirical relationship based on measurements collected from Clayoquot Sound marsh samples. Comparison of these two regression equations suggest that estimates of %C are very similar for values of %C equal to or less than 30%, but the %C estimates diverge for percentages above 30% (Fig. A3), with calculated differences in %C exceeding 20% at %LOIs above 80%. However, we note that the interquartile (Q1:Q3) range of our %C values fall between 4.4 and 28.8%, suggesting that most of our samples have %C values of less than 30%, where the two equations produce similar results.  To test the potential impact of the different equations, we compared C stocks (estimated to peat base) that were calculated using the two different %C-LOI relationships for the eight cores that were $^{210}$Pb dated (Fig. A4). Using the Craft et al. (1991) regression inflates our C stock values by about 30%, but this increase is not sufficiently large to account for the full difference between our C stocks and the global values in Chmura et al. (2003), which are three times greater than our estimated C stocks."*

**The comparison of C accumulation rates in tidal marshes of Canada's Pacific coast to that of boreal forests is interesting and one cannot argue with the point that the considerably greater area of boreal forest makes them (presently) a greater C sink, despite the slow rates of C storage in the latter ecosystem. However, authors should recognize that with climate change the increased prevalence of forest fires would result in episodic losses of the carbon. If fire frequencies are too high, then there may not be time for succession to proceed to the needle leaf forest, shifting the landscape to a semi-permanent deciduous forest, with reduced carbon storage potential (see Melvin et al. 2015 *Ecosystems* 18:1472-1488). As sea level rise is not a threat to the Canadian Pacific salt marshes they are likely to continue to function as efficient C sinks despite global warming, and policy makers should be alerted to this fact.**

*We agree that these caveats should be added to the paper and have updated this discussion based on more recent research on the impact of climate change and wildfires on the permanence of carbon storage (see Section 4.4)*

**Authors compare their results to averages reported in the review by Ouyang and Lee (2014). As this review has a number of errors with respect to double-counting records (e.g., averages of 3 sites were included as a 4th site) and attribution of geographic locations, its reports should be used with caution.**

*We agree that the Ouyang and Lee (2014) has incorporated some errors. Our comparison with the Ouyang and Lee (2014) paper was basically intended to point out the absence of dating for those records. To address this point we have incorporated additional caveats about Ouyang and Lee (2014) paper (i.e., potential issues with double-counting in addition to the reliance on $^{137}$Cs*

**RESPONSE TO REVIEWER COMMENTS: Chastain, S. G., Kohfeld, K. E., Pellatt, M. G., Olid, C., and Gailis, M.: Quantification of Blue Carbon in Salt Marshes of the Pacific Coast of Canada, Biogeosciences Discuss. [preprint], https://doi.org/10.5194/bg-2021-157, in review, 2021.**

*dates already mentioned) prior to our main comparison with the compilation in the Discussion* (ln 437-441):

*"One effort has been made to assemble a global compilation and synthesis of CARs within salt marsh ecosystems (Ouyang and Lee, 2014). Here we compare our results with this compilation with the caveats that (a) Ouyang and Lee (2014) relied heavily on $^{137}$Cs dating or marker horizon methods for their estimates, and (b) appears to have some instances of double-counting of sites (e.g., averages of 3 sites were included as a 4th site) and minor issues with attribution of some geographic locations. That said, we note that…"*

**Some cores had high levels of compaction, due to use of percussion corers. (This type of coring should be the last choice when working in wetland soils as there are other devices that can be used which produce negligible or no compaction. For instance, authors do not mention trying a narrow diameter Dutch gouge corer, which often saves the day – or simply shoveling out a block and coring through the excavated material.) Although the compaction not a problem when calculating stocks to the base of the marsh deposit, it can affect bulk densities, thus carbon densities and the calculation of accumulation rates (one of the dated cores had 41% compaction). At line 200, the text states, "Here we estimated the accumulated C to the corrected (uncompacted) depth of 20 cm". Use of lead-210 inventories and 30 yr stocks help to address the complication of compaction, but authors should note how compaction was corrected for and how bulk densities were adjusted – this is very important and should be in the methodology. I assume that there was a threshold for compaction level beyond which cores were not used for calculation of bulk or carbon density and certainly lead-210.**

*We have added an explanation in the Methods section to explain why we sampled with a percussion corer, in which we quantify the effects of compaction on our sediment cores. We also provide a brief explanation for how we have accounted for compaction* (ln 157-168)*:*

*"Use of the percussion corer resulted in sediment compaction during sample collection, which averaged about 20% across all cores (range 0-55%) (Table A1). Nevertheless, we opted to use a percussion corer instead of a gouge corer because the percussion corer had a closed chamber with internal PVC sleeves. Our experience with this sedimentary has demonstrated that a gouge corer would have been susceptible to disturbance and sediment mixing due to the nature of the open chamber of the corer. Because the nature of the marsh sediments, we also did not use a Russian corer because compaction would have been similar to what we experienced with the percussion corer, and we did not want to introduce increased contamination through the pivoting nature of the sampling chamber with the Russian corer. Digging pits with a shovel was not an option as this study took place in a national park and biosphere reserve. We note below that correction for compaction was not necessary for estimation of C stocks because the C stocks were estimated directly from sediment cores and not from the overall depth of marsh soils (thus all carbon in the peat layer, regardless of compression, is included in the calculation).*

**RESPONSE TO REVIEWER COMMENTS: Chastain, S. G., Kohfeld, K. E., Pellatt, M. G., Olid, C., and Gailis, M.: Quantification of Blue Carbon in Salt Marshes of the Pacific Coast of Canada, Biogeosciences Discuss. [preprint], https://doi.org/10.5194/bg-2021-157, in review, 2021.**

*However, when we do need to account for compaction (e.g. Figures 2-3), we use a compaction factor (Howard et al. 2014; Gailis et al. 2021) estimated for each core by dividing the length of core penetration by the length of core recovered (Table A1)."*

**Shouldn't the regression for the relationship of %LOI and %C be forced through zero? With a negative intercept a sample with no organic matter, thus 0% LOI would have a negative amount of carbon – an impossibility.**

*Thank you for pointing this out. The relationship between %C and %LOI suggests that we measure zero %C in samples where LOI is not completely zero (below approximately 10% LOI). Although negative values of %C are obviously not possible, forcing the equation through zero would overestimate %C in these low LOI samples.  Therefore, all calculations producing a negative value for %C were adjusted to zero %C.  This occurred in 41 of 835 samples measured. Our methods have been clarified to reflect this change ==(LN 193).==*

**Clarification of and distinction amongst the terms "topsoil", "humus" and "peat" is needed. What is "topsoil" in a marsh? This term is not commonly used for wetland soils. The manuscript states see "Supplemental Information", but there is no explanation there. Also, the term "humus" is seldom used in wetland soils. Presumably it plant litter that is gradually broken down with depth? A bit of explanation would be helpful, even if just in a footnote to the Appendix table.**

*We take this point and have changed the term "topsoil" (which was used to describe the fibrous organic material within and below the root zone) as "peat."  However, we have kept use of the term "humus" as term that has been used as a descriptor in other salt marsh publications (e.g. Goni and Thomas, 2000; Santin et al. 2008)*

**Line 518- Why would tidal amplitude be a driver of methane emissions? The paper cited on this line (Poffenbarger et al. 2011) reports that salinity, as a proxy for marine sulfates, is an important correlate.**

*We appreciate this comment and we can replace the Poffenbarger et al. (2011) publication in this context, as there are several better citations that have measured changes in methane emissions associated with tidal activity and sea level rise on ==LN 608==: (e.g. Abdul-Aziz et al. 2018; Huang et al. 2019; Huertas et al. 2019; Li et al. 2021; Wei et al. 2020.).*

**The text and Figure B1 include "backshore" vegetation, a term not commonly seen in salt marsh ecology – it would be good to cite a paper that describes what this designates, beyond "less salt tolerant" vegetation.**

**RESPONSE TO REVIEWER COMMENTS: Chastain, S. G., Kohfeld, K. E., Pellatt, M. G., Olid, C., and Gailis, M.: Quantification of Blue Carbon in Salt Marshes of the Pacific Coast of Canada, Biogeosciences Discuss. [preprint], https://doi.org/10.5194/bg-2021-157, in review, 2021.**

*We have taken this term from the following resource: Green Shores | Resources*
*https://stewardshipcentrebc.ca/green-shores-home/gs-resources/glossary/*

*Here "Backshore" is defined as "The upper zone of a beach (or land above the OHWM) beyond the reach of normal waves and tides, landward of the beach face. The backshore is subject to periodic flooding by storms and extreme tides, and is often the site of dunes and back-barrier wetlands"*

*We have changed this term to "upland vegetation."*

**On Line 585 is the phrasing "freshwater-dominated backshore or salt-tolerant meadow" intended to indicate that these two are synonymous? I note that *Plantago maritima* is included in the "circle" of backshore vegetation, yet the text (line 114) includes it in high marsh. The distribution of *Plantago maritima* on the east and west Atlantic coasts does not suggest it has a low salt tolerance, so it might be advisable to adjust the bounds of the circle.**
*Point taken and we have adjusted the bounds of the circle.*

**Technical Editing**
**Authors appropriately compare data to IPCC estimates. It would be preferable to cite the source chapter in the IPCC document: Kennedy HA, Alongi DM, Karim A, Chen G, Chmura GL, Crooks S, Kairo JG, Liao B, Lin G. 2013. Chapter 4 Coastal Wetlands In: *Supplement to the 2006 IPCC Guidelines on National Greenhouse Gas Inventories: Wetlands*.**
*changed.*

**Line 115 - Note that there has been a botanical revision of *Glaux maritima* to *Lysimachia maritima.***
*Changed.*

**Line 185 - Khrishnaswamy should be spelled Krishnaswamy**
*changed.*

**Line 356 – This statement could be more direct and not couched as "probably". If there is little difference in C density, then it is obvious that the shallower the soil/sediment/peat, the less carbon stock in that location.**
*"probably" removed.*

**Line 440 - Ryczik should be spelled Rybczyk.**

*changed.*

**Line 583 – why not replace "from close to" with "near"?**

*changed.*

**RESPONSE TO REVIEWER COMMENTS:** Chastain, S. G., Kohfeld, K. E., Pellatt, M. G., Olid, C., and Gailis, M.: Quantification of Blue Carbon in Salt Marshes of the Pacific Coast of Canada, Biogeosciences Discuss. [preprint], https://doi.org/10.5194/bg-2021-157, in review, 2021.
* * *
**REVIEWER 2.**

*We thank Reviewer 2 for the helpful comments provided on this manuscript. Below we have put Reviewer comments in bold text and our responses in italicized text.*

**The paper reports data on 'blue' carbon stocks and 'blue' carbon accumulation rates from seven salt marshes at the west coast of Vancouver Island, BC, Canada. These seven salt marshes cover a total area of 47 ha. The authors differentiated between high and low marsh through identification of indicator plant species. They took in total 34 cores of the organic (peat) layer down to the underlying sand or gravel bed. In 10 cases the cores did not penetrate into the sand, clay or gravel layer. Eight cores (four from high and four from low marsh) were dated using 210Pb. The authors found an average total C stock to the base of the peat layer of 67 ± 9 Mg C ha-1 (mean ± SE) for all cores, which was less than one third of the globally averaged estimate of 250 Mg C ha-1 for salt marsh C stocks. In contrast, the average base of peat carbon accumulation rate (CAR) was 184 ± 50 g C m-2 yr-1, and in the high marsh even on average 303 ± 45 g C m-2 yr-1, which was about five times higher than in the low marsh areas. It has to be noted, though, that these CARs were based on four dated columns only per low and high marsh. In the discussion part, the authors put their findings into perspective of data from other salt marsh ecosystems along the Pacific and Atlantic coast of North America, and claim that they have addressed the knowledge gap regarding the carbon accumulation potential of these ecosystems. Finally, the compare the carbon accumulation of their salt marsh system with that of Canadian boreal forest and conclude that the carbon accrual rates are much higher in the salt marsh, but acknowledge that the salt marshes cover approximately only 0.016-0.1% of the area covered by boreal forests in Canada and that their absolute magnitude of carbon accumulation is only minor.**

**While the work appears to have been conducted in a scientifically sound way, and also the data have been well evaluated and compared with the literature, the representativeness of the studied system remains vague. The authors have taken 34 peat deposit cores in a range of about 25 km in an area that is subject to negative relative sea level rise due to uplift of this part of the coast. They attribute the below average carbon stocks of their salt marshes, amongst other reasons, to this relative sea level drop. But they do not detail to which extent this is representative or not for the vast Pacific coastline of North America, not to speak of the Atlantic coast. Also the effect of tidal range (the be more precise, its local differences) on the partitioning of marshes in 'high' and 'low' marshes, which obviously has a large effect on carbon accumulation rates and total C stocks, has not been addressed. That means, the authors did not put their 'mesotidal' system into perspective of other parts of the NE Pacific coast.**

**They only wrote "We expect that these mesotidal estuarine marshes, often constrained in size by surrounding topography, are typical of the marshes found on the Pacific coast of**

**RESPONSE TO REVIEWER COMMENTS: Chastain, S. G., Kohfeld, K. E., Pellatt, M. G., Olid, C., and Gailis, M.: Quantification of Blue Carbon in Salt Marshes of the Pacific Coast of Canada, Biogeosciences Discuss. [preprint], https://doi.org/10.5194/bg-2021-157, in review, 2021.**

**British Columbia" (L510-511). How is the situation in systems with higher or lower tidal range than the one encountered here? In a nutshell, the authors should address the open question regarding the representativeness of their system in more detail.**

*Regarding the first point, the reviewer is correct that, within the financial constraints of the project, we have sampled 34 peat deposits from a marsh area of 47 ha within the Clayoquot Sound UNESCO Biosphere Reserve and Pacific Rim National Park Reserve and have been able to provide $^{210}$Pb dating for 8 (4 high marsh, 4 low marsh) cores. While this sampling may sound limited, it provides a substantial expansion over what has been sampled previously and adds the first $^{210}$Pb dated materials for the area. While some studies have provided a larger number of sampled and dated cores (e.g. Callaway et al. 2012; Suir et al. 2019; Brown, 2021), the assessment of carbon accumulation rates in many other marsh blue carbon studies are often completely lacking in $^{210}$Pb-dated quantification, particularly of high and low marsh environments separately (see e.g. supplemental information from Ouyang and Lee 2014 compilation). Therefore, this work represents a step forward in quantifying salt marsh processes, particularly on a previously under-sampled coastline.*

*That said, we appreciate the Reviewer's concern that we clarify our assumptions regarding our study's representativeness for the Pacific Coast of Canada and have addressed this concern in the following ways:*

(1) *In our discussion of the impacts of sea level rise on vertical accretion rates and carbon accumulation rates (Ln 471-473), we provide a better estimate of sea level change along the Pacific Coast of Canada, which averages -0.76 ±1.32 mm/y (James et al. 2014).*

(2) *Regarding the second point, we have included in the methods that we were not able to measure the precise tidal range at each location and note that the average tidal range in nearby Tofino is ~2.7 m. (Ln 104-105). (We do note, however, that recent global syntheses (Wang et al. 2020) have suggested that tidal range variables were not a significant driver of CAR in tidal wetlands.)*

(3) *Additionally, within this system, the partitioning between high and low marsh zones (defined previously by tidal range) appears very closely linked with associated vegetation (which is seen in our canonical correspondence analysis). We have noted this in Ln 113-114. We have also added the following sentence to the end of the paragraph describing our marsh zonation methodology (Ln 126-130):*

> *"This tight coupling between vegetation type and marsh zone has also been observed in other studies on the west coast of Vancouver Island (Deur et al. 2000) and studies in nearby Boundary Bay (Gailis et al. 2021). Vegetation associations related to salinity and inundation are well documented and commonly used in salt marsh delineation (MacKenzie and Moran. 2004)."*

(4) *Importantly, we do not make any claim that these sites are at all representative of the Atlantic coast, and in fact, intended to make the point that the Pacific Coast environment*

*is substantially different from the geomorphological and depositional environments along the passive margin of the Atlantic coast, and that formal estimates of C sequestration require better documentation of C stocks, accumulation rates, and areas. We have (hopefully) addressed this concern in our rephrasing of our estimate of pan-Canadian carbon accumulation rates (Section 4.4, Ln 550-594), where we have (a) recalculated the Canada-wide carbon accumulation using an Atlantic Canada estimate and the global average CAR estimate from the IPCC, and (b) explicitly stated that a pan-Canadian CAR estimate (and associated uncertainty) should be a subject of future research.*

**L136-137: How long were the cores stored under refrigeration?**

*Cores were collected in June and September, 2016. While in the field, cores were stored in portable coolers with ice packs. Long-term storage of cores was at Parks Canada laboratory in Vancouver, BC at 4C. This information has been added to the Methods (LN 155-156)*

**L207: How representative were these eight dated cores for your whole system (four for the high marsh, and four for the low marsh)?**

*As described above, we used the species composition of marsh vegetation communities to determine low and high marsh designations, by examining vegetation found within 50x50cm quadrats. We used these vegetation communities to establish representativeness of the high and low marsh sites that we sampled for $^{210}$Pb analysis. We have clarified this point in the methods (ln 205-206)*

**L 485-488: Here you write "Our Clayoquot Sound data represent only a small area of a single region of the west coast, but if we assume our CAR estimate of 184 ± 50 g C m-2 yr-1 from Clayoquot Sound approximates the average for all tidal salt marshes in Canada,…". But the question is whether this generalization of your findings is justified. And if so, on what basis / with what assumptions?**

*We agree completely with the reviewer that our calculation is a vast oversimplification (although not unlike oversimplifications that have recently been published in other blue carbon literature). Our purpose in this text was not to provide a definitive estimate of blue carbon accumulation rates in Canadian salt marshes, but to provide a "back-of-the-envelope" comparison of the scale of accumulation relative to other ecosystems. We have revised this section to make the speculative nature of our calculation clearer, and to emphasize what work needs to be done to make this estimate more robust. (see revised Section 4.4).*

*(We note, however, that the CAR used in our calculation does fall within the range of expected values for worldwide and NW Atlantic CAR estimates of Ouyang and Lee (2014). However, in our revision we expand this analysis to encompass a wider range of possible CARs.)*

**RESPONSE TO REVIEWER COMMENTS: Chastain, S. G., Kohfeld, K. E., Pellatt, M. G., Olid, C., and Gailis, M.: Quantification of Blue Carbon in Salt Marshes of the Pacific Coast of Canada, Biogeosciences Discuss. [preprint], https://doi.org/10.5194/bg-2021-157, in review, 2021.**

**References**

Abdul-Aziz, O. I., Ishtiaq, K. S., Tang, J., Moseman-Valtierra, S., Kroeger, K. D., Gonneea, M. E., et al. (2018). Environmental controls, emergent scaling, and predictions of greenhouse gas (GHG) fluxes in coastal salt marshes. Journal of Geophysical Research: Biogeosciences, 123, 2234– 2256. https://doi-org.proxy.lib.sfu.ca/10.1029/2018JG004556

Brown, D. R. (2021), Coastal wetland soil carbon sequestration revealed from sediment core profiles, Ph.D. thesis, Southern Cross University, DOI:https://doi.org/10.25918/thesis.167

Callaway, John & Borgnis, Evyan & Turner, Robert & Milan, Charles. (2012). Carbon Sequestration and Sediment Accretion in San Francisco Bay Tidal Wetlands. Estuaries and Coasts. 35. 10.1007/s12237-012-9508-9.

Craft, C. & Seneca, E. & Broome, Stephen. (1991). Loss on Ignition and Kjeldahl Digestion for Estimating Organic Carbon and Total Nitrogen in Estuarine Marsh Soils: Calibration with Dry Combustion. Estuaries. 14. 175-179. 10.2307/1351691.

Deur, D.: A Domesticated Landscape: Native American plant cultivation on the Northwest coast of North America, PhD dissertation, Louisiana State University, Baton Rouge, LA, USA, pp. 69-251, 2000.

Goñi, M.A., Thomas, K.A. Sources and transformations of organic matter in surface soils and sediments from a tidal estuary (North Inlet, South Carolina, USA). Estuaries **23,** 548–564 (2000). https://doi-org.proxy.lib.sfu.ca/10.2307/1353145

Huang J, Luo M, Liu Y, Zhang Y, Tan J. Effects of Tidal Scenarios on the Methane Emission Dynamics in the Subtropical Tidal Marshes of the Min River Estuary in Southeast China. Int J Environ Res Public Health. 2019;16(15):2790. Published 2019 Aug 5. doi:10.3390/ijerph16152790

Huertas, I. E., de la Paz, M., Perez, F. F., Navarro, G., and Flecha, S.: Methane Emissions From the Salt Marshes of Doñana Wetlands: Spatio-Temporal Variability and Controlling Factors, Frontiers in Ecology and Evolution, 7, 2019.

James, TS, JA Henton, LJ Leonard, A Darlington, DL Forbes, M Craymer (2014), Relative sea-level projections in Canada and the adjacent mainland United States, Geological Survey of Canada Open File 7737, 72 p. doi: 10.4095/295574

Kurz W.A., ShawC.H., BoisvenueC., StinsonG., MetsarantaJ., LeckieD., DykA., SmythC., and NeilsonE.T.. Carbon in Canada's boreal forest — A synthesis. Environmental Reviews. **21**(4): 260-292. https://doi.org/10.1139/er-2013-0041

**RESPONSE TO REVIEWER COMMENTS: Chastain, S. G., Kohfeld, K. E., Pellatt, M. G., Olid, C., and Gailis, M.: Quantification of Blue Carbon in Salt Marshes of the Pacific Coast of Canada, Biogeosciences Discuss. [preprint], https://doi.org/10.5194/bg-2021-157, in review, 2021.**

*Li Y, Wang D, Chen Z, Chen J, Hu H, Wang R. Methane Emissions during the Tide Cycle of a Yangtze Estuary Salt Marsh. Atmosphere. 2021; 12(2):245. https://doi.org/10.3390/atmos12020245*

*MacKenzie, W.H. and J.R. Moran. 2004. Wetlands of British Columbia: a guide to identification. Res. Br., B.C. Minisitry of Forests, Victoria, B.C. Land Management Handbook No. 52*

*Natural Resources Canada,The State of Canada's Forests. Annual Report 2020. 2020, Canadian Forest Service, Ottawa. 88 p. https://www.nrcan.gc.ca/our-natural-resources/forests-forestry/state-canadas-forests-report/16496*

*Ouyang, X. and Lee, S. Y.: Updated estimates of carbon accumulation rates in coastal marsh sediments, Biogeosciences, 11, 5057–5071, https://doi.org/10.5194/bg-11-5057-2014, 2014.*

*Santín, C., González-Pérez, M., Otero, X. L., Vidal-Torrado, P., Macías, F., & Álvarez, M. Á. (2008). Characterization of humic substances in salt marsh soils under sea rush (Juncus maritimus). Estuarine, Coastal and Shelf Science, 79(3), 541-548. doi:https://doi.org/10.1016/j.ecss.2008.05.007*

*Suir et al. (2019), Comparing carbon accumulation in restored and natural wetland soils of coastal Louisiana, International Journal of Sediment Research, 34 (2019): 600-607.*

*van Ardenne, L. B., Hughes, J. F., & Chmura, G. L. (2021). Tidal marsh sediment and carbon accretion on a geomorphologically dynamic coastline. Journal of Geophysical Research: Biogeosciences, 126, e2021JG006507. https://doi.org/10.1029/2021JG006507*

*Wang, F., Sanders, C.J., Santos, I.R., Tang, J., Schuerch, M., Kirwan, M.L., Kopp, R.E., Zhu, K., Li, X., Yuan, J. and Liu, W., 2021. Global blue carbon accumulation in tidal wetlands increases with climate change. National science review, 8(9), 296.*

*Wei, S., Han, G., Chu, X., Song, W., He, W., Xia, J., and Wu, H.: Effect of tidal flooding on ecosystem CO2 and CH4 fluxes in a salt marsh in the Yellow River Delta, Estuarine, Coastal and Shelf Science, 232, 106512, 2020.*

*Zhao, B., Zhuang, Q., Shurpali, N. et al. North American boreal forests are a large carbon source due to wildfires from 1986 to 2016. Sci Rep **11**, 7723 (2021). https://doi.org/10.1038/s41598-021-87343-3*

---

## Referee Report (RR1)

Chastain, S. G., Kohfeld, K. E., Pellatt, M. G., Olid, C., and Gailis, M.: Quantification of Blue Carbon in Salt Marshes of the Pacific Coast of Canada, Biogeosciences Discuss. [preprint], https://doi.org/10.5194/bg-2021-157, in review, 2021.

General comments on review of revised manuscript:

Authors' responses do not seem to reflect an understanding of the reviewer comments regarding geomorphology, proper coring methods and acceptable compaction levels, statistical analyses and hydroperiods relative to tidal ranges (thus methane emissions).  Unfortunately, the responses and text revisions are not appropriate and I cannot recommend accepting the manuscript in its present form.  There are 5 major "revisions" or "responses" that are unacceptable.  Below I repeat my original comment, the authors' response and my new comments on the authors' responses and text revisions, the latter identified by ALL CAPS.

**1.  Original Reviewer Comment**
**This work on the British Columbia coast could even further advance blue carbon science by providing details on the geomorphic context of each marsh. There is nascent research showing that the C stock of marshes is related to their geomorphic context (see van Ardenne, Jolicouer, Bérubé, Burdick, Chmura. 2018. The Importance of Geomorphic Context for Estimating the Carbon Stock of Salt Marshes. *Geoderma* 330:264-275). It would be useful to know if it plays a role in these British Columbia marshes, e.g., behind spits, on lagoons, fluvial marshes (as per Kelley JT, Gehrels WR, Belknap, DF, 1995. Late Holocene relative sea—level rise and the geological development of Tidal Marshes at Wells, Maine, U.S.A. *J. Coast. Res*. 11, 136–153.) or at least be available for future meta-analyses.**

*A direct comparison with the geomorphic contexts in van Ardenne et al. 2018 is somewhat challenging because the terrain around our study area does not involve formation of spits and lagoons. Many of our sites were enclosed bays but they were not really cut off by spits. All locations were somewhat close to fluvial sources of varying size. Thus, applying the exact categories of van Ardenne et al. (2018) could be somewhat contrived here.*

*We do note that recent work by van Ardenne et al. (2021) has examined this question – albeit in fresher marsh systems - on the central BC coast. They argue that relating carbon density and marsh depth to geomorphology is difficult on a geomorphologically dynamic coastline as is found in our study area. We suggest that this might be an interesting topic to revisit in future.*

*We have added a comment about geomorphology here (ln 106-108):*
*"These sites are typical of salt marshes along Canada's Pacific coast because they include small, pocket marshes encompassing an enclosed, semi-circular area of coastline as well as larger, estuarine marshes. Unlike geomorphological settings in Atlantic Canada (e.g. van Ardenne et al. 2018), we do not see extensive formation of spits and lagoons; many of our sites were in enclosed bays but were not cut off by spits. All sites were somewhat close to fluvial sources of varying size. Surface water salinity in the surrounding waters ranged from 5.9 at KCS to 24 in Grice Bay, and 29 at Roberts Point six km south of CRF (Postlethwaite et al. 2018)."*

**REVIEWER COMMENT ON AUTHOR STATEMENT AND REVISED TEXT**
NEW TEXT ON LINE 106-108 NEEDS CORRECTION.  REVIEWER COMMENTS DID NOT REQUEST A DIRECT COMPARISON WITH VAN ARDENNE ET AL. BUT AS AN EXAMPLE OF HOW TO PUT THE BC SITE IN THE CONTEXT OF THE GEOMORPHOLOGICAL CLASSIFICATION OF KELLEY ET AL.  THUS, THE COMMENTS

ABOUT SPITS, ETC. ARE INAPPROPRIATE AND SHOULD BE DELETED. PROPER GEOMORPHOLOGICAL TERMINOLOGY IS REQUIRED HERE, RATHER THAN TERMS NOW ADDED SUCH AS "SEMI-CIRCULAR AREA OF COASTLINE" AND "ESTUARINE MARSHES". I SUSPECT THAT THE SEMI-CIRCULAR AREAS OF COASTLINES ARE BAYS - WHICH FALL UNDER THE DEFINITION OF "ESTUARY".

**2. Reviewer initial comment**
**Some cores had high levels of compaction, due to use of percussion corers. (This type of coring should be the last choice when working in wetland soils as there are other devices that can be used which produce negligible or no compaction. For instance, authors do not mention trying a narrow diameter Dutch gouge corer, which often saves the day – or simply shoveling out a block and coring through the excavated material.) Although the compaction not a problem when calculating stocks to the base of the marsh deposit, it can affect bulk densities, thus carbon densities and the calculation of accumulation rates (one of the dated cores had 41% compaction). At line 200, the text states, "Here we estimated the accumulated C to the corrected (uncompacted) depth of 20 cm". Use of lead-210 inventories and 30 yr stocks help to address the complication of compaction, but authors should note how compaction was corrected for and how bulk densities were adjusted – this is very important and should be in the methodology. I assume that there was a threshold for compaction level beyond which cores were not used for calculation of bulk or carbon density and certainly lead-210.**
*We have added an explanation in the Methods section to explain why we sampled with a percussion corer, in which we quantify the effects of compaction on our sediment cores. We also provide a brief explanation for how we have accounted for compaction (ln 157-168):*
*"Use of the percussion corer resulted in sediment compaction during sample collection, which averaged about 20% across all cores (range 0-55%) (Table A1). Nevertheless, we opted to use a percussion corer instead of a gouge corer because the percussion corer had a closed chamber with internal PVC sleeves. Our experience with this sedimentary has demonstrated that a gouge corer would have been susceptible to disturbance and sediment mixing due to the nature of the open chamber of the corer. Because the nature of the marsh sediments, we also did not use a Russian corer because compaction would have been similar to what we experienced with the percussion corer, and we did not want to introduce increased contamination through the pivoting nature of the sampling chamber with the Russian corer. Digging pits with a shovel was not an option as this study took place in a national park and biosphere reserve. We note below that correction for compaction was not necessary for estimation of C stocks because the C stocks were estimated directly from sediment cores and not from the overall depth of marsh soils (thus all carbon in the peat layer, regardless of compression, is included in the calculation).*

**REVIEWER COMMENT ON AUTHOR STATEMENT AND REVISED TEXT**
THE NEXT TEXT ON LN 157-168 (WHICH DOES NOT SEEM TO BE RECOGNIZED AS AN ADDITION IN TRACK CHANGES) IS INAPPROPRIATE AND MUST BE DELETED AS IT WOULD BE EXTREMELY MISLEADING TO ANY READERS WITHOUT THEIR OWN CORING EXPERIENCE. FIRST, A RUSSIAN CORER DOES NOT COMPACT SEDIMENT AS IT CUTS IT FROM THE SIDE AND DOES NOT RESULT IN CONTAMINATION ACROSS DEPTHS! HOWEVER, IT IS IMPRACTICAL IN SOME WETLANDS THAT HAVE DENSE MINERAL SOIL. THOSE OF US WITH DECADES OF EXPERIENCE CORING A RANGE OF MARSH

SEDIMENT TYPES KNOW THAT ANY DISTURBANCE AND SEDIMENT MIXING USING A GOUGE CORER WOULD BE MINIMAL.  (PERHAPS THE AUTHORS HAVE NEVER USED A GOUGE CORER?)  THIS REVIEWER HAS USED A GOUGE CORER IN BC SALT MARSHES AND FOUND THAT IT CAN BE VERY EFFECTIVE, PARTICULARLY IN THE SHALLOW DEPOSITS FOUND IN THIS STUDY. TO SAMPLE WITH A SHOVEL (SPADE IS BEST) REQUIRES A *HOLE*, *NOT A PIT* AND ONCE THE BLOCK OF SEDIMENT IS CORED THROUGH, THEN THE SURROUNDING MATERIAL CAN EASILY BE PLACED BACK IN THE HOLE, SOMETHING THAT THIS REVIEWER HAS FOUND TO BE A SUCCESSUL APPROACH.

NOTE THAT THESE SAME SUGGESTIONS FOR USE OF CORERS FOR QUESTIONS OF CARBON ACCUMULATION ARE FOUND IN

*1) COASTAL BLUE CARBON: METHODS FOR ASSESSING CARBON STOCKS AND EMISSIONS FACTORS PUBLISHED BY THE BLUE CARBON INITIATIVE AND FREELY AVAILABLE ONLINE (*[http://thebluecarboninitiative.org/manual/](http://thebluecarboninitiative.org/manual/)*)*

*AND BY*

*2)* SMEATON C, BARLOW NLM, AUSTIN WEN. 2020. CORING AND COMPACTION: BEST PRACTICE IN BLUE CARBON STOCK AND BURIAL ESTIMATIONS. *GEODERMA* 364

SMEATON ET AL NOTE: "A COMPARISON OF GOUGE AND HAMMER CORING TECHNIQUES IN INTERTIDAL WETLAND SOILS HIGHLIGHTS A SIGNIFICANT EFFECT OF SOIL COMPACTION OF UP TO 28% ASSOCIATED WITH THE WIDELY APPLIED HAMMER CORING METHOD EMPLOYED IN BLUE CARBON RESEARCH. …….  WE SHOW THAT HAMMER CORING IS UNSUITABLE FOR THE CALCULATION OF OC STOCKS AND SHOULD BE AVOIDED IN FAVOUR OF RUSSIAN OR GOUGE CORES. COMPACTION CHANGES BOTH SOIL DRY BULK DENSITY AND POROSITY AND WE SHOW THAT RESULTANT RADIOMETRIC CHRONOLOGIES ARE COMPROMISED, ALMOST DOUBLING MASS ACCUMULATION RATES. WHILE WE SHOW THAT THE OC (%) CONTENT OF THESE SEDIMENTS IS LARGELY UNCHANGED BY CORING METHOD, THE IMPLICATION FOR OC BURIAL RATES ARE PROFOUND BECAUSE OF THE SIGNIFICANT EFFECT OF HAMMER CORING ON THE CALCULATION OF SOIL MASS ACCUMLATION RATES."

THUS, AUTHORS MUST NOT REPORT ACCUMULATION RATES IN THOSE CORES THAT SUFFERED EXCESSIVE COMPACTION – THOSE COMPACTED GREATER THAN 20% MUST NOT BE USED FOR ACCUMULATION RATES.

*AUTHOR RESPONSE CONTINUES*

*However, when we do need to account for compaction (e.g. Figures 2-3), we use a compaction factor (Howard et al. 2014; Gailis et al. 2021) estimated for each core by dividing the length of core penetration by the length of core recovered (Table A1)."*

**REVIEWER COMMENT ON AUTHOR STATEMENT AND REVISED TEXT**
THE CITED PAPER BY HOWARD ET AL 2014 MAKES NO MENTION OF COMPACTION AND GAILIS ET AL IS NEITHER A METHODS NOR A REVIEW PAPER, THUS NEITHER CITATION IS AN APPROPRIATE SUPPORTING REFERENCE

**3. Reviewer initial comment**
**Shouldn't the regression for the relationship of %LOI and %C be forced through zero? With a negative intercept a sample with no organic matter, thus 0% LOI would have a negative amount of carbon – an impossibility.**

*Thank you for pointing this out. The relationship between %C and %LOI suggests that we measure zero %C in samples where LOI is not completely zero (below approximately 10% LOI). Although negative values of %C are obviously not possible, forcing the equation through zero would overestimate %C in these low LOI samples. Therefore, all calculations producing a negative value for %C were adjusted to zero %C. This occurred in 41 of 835 samples measured. Our methods have been clarified to reflect this change* using the following equation, setting any negative %C value resulting from the use of a negative intercept equal to zero

**REVIEWER COMMENT ON AUTHOR STATEMENT AND REVISED TEXT**
AUTHORS DO NOT SEEM TO UNDERSTAND THE COMMENT – FORCING THE LINEAR REGRESSION THROUGH ZERO DOES NOT MEAN SIMPLY DROPPING THE INTERCEPT AFTER OBTAINING THE REGRESSION MODEL. WHEN RUNNING THE REGRESSION ONE SIMPLY CHOOSES NO INTERCEPT FOR THE MODEL, THUS A NEW REGRESSION EQUATION IS REQUIRED. DOING THIS WOULD SIMPLY MEAN THAT 0% LOI IS EQUIVALENT TO 0% C. AUTHORS HAVE NO REASON TO CONCLUDE THAT FORCING THE REGRESSION THROUGH ZERO WOULD OVERESTIMATE THE %C.

**4. Reviewer initial comment**
**Clarification of and distinction amongst the terms "topsoil", "humus" and "peat" is needed. What is "topsoil" in a marsh? This term is not commonly used for wetland soils. The manuscript states see "Supplemental Information", but there is no explanation there. Also, the term "humus" is seldom used in wetland soils. Presumably it plant litter that is gradually broken down with depth? A bit of explanation would be helpful, even if just in a footnote to the Appendix table.**
*We take this point and have changed the term "topsoil" (which was used to describe the fibrous organic material within and below the root zone) as "peat." However, we have kept use of the term "humus" as term that has been used as a descriptor in other salt marsh publications (e.g. Goni and Thomas, 2000; Santin et al. 2008)*

**REVIEWER COMMENT ON AUTHOR STATEMENT AND REVISED TEXT**
AUTHORS ACTUALLY DELETED HUMUS FROM THE TEXT, BUT NOT THE TABLES. AUTHORS WILL NEED TO FIND AN ALTERNATIVE TERM FOR HUMUS – ONE THAT IS WIDELY USED BY THOSE WORKING WITH SALT MARSH SOILS, NEITHER PAPER CITED SUPPORTS THEIR USE OF HUMUS. SANTIN ET AL NEVER USE THE TERM HUMUS – THEIR STUDY IS ABOUT HUMIC ACIDS. GONI AND THOMAS SIMPLY USED THE TERM HUMUS TO IDENTIFY A PARTICULAR SIZE FRACTION OF ORGANIC MATTER, NOT AN ENTIRE PORTION OF THE SOIL.

5. **Reviewer initial comment**
**Line 518- Why would tidal amplitude be a driver of methane emissions? The paper cited on this line (Poffenbarger et al. 2011) reports that salinity, as a proxy for marine sulfates, is an important correlate.**
*We appreciate this comment and we can replace the Poffenbarger et al. (2011) publication in this context, as there are several better citations that have measured changes in methane emissions associated with tidal activity and sea level rise on LN 608: (e.g. Abdul-Aziz et al. 2018; Huang et al. 2019; Huertas et al. 2019; Li et al. 2021; Wei et al. 2020.).*

**REVIEWER COMMENT ON AUTHOR STATEMENT AND REVISED TEXT**
THE TEXT RELATED TO METHANE SHOULD BE DELTED ENTIRELY.

AUTHORS WRITE "THE MESOTIDAL NATURE OF SOME OF THESE MARSH  LOCATIONS COULD MEAN THAT SOME OF THESE MARSHES EMIT SUBSTANTIAL CONTRIBUTIONS OF METHANE, WHICH MAY COUNTER THEIR EFFECTS AS C SINKS (E.G. ABDUL-AZIZ ET AL. 2018; HUANG ET AL. 2019; HUERTAS ET AL. 2019; LI ET AL. 2021; WEI ET AL. 2020)."  THESE PAPERS ARE ABOUT DURATION OF FLOODING – GREATER TIDAL RANGES MEAN LOWER HYDROPERIODS, NOT LONGER.  THUS ONE WOULD EXPECT LESS METHANE PRODUCTION AND CERTAINLY OXIDATION OF THE METHANE THAT IS PRODUCED.  NOTE THAT CHMURA FOUND NEGLIBLE METHANE EMISSIONS IN A MACROTIDAL MARSH.

NOTE SOME OF THE REFERENCES ARE IDENTIFIED BY URLS ONLY ACCESSIBLE TO THE SIMONE FRASER UNIVERSITY SYSTEM!

---

## Author Response (AR2)

**Chastain, S. G., Kohfeld, K. E., Pellatt, M. G., Olid, C., and Gailis, M.: Quantification of Blue Carbon in Salt Marshes of the Pacific Coast of Canada, Biogeosciences Discuss. [preprint], https://doi.org/10.5194/bg- 2021-157, in review, 2021.**

*We thank Dr. Chmura for her review of this manuscript. Dr. Chmura outlined 5 major revisions for us to address. Below we detail our responses to her comments. In our response we:*

1. *Attempt to find terminology that both accurately reflects definitions of "bay" and "estuary" while also being suitable to the reviewer,*
2. *Provide a detailed description (with photographs) as to why the reviewer's choice of coring device is not suitable for our study area,*
3. *Refute the reviewer's claim that our compacted cores are unsuitable for estimations of 210Pb accumulation rates, describe the flaws in the Smeaton et al. (2020) arguments, and demonstrate via a modeling exercise how accumulation rates can be properly estimated in terms of cumulative mass instead of depth (which is what we have done).*
4. *Explain why there is no processed-based reason why the empirically derived %C-%LOI relationship needs to have a zero intercept, and provide multiple examples from the literature where similar, regional relationships with non-zero intercepts have been used.*
5. *Replace the term 'humus' with 'soil,' which is further described in table and figure notes*
6. *Replace the term 'mesotidal' with 'brackish' to circumvent the discussion around tidal height and duration and focus on future work needed to assess the impact of methane emissions from brackish marshes on the total carbon budget.*

*We thank the editor for the opportunity to elucidate these points and hope that these responses are sufficient for publication.*

- *Original reviewer comments are **bold grey**; our original responses are italicized grey.*
- *New reviewer comments are **bold red**; our response to new comments is italicized red.*
- *We provide the location of changes using the line numbers from the 'track changes' version of our resubmitted manuscript, with change and associated line numbers ==highlighted in yellow==.*

*Sincerely,*
*The authors*
* * *
**General comments on review of revised manuscript:**

**Authors' responses do not seem to reflect an understanding of the reviewer comments regarding geomorphology, proper coring methods and acceptable compaction levels, statistical analyses and hydroperiods relative to tidal ranges (thus methane emissions). Unfortunately, the responses and text revisions are not appropriate and I cannot recommend accepting the manuscript in its present form. There are 5 major "revisions" or "responses" that are unacceptable. Below I repeat my original comment, the authors' response and my new comments on the authors' responses and text revisions, the latter identified by ALL CAPS.**
* * *
1. **Original Reviewer Comment**
   **This work on the British Columbia coast could even further advance blue carbon science by providing details on the geomorphic context of each marsh. There is nascent research showing that the C stock of marshes is related to their geomorphic context (see van Ardenne, Jolicouer, Bérubé, Burdick, Chmura. 2018. The Importance of Geomorphic Context for Estimating the Carbon Stock of Salt Marshes. *Geoderma* 330:264-275). It would be useful to know if it plays a**

**role in these British Columbia marshes, e.g., behind spits, on lagoons, fluvial marshes (as per Kelley JT, Gehrels WR, Belknap, DF, 1995. Late Holocene relative sea—level rise and the geological development of Tidal Marshes at Wells, Maine, U.S.A. *J. Coast. Res*. 11, 136–153.) or at least be available for future meta-analyses.**

*A direct comparison with the geomorphic contexts in van Ardenne et al. 2018 is somewhat challenging because the terrain around our study area does not involve formation of spits and lagoons. Many of our sites were enclosed bays but they were not really cut off by spits. All locations were somewhat close to fluvial sources of varying size. Thus, applying the exact categories of van Ardenne et al. (2018) could be somewhat contrived here. We do note that recent work by van Ardenne et al. (2021) has examined this question – albeit in fresher marsh systems - on the central BC coast. They argue that relating carbon density and marsh depth to geomorphology is difficult on a geomorphologically dynamic coastline as is found in our study area. We suggest that this might be an interesting topic to revisit in future.*

*We have added a comment about geomorphology here (ln 106-108):*
*"These sites are typical of salt marshes along Canada's Pacific coast because they include small, pocket marshes encompassing an enclosed, semi-circular area of coastline as well as larger, estuarine marshes. Unlike geomorphological settings in Atlantic Canada (e.g. van Ardenne et al. 2018), we do not see extensive formation of spits and lagoons; many of our sites were in enclosed bays but were not cut off by spits. All sites were somewhat close to fluvial sources of varying size. Surface water salinity in the surrounding waters ranged from 5.9 at KCS to 24 in Grice Bay, and 29 at Roberts Point six km south of CRF (Postlethwaite et al. 2018)."*

**REVIEWER COMMENT ON AUTHOR STATEMENT AND REVISED TEXT**
**NEW TEXT ON LINE 106-108 NEEDS CORRECTION. REVIEWER COMMENTS DID NOT REQUEST A DIRECT COMPARISON WITH VAN ARDENNE ET AL. BUT AS AN EXAMPLE OF HOW TO PUT THE BC SITE IN THE CONTEXT OF THE GEOMORPHOLOGICAL CLASSIFICATION OF KELLEY ET AL. THUS, THE COMMENTS ABOUT SPITS, ETC. ARE INAPPROPRIATE AND SHOULD BE DELETED. PROPER GEOMORPHOLOGICAL TERMINOLOGY IS REQUIRED HERE, RATHER THAN TERMS NOW ADDED SUCH AS "SEMI-CIRCULAR AREA OF COASTLINE" AND "ESTUARINE MARSHES". I SUSPECT THAT THE SEMI-CIRCULAR AREAS OF COASTLINES ARE BAYS - WHICH FALL UNDER THE DEFINITION OF "ESTUARY".**

*Author Response:*
*We appreciate this clarification, but unfortunately the request from the reviewer remains unclear. We examined the Kelley et al. (1995) paper, but it is neither a review paper nor a methodology paper: its purpose was to discuss the specific evolution of back barrier marshes along the Webhannet and Little Rivers of Wells, Maine USA. The paper itself does not describe a list of geomorphological terminology applicable to the characteristics of the salt marshes examined on the west coast of Vancouver Island. In our previous revision, we therefore attempted to add a comparison with the Van Ardenne et al. (2018) paper because of its focus on geomorphological classification and its relationship with carbon dynamics. We are happy to remove this comparison.*

*We are also puzzled by the reviewer's definition of an estuary versus a bay.  Essentially, an estuary is a partially enclosed coastal body of brackish water with one or more rivers or streams flowing into it, and with a free connection to the open sea (Pritchard, 1967).  In contrast, a bay is simply a depression marked by a penetration whereby land-locked waters are contained by the proportion of the width of its mouth (United Nations Convention on the Law of the Sea). Therefore, an estuary can exist within in a bay, but a bay is not necessarily an estuary.*

*With this said, we have changed the section to state (line 103):*
*"These sites are typical of salt marshes along Canada's Pacific coast because they include small marshes along protected shorelines and bays as well as larger estuarine marshes near creeks and rivers."*
* * *
2. **Reviewer initial comment**
**Some cores had high levels of compaction, due to use of percussion corers. (This type of coring should be the last choice when working in wetland soils as there are other devices that can be used which produce negligible or no compaction. For instance, authors do not mention trying a narrow diameter Dutch gouge corer, which often saves the day – or simply shoveling out a block and coring through the excavated material.) Although the compaction not a problem when calculating stocks to the base of the marsh deposit, it can affect bulk densities, thus carbon densities and the calculation of accumulation rates (one of the dated cores had 41% compaction). At line 200, the text states, "Here we estimated the accumulated C to the corrected (uncompacted) depth of 20 cm". Use of lead-210 inventories and 30 yr stocks help to address the complication of compaction, but authors should note how compaction was corrected for and how bulk densities were adjusted – this is very important and should be in the methodology. I assume that there was a threshold for compaction level beyond which cores were not used for calculation of bulk or carbon density and certainly lead-210.**
*We have added an explanation in the Methods section to explain why we sampled with a percussion corer, in which we quantify the effects of compaction on our sediment cores. We also provide a brief explanation for how we have accounted for compaction (ln 157-168):*
*"Use of the percussion corer resulted in sediment compaction during sample collection, which averaged about 20% across all cores (range 0-55%) (Table A1). Nevertheless, we opted to use a percussion corer instead of a gouge corer because the percussion corer had a closed chamber with internal PVC sleeves. Our experience with this sedimentary has demonstrated that a gouge corer would have been susceptible to disturbance and sediment mixing due to the nature of the open chamber of the corer. Because the nature of the marsh sediments, we also did not use a Russian corer because compaction would have been similar to what we experienced with the percussion corer, and we did not want to introduce increased contamination through the pivoting nature of the sampling chamber with the Russian corer. Digging pits with a shovel was not an option as this study took place in a national park and biosphere reserve. We note below that correction for compaction was not necessary for estimation of C stocks because the C stocks were estimated directly from sediment cores and not from the overall depth of marsh soils (thus all carbon in the peat layer, regardless of compression, is included in the calculation).*

REVIEWER COMMENT ON AUTHOR STATEMENT AND REVISED TEXT
THE NEXT TEXT ON LN 157-168 (WHICH DOES NOT SEEM TO BE RECOGNIZED AS AN ADDITION IN TRACK CHANGES) IS INAPPROPRIATE AND MUST BE DELETED AS IT WOULD BE EXTREMELY MISLEADING TO ANY READERS WITHOUT THEIR OWN CORING EXPERIENCE. FIRST, A RUSSIAN CORER DOES NOT COMPACT SEDIMENT AS IT CUTS IT FROM THE SIDE AND DOES NOT RESULT IN CONTAMINATION ACROSS DEPTHS! HOWEVER, IT IS IMPRACTICAL IN SOME WETLANDS THAT HAVE DENSE MINERAL SOIL. THOSE OF US WITH DECADES OF EXPERIENCE CORING A RANGE OF MARSH SEDIMENT TYPES KNOW THAT ANY DISTURBANCE AND SEDIMENT MIXING USING A GOUGE CORER WOULD BE MINIMAL. (PERHAPS THE AUTHORS HAVE NEVER USED A GOUGE CORER?) THIS REVIEWER HAS USED A GOUGE CORER IN BC SALT MARSHES AND FOUND THAT IT CAN BE VERY EFFECTIVE, PARTICULARLY IN THE SHALLOW DEPOSITS FOUND IN THIS STUDY. TO SAMPLE WITH A SHOVEL (SPADE IS BEST) REQUIRES A *HOLE, NOT A PIT* AND ONCE THE BLOCK OF SEDIMENT IS CORED THROUGH, THEN THE SURROUNDING MATERIAL CAN EASILY BE PLACED BACK IN THE HOLE, SOMETHING THAT THIS REVIEWER HAS FOUND TO BE A SUCCESSUL APPROACH. NOTE THAT THESE SAME SUGGESTIONS FOR USE OF CORERS FOR QUESTIONS OF CARBON ACCUMULATION ARE FOUND IN

*1) COASTAL BLUE CARBON: METHODS FOR ASSESSING CARBON STOCKS AND EMISSIONS FACTORS PUBLISHED BY THE BLUE CARBON INITIATIVE AND FREELY AVAILABLE ONLINE (*http://thebluecarboninitiative.org/manual/*)*

*AND BY*

*2) SMEATON C, BARLOW NLM, AUSTIN WEN. 2020. CORING AND COMPACTION: BEST PRACTICE IN BLUE CARBON STOCK AND BURIAL ESTIMATIONS. GEODERMA 364*

SMEATON ET AL NOTE: "A COMPARISON OF GOUGE AND HAMMER CORING TECHNIQUES IN INTERTIDAL WETLAND SOILS HIGHLIGHTS A SIGNIFICANT EFFECT OF SOIL COMPACTION OF UP TO 28% ASSOCIATED WITH THE WIDELY APPLIED HAMMER CORING METHOD EMPLOYED IN BLUE CARBON RESEARCH. …….
WE SHOW THAT HAMMER CORING IS UNSUITABLE FOR THE CALCULATION OF OC STOCKS AND SHOULD BE AVOIDED IN FAVOUR OF RUSSIAN OR GOUGE CORES. COMPACTION CHANGES BOTH SOIL DRY BULK DENSITY AND POROSITY AND WE SHOW THAT RESULTANT RADIOMETRIC CHRONOLOGIES ARE COMPROMISED, ALMOST DOUBLING MASS ACCUMULATION RATES. WHILE WE SHOW THAT THE OC (%) CONTENT OF THESE SEDIMENTS IS LARGELY UNCHANGED BY CORING METHOD, THE IMPLICATION FOR OC BURIAL RATES ARE PROFOUND BECAUSE OF THE SIGNIFICANT EFFECT OF HAMMER CORING ON THE CALCULATION OF SOIL MASS ACCUMLATION RATES."

THUS, AUTHORS MUST NOT REPORT ACCUMULATION RATES IN THOSE CORES THAT SUFFERED EXCESSIVE COMPACTION – THOSE COMPACTED GREATER THAN 20% MUST NOT BE USED FOR

**ACCUMULATION RATES.**

_AUTHOR RESPONSE CONTINUES: However, when we do need to account for compaction (e.g. Figures 2-3), we use a compaction factor (Howard et al. 2014; Gailis et al. 2021) estimated for each core by dividing the length of core penetration by the length of core recovered (Table A1)."_

**REVIEWER COMMENT ON AUTHOR STATEMENT AND REVISED TEXT**
**THE CITED PAPER BY HOWARD ET AL 2014 MAKES NO MENTION OF COMPACTION AND GAILIS ET AL IS NEITHER A METHODS NOR A REVIEW PAPER, THUS NEITHER CITATION IS AN APPROPRIATE SUPPORTING REFERENCE**

_Author Response_
_The reviewer brings up two points here: (1) the effect of coring device on compaction, and (2) the potential impact of compaction on calculated accumulation rates. We address these two points separately below._

_POINT 1. We appreciate that the reviewer has decades of experience coring marsh sediments, but so do some of the authors of this manuscript. We absolutely agree that there are issues with percussion coring, but we also are well aware of the issues with gouge corers and Russian corers. These corers are not suitable for our study site as we found the compression and disturbance from a Gouge corer and Russian corer to be unacceptable. We have included some photographs (Figures 2.1-2.3) showing the compaction (~50% caused by a gouge corer, and although hard to see, the inability to rotate the Russian corer in the mix of sand and sticky sediment at the sites. Note as for using a shovel, we have stated that this was not permissible in a National Park Reserve._

[Figure]

*Fig 2.1. Gouge corer in Grice Bay marsh sediments.*

[Figure]

*Figure 2.2. Note compression of sediments with the gouge core. Black line is depth of core.   Note sediment is compresses over 50%.*

[Figure]

*Figure 2.3. We were unable to recover a sediment core with the Russian Corer due to disturbance on resistance of sediment.*

*Note that, according to Frew (2014), Gouge augers are easily transportable tools that permit relatively quick survey of subsurface sediments in terrestrial environments. Sampling is particularly rudimentary and involves thrusting a semi cylindrical chamber into deposits and twisting the device using a handle at the surface to capture the sample. Consecutive drives are enabled by the addition of extension rods.  The retrieved sample is subject to significant disturbance as the open chamber is prone to resampling of material from depths above those required, especially where sands underlie the softer organic material above. Additionally, more-consolidated material can force its way upwards over less consolidated horizons within the chamber. For these reasons, it is not recommended that the gouge auger be used to retrieve samples for analysis. (Frew, 2014).*

*As for Russian corers, De Vleeschouwer et al. (2010) state "The Russian corer is not designed to cut the living plant mat cleanly and will strongly compress the core." This is particularly problematic in sandy/gravelly sediments where researchers have identified: "The Russian corer is used to core terrestrial and wetland soft sediments; clay, gyttja, or peat, but cannot be used to core in sand or other coarse-grained sediments."  https://corerepository.ldeo.columbia.edu/content/types-samples.*

*As such, we have left the text on lines 153-158 unchanged.*

*POINT 2. Regarding the use of compressed cores to determine carbon accumulation rates: as we mention in our manuscript, compaction is not a problem when calculating stocks to the base of the marsh deposit. However, measurements of the stocks at a fixed depth in compacted cores lead to a misleading impression of salt marshes' organic carbon sequestration efficiency due to varying densities and accumulation rates. We added a short description about the advantages of using stocks accumulated at a common age-horizon instead. Contrary to what the reviewer states in her comments, compaction does not affect the calculation of the $^{210}$Pb-derived accumulation rates when they are estimated in terms of cumulative mass (g cm$^{-2}$) and not depth (cm). In the text below, we discuss why the paper Smeaton et al. (Geoderma 2020) cited by the reviewer is fundamentally wrong when claiming that $^{210}$Pb dating models do not work in compacted cores. We also simulate compaction in an undisturbed core to evaluate the potential differences in the $^{210}$Pb derived chronology.*

*Smeaton et al. (2020) claim that compaction affects the calculation of mass accumulation rate (MAR) (g cm$^{-2}$ yr$^{-1}$) when applying the conventional $^{210}$Pb dating models. To prove this, the authors combine a published $^{210}$Pb profile from a saltmarsh (Barlow et al., 2014) with density data from two new cores collected using gouge and hammer cores. The densities of these new cores are higher than the original core due to compaction during sampling. Although the objective of the paper is to evaluate the applicability of the $^{210}$Pb dating models in compacted cores, the paper's methodology is fundamentally flawed because compaction does not only affect the density of the material but also the distribution of a certain element ($^{210}$Pb in this case) along the profile. For that reason, density and porosity profiles and $^{210}$Pb concentrations (Bq/kg) must be corrected to model the effects of compaction on the $^{210}$Pb-derived chronologies correctly. To highlight that systematic error bias occurs in both density and $^{210}$Pb concentrations, we provide a hypothetical example of the compaction of two sections or slices of a saltmarsh core (Figure 2.4). In Figure 2.4, two consecutive core sections (1 cm thick) have been compacted, resulting in one single slice (1 cm) (we keep the slicing at 1 cm thick as is usually done in these types of studies). While the concentration of $^{210}$Pb in intact sections are 10 and 8 Bq/kg, respectively, the resulting section corresponds to 9 Bq/kg (18 Bq of $^{210}$Pb distributed in 2 kg of mass), which shows that the vertical distribution of $^{210}$Pb along the core changes under compaction. Thus, (Smeaton et al., 2020) should have also corrected the $^{210}$Pb concentration profile due to compaction before applying the $^{210}$Pb dating model.*

[Figure]

$$C_1 = \frac{10\ Bq}{1\ kg} = 10\ Bq/kg \qquad I_1 = 10\ Bq/cm2 \qquad C_1' = \frac{18\ Bq}{2\ kg} = 9\ Bq/kg$$

$$C_2 = \frac{8\ Bq}{1\ kg} = 8\ Bq/kg \qquad I_2 = 8\ Bq/cm2 \qquad I' = \frac{10 + 8\ Bq}{1\ cm2} = 18\ Bq/m2$$

**Figure 2.4.** *Effect of compaction on $^{210}Pb$ concentration (Bq/kg) and inventory (Bq/m²).*

*Another piece of evidence showing that (Smeaton et al., 2020) applied the $^{210}Pb$ dating models erroneously is the extremely old ages found in the compacted cores (Figure 4 in paper). Due to the half-life of $^{210}Pb$ (22.3 years) and large uncertainties usually found in the older layers due to the low concentration of $^{210}Pb$, $^{210}Pb$ dating models usually provide accurate chronologies for the past 100-150 years. Thus, the 200-250 years found in (Smeaton et al., 2020) seem to suggest some errors when the $^{210}Pb$ dating models were applied. These extremely old ages can be ascribed to using higher density values for the gouge and hammer cores while keeping invariant the $^{210}Pb$ profile concentrations. Besides providing inconsistent marsh ages, increasing density values without changing $^{210}Pb$ concentrations increases the amount of $^{210}Pb$ accumulated in the core (Inventory, Bq/m²). This is a big mistake, as the flux of $^{210}Pb$ (flux of $^{210}Pb$ (Bq/m2/yr= ln(2)/22.3 years·Inventory (Bq/m2)) in a given area is constant. Thus, all cores must have the same $^{210}Pb$ inventory. We tried to find the raw data that Smeaton et al (2020) used to estimate the different age-depth models, but neither the $^{210}Pb$ concentration profiles nor the density profiles are provided in the paper.*

*To further prove that the chronology derived from the CFCS model (model that we used in our manuscript) is not affected by compaction when depth is represented as cumulative mass (g cm⁻²) and not in cm, we have simulated compaction on an initial undisturbed tidal marsh sediment and evaluated potential deviations in mass accumulation rates (MAR, g cm⁻² yr⁻¹) and chronology. We used the ideal excess $^{210}Pb$ profile of seagrass sediment provided in (Arias-Ortiz et al., 2018). We chose the seagrass profile and not the tidal marsh provided in the review because the length of the tidal marsh was higher than 1 m, which made the calculations more difficult to follow. The ideal $^{210}Pb$ profile was modelled considering the following:*
*1) A constant flux of excess $^{210}Pb$ of 120 Bq m⁻² yr⁻¹.*
*2) A mass accumulation rate of 0.2 g cm⁻² yr⁻¹.*
*3) And a dry bulk density (DBD) of 0.1.03 g cm⁻³.*

*We assumed that the ideal core was subjected to a 50% compaction during sampling, meaning that its original length (30 cm) was reduced to its half (15 cm)(see Appendix for detailed calculations). We assumed that the compaction occurred homogenously along the whole profile and combined*

*two consecutive marsh layers in one. After this, we recalculated the cumulative mass (g cm$^{-2}$), excess $^{210}$Pb inventory (Bq m$^{-2}$) and $^{210}$Pb concentration (Bq kg$^{-1}$) per each layer. Then, the CFCS was applied in both cores (uncompacted and compacted) using cumulative mass (g cm$^{-2}$) instead of depth (cm) (Figure 2.5). Results showed that the CFCS model provided similar chronologies for both the ideal and the compacted core (Figure 2.6), which confirms that compaction does not affect the derived MAR when those are obtained using the cumulative mass profile (Figure 2.4 and 2.5, Table 1\*\*). Differences in MAR between ideal and compacted cores were only 0.18%.*

*With this, we have proven that compaction does not affect MAR and marsh ages when the $^{210}$Pb models are applied using cumulative mass instead of depth. As we used this methodology to estimate our CAR and stocks, we can confirm that our results are valid.*

*We also provide two papers (Gifford & Roderick, 2003; Wendt & Hauser, 2013) where the use of a single equivalent soil mass layer from the surface, or the use of cumulative mass coordinated, is described and used to facilitate organic carbon quantification in soil organic layers.*

*In our current revision, we have addressed this point in ln 161-163, where we have (a) indicated that our 210Pb-derived accumulation rates are calculated using cumulative mass, (b) have provided a reference to Gifford and Roderick (2003), and (c) have made clearer (as per the reviewer's request) that we have followed a method previously used in Gailis et al. (2021):*

*"Furthermore, when we have estimated $^{210}$Pb-derived accumulation rates (Figure 6), we have done so in terms of cumulative mass (g cm$^{-2}$) instead of depth (e.g. Gifford and Roderick, 2003). When we do need to account for compaction (e.g. Figure 3), we use a compaction factor as described in Gailis et al. (2021), estimated for each core by dividing the length of core penetration by the length of core recovered (Table A1)."*

Chastain, S. G., Kohfeld, K. E., Pellatt, M. G., Olid, C., and Gailis, M.: Quantification of Blue Carbon in Salt Marshes of the Pacific Coast of Canada, Biogeosciences Discuss. [preprint], https://doi.org/10.5194/bg- 2021-157, in review, 2021.

[Figure]

*Figure 2.5.* Ideal and compacted $^{210}$Pb concentration profile.

[Figure]

*Figure 2.6.* Age-depth (expressed as cumulative mass) model obtained after applying the CF:CS model for an ideal (uncompacted) and compacted core.

**Chastain, S. G., Kohfeld, K. E., Pellatt, M. G., Olid, C., and Gailis, M.: Quantification of Blue Carbon in Salt Marshes of the Pacific Coast of Canada, Biogeosciences Discuss. [preprint], https://doi.org/10.5194/bg- 2021-157, in review, 2021.**

**Table 1.** *Comparison of MAR between the ideal and compacted $^{210}$Pb profile.*

| Core | Total depth (cm) | Cumulative mass (g cm$^{-2}$) | MAR (g cm$^{-2}$ yr$^{-1}$) |
|---|---|---|---|
| Ideal | 30 | 30.9 | 0.2000 |
| Compacted | 15 | 30.9 | 0.2004 |

***\*\*Please note: We will gladly make the excel file containing these calculations available to the editor upon request***
* * *
**3. Reviewer initial comment**
**Shouldn't the regression for the relationship of %LOI and %C be forced through zero? With a negative intercept a sample with no organic matter, thus 0% LOI would have a negative amount of carbon – an impossibility.**

*Thank you for pointing this out. The relationship between %C and %LOI suggests that we measure zero %C in samples where LOI is not completely zero (below approximately 10% LOI). Although negative values of %C are obviously not possible, forcing the equation through zero would overestimate %C in these low LOI samples. Therefore, all calculations producing a negative value for %C were adjusted to zero %C. This occurred in 41 of 835 samples measured. Our methods have been clarified to reflect this change using the following equation, setting any negative %C value resulting from the use of a negative intercept equal to zero …*

**REVIEWER COMMENT ON AUTHOR STATEMENT AND REVISED TEXT**
**AUTHORS DO NOT SEEM TO UNDERSTAND THE COMMENT – FORCING THE LINEAR REGRESSION THROUGH ZERO DOES NOT MEAN SIMPLY DROPPING THE INTERCEPT AFTER OBTAINING THE REGRESSION MODEL. WHEN RUNNING THE REGRESSION ONE SIMPLY CHOOSES NO INTERCEPT FOR THE MODEL, THUS A NEW REGRESSION EQUATION IS REQUIRED. DOING THIS WOULD SIMPLY MEAN THAT 0% LOI IS EQUIVALENT TO 0% C. AUTHORS HAVE NO REASON TO CONCLUDE THAT FORCING THE REGRESSION THROUGH ZERO WOULD OVERESTIMATE THE %C.**

*We understood the original purpose of the reviewer comment, which requested that we recalculate the regression relationship between %LOI and %C so that it is forced through zero and then, accordingly, revise all estimates (and figures) of %C, Carbon stocks, soil carbon densities, and carbon accumulation rates, etc. However, we disagree with the original premise of this comment, i.e. that the relationship between %LOI and %C must have an intercept of zero.*

*Several previous studies have indicated that LOI has the potential to overestimate soil carbon because the ignition process drives off both organic matter as well as water bound in any clay minerals that are present in the sample (e.g. Howard, 1966; Howard and Howard, 1990; Santisteban et al. 2004).  Howard (1966) originally showed that the intercept for zero %C was actually 2% LOI in the soils they examined. Santisteban et al. (2004) produced an intercept of -1.83 (%OC = 0.634 LOI$_{550}$ (%) – 1.83) and showed that the intercept depended on the type of soils compared. Poppe and Rybczyk (2021) used a polynomial relationship to account for the lower %C values at small values of %LOI and still produced an intercept of –0.4496 (%OC = 0.0035 %LOI$^2$ + 0.4135%LOI – 0.4496). In their revision of estimates of global carbon stocks, Ouyang and Lee (2020) also produced an empirical relationship with a non-zero intercept for salt marshes (%OC = 0.52(%LOI) – 1.17).*

**Chastain, S. G., Kohfeld, K. E., Pellatt, M. G., Olid, C., and Gailis, M.: Quantification of Blue Carbon in Salt Marshes of the Pacific Coast of Canada, Biogeosciences Discuss. [preprint], https://doi.org/10.5194/bg- 2021-157, in review, 2021.**

*In summary, these studies indicate that there is no process-based reason for forcing this regression equation through zero, and in fact, the nature of the loss-on-ignition method suggests that we are more likely to expect there to be some small value of %LOI when %C reaches zero. As a result, forcing the intercept through zero could artificially inflate the estimates of %OC at low values of %OC. Most of these publications also suggest that soil-specific, empirical equations are the best approach for determining %OC from %LOI.* ==*As such, we have kept the original equation and left the text as written.*==

*Finally, we note that, ultimately, the choice of these two equations makes very little difference to our final estimation of carbon accumulation rates. Below we compare the average carbon accumulation rates (+/- SE) for the six cores on which we conducted 210Pb measurements and therefore also estimated carbon accumulation rates. We show that forcing the regression equation through zero actually makes only a minimal difference to the final estimates of CAR, as differences are within the estimated standard error for each core (Figure 3.1).*

[Figure]

*Figure 3.1: Comparison of carbon accumulation rates estimated to basal peat layer using original and proposed (zero-intercept) regression equations, demonstrating that both options are within the estimated standard error for each core. Dotted blue line shows the regression line between these two calculations, and the solid blue line represents a 1:1 line. Regression through zero intercept was: %C = 0.411\*%LOI ($R^2$ = 0.98). Original regression as stated in paper: %C = 0.44 %LOI − 1.9 ($R^2$ = 0.96).*

**BG-2021-157 – 2ᴺᴰ RESPONSE TO REVIEWERS - 2 September 2022**

**Chastain, S. G., Kohfeld, K. E., Pellatt, M. G., Olid, C., and Gailis, M.: Quantification of Blue Carbon in Salt Marshes of the Pacific Coast of Canada, Biogeosciences Discuss. [preprint], https://doi.org/10.5194/bg- 2021-157, in review, 2021.**
* * *
**4. Reviewer initial comment**
Clarification of and distinction amongst the terms "topsoil", "humus" and "peat" is needed. What is "topsoil" in a marsh? This term is not commonly used for wetland soils. The manuscript states see "Supplemental Information", but there is no explanation there. Also, the term "humus" is seldom used in wetland soils. Presumably it plant litter that is gradually broken down with depth? A bit of explanation would be helpful, even if just in a footnote to the Appendix table.

*We take this point and have changed the term "topsoil" (which was used to describe the fibrous organic material within and below the root zone) as "peat." However, we have kept use of the term "humus" as term that has been used as a descriptor in other salt marsh publications (e.g. Goni and Thomas, 2000; Santin et al. 2008)*

**REVIEWER COMMENT ON AUTHOR STATEMENT AND REVISED TEXT**
**AUTHORS ACTUALLY DELETED HUMUS FROM THE TEXT, BUT NOT THE TABLES. AUTHORS WILL NEED TO FIND AN ALTERNATIVE TERM FOR HUMUS – ONE THAT IS WIDELY USED BY THOSE WORKING WITH SALT MARSH SOILS, NEITHER PAPER CITED SUPPORTS THEIR USE OF HUMUS. SANTIN ET AL NEVER USE THE TERM HUMUS – THEIR STUDY IS ABOUT HUMIC ACIDS. GONI AND THOMAS SIMPLY USED THE TERM HUMUS TO IDENTIFY A PARTICULAR SIZE FRACTION OF ORGANIC MATTER, NOT AN ENTIRE PORTION OF THE SOIL.**

*We used the term "soil" to describe the surface organic layer that is distinct from the underlying peat layer. This is defined in Table A2 and in Figure A1.*

**5. Reviewer initial comment**
Line 518- Why would tidal amplitude be a driver of methane emissions? The paper cited on this line (Poffenbarger et al. 2011) reports that salinity, as a proxy for marine sulfates, is an important correlate.

*We appreciate this comment and we can replace the Poffenbarger et al. (2011) publication in this context, as there are several better citations that have measured changes in methane emissions associated with tidal activity and sea level rise on LN 608: (e.g. Abdul-Aziz et al. 2018; Huang et al. 2019; Huertas et al. 2019; Li et al. 2021; Wei et al. 2020.).*

**REVIEWER COMMENT ON AUTHOR STATEMENT AND REVISED TEXT**
**THE TEXT RELATED TO METHANE SHOULD BE DELTED ENTIRELY. AUTHORS WRITE "THE MESOTIDAL NATURE OF SOME OF THESE MARSH LOCATIONS COULD MEAN THAT SOME OF THESE MARSHES EMIT SUBSTANTIAL CONTRIBUTIONS OF METHANE, WHICH MAY COUNTER THEIR EFFECTS AS C SINKS (E.G. ABDUL-AZIZ ET AL. 2018; HUANG ET AL. 2019; HUERTAS ET AL. 2019; LI ET AL. 2021; WEI ET AL. 2020)." THESE PAPERS ARE ABOUT DURATION OF FLOODING – GREATER TIDAL RANGES MEAN LOWER HYDROPERIODS, NOT LONGER. THUS ONE WOULD EXPECT LESS METHANE PRODUCTION AND CERTAINLY OXIDATION OF THE METHANE THAT IS PRODUCED. NOTE THAT CHMURA FOUND NEGLIBLE METHANE EMISSIONS IN A MACROTIDAL MARSH. NOTE SOME OF THE REFERENCES ARE IDENTIFIED BY URLS ONLY ACCESSIBLE TO THE SIMONE FRASER UNIVERSITY SYSTEM!**

*Thank you for this clarification. The goal of this text is to point out that some of these marshes are BRACKISH, at least seasonally, and that brackish marshes can emit methane and therefore affect the overall*

*carbon sequestration potential. We did not intend to delve into a specific discussion of role of tidal range / duration on methane emissions. Rather, we wish to indicate that future work on understanding the carbon/greenhouse gas dynamics requires further investigation in these systems. The references provided – along with the original Poffenbarger et al. (2011) paper - support this point. We have therefore changed the work "mesotidal" to "brackish" in this sentence (LN 579-581).*

*REFERENCES CITED*

*Arias-Ortiz, A., Masqué, P., Garcia-Orellana, J., Serrano, O., Mazarrasa, I., Marbá, N., et al. (2018). Reviews and syntheses: 210Pb-derived sediment and carbon accumulation rates in vegetated coastal ecosystems - Setting the record straight. Biogeosciences, 15(22), 6791–6818. https://doi.org/10.5194/bg-15-6791-2018*

*Barlow, N. L. M., Long, A. J., Saher, M. H., Gehrels, W. R., Garnett, M. H., & Scaife, R. G. (2014). Salt-marsh reconstructions of relative sea-level change in the North Atlantic during the last 2000 years. Quaternary Science Reviews, 99, 1–16. https://doi.org/10.1016/j.quascirev.2014.06.008*

*De Vleeschouwer, F., Chambers, F.M. & Swindles, G.T. (2010): Coring and sub-sampling of peatlands for palaeoenvironmental research. Mires and Peat 7: Art. 1. (Online: http://www.mires-and-peat.net/pages/volumes/map07/map0701.php)*

*Frew, Craig (2014) Geomorphological Techniques, Chap. 4, Sec. 1.1 British Society for Geomorphology*

*Gifford, R. M., & Roderick, M. L. (2003). Soil carbon stocks and bulk density: Spatial or cumulative mass coordinates as a basis of expression? Global Change Biology, 9(11), 1507–1514. https://doi.org/10.1046/j.1365-2486.2003.00677.x*

*Howard PJA (1966) The carbon-organic matter factor in various soil types. Oikos 15:229-236*

*Howard, PJA and DM Howard (1990) Use of organic carbon and loss-on-ignition to estimate soil organic matter in different soil types and horizons, Biol Fertil Soils (1990) 9:306-310.*

*Ouyang, X., Lee, S.Y. Improved estimates on global carbon stock and carbon pools in tidal wetlands. Nat Commun 11, 317 (2020). https://doi.org/10.1038/s41467-019-14120-2*

*Poppe KL, Rybczyk JM (2021) Tidal marsh restoration enhances sediment accretion and carbon accumulation in the Stillaguamish River estuary, Washington. PLOS ONE 16(9): e0257244. https://doi.org/10.1371/journal.pone.0257244*

*Pritchard, D. W. (1967). "What is an estuary: physical viewpoint". In Lauf, G. H. (ed.). Estuaries. A.A.A.S. Publ. Vol. 83. Washington, DC. pp. 3–5. hdl:1969*

*J. Santisteban, R. López, E. Pamo, C. Dabrio, M. Zapata, M. Gil-García, et al., Loss on Ignition: A Qualitative or Quantitative Method for Organic Matter and Carbonate Mineral Content in Sediments?, Journal of Paleolimnology 2004 Vol. 32, DOI: 10.1023/B:JOPL.0000042999.30131.5b*

*Smeaton, C., Barlow, N. L. M., & Austin, W. E. N. (2020). Coring and compaction: Best practice in blue carbon stock and burial estimations. Geoderma, 364(January). https://doi.org/10.1016/j.geoderma.2020.114180*

*Wendt, J. W., & Hauser, S. (2013). An equivalent soil mass procedure for monitoring soil organic carbon in multiple soil layers. European Journal of Soil Science, 64(1), 58–65. https://doi.org/10.1111/ejss.12002*